# Reinforcement Learning for Heterogeneous DAG Scheduling with Weighted Cross-Attention

## Abstract

Efficient scheduling of directed acyclic graphs (DAGs) in heterogeneous environments is challenging due to diverse resource capacities and intricate dependencies. In practice, the need for adaptability across environments with varying resource pools, task types, and other settings, alongside rapid schedule generation, complicates these challenges. We propose WeCAN, an end-to-end reinforcement learning framework for heterogeneous DAG scheduling featuring task-resource compatibility. WeCAN rapidly generates schedules through single-pass network inference. Leveraging the weighted cross-attention layer, WeCAN utilizes all available environment information while preserving adaptability across diverse heterogeneous environments. Moreover, we analyze the optimality gap inherent in list-scheduling-based methods, revealing their inability to guarantee optimal solutions and their reduced performance in certain cases. Under the single-pass setting, we develop a method to enable skip actions, addressing this gap without sacrificing computational efficiency. Our approach delivers robust performance and adaptability, outperforming state-of-the-art methods across diverse datasets.

## 1 Introduction

Task scheduling problems are critical for optimizing computational performance in domains such as data centers (Mao et al., 2019), distributed systems, and large-scale cloud platforms (Lin et al., 2024). These problems are often modeled using directed acyclic graphs (DAGs), where nodes represent tasks and edges specify dependencies. The objective is typically to minimize the makespan (total completion time) subject to task dependencies and resource constraints. It is well known that DAG scheduling is an NP-hard problem even in homogeneous settings (Hartmanis, 1982). In heterogeneous environments, tasks need to be assigned to suitable pools (computational resources) possessing differing capacities and specific task-pool compatibility coefficients. This heterogeneity adds significant complexity to scheduling.

Traditional approaches often employ heuristics, such as list scheduling (Graham, 1969) which assigns tasks to pools iteratively based on priority scores. These scores are often calculated using computationally inexpensive metrics, such as the critical-path length or the number of remaining operations (Haupt, 1989). Variations include Tetris (Grandl et al., 2014) using dynamic scores that change during scheduling, or HEFT (Topcuoglu et al., 2002) based on inserting tasks into an existing timeline. However, the design of such heuristics often relies heavily on human expertise and struggles to fully utilize all available problem information effectively.

The application of machine learning (ML) to combinatorial optimization (CO) problems, originating with Hopfield & Tank (Hopfield & Tank, 1985), has gained significant attention in recent years. As most CO problems establish a one-to-one correspondence between solutions and action sequences, most ML approaches learn policies to construct solutions sequentially (Kool et al., 2019; Kwon et al., 2020; Liu et al., 2024; Zhou et al., 2024). Alternative approaches include learning policies to refine existing solutions (Wu et al., 2021; Ma et al., 2021; 2024) or simplify problem instances (Li et al., 2021; Hou et al., 2023; Ye et al., 2024). Researchers have also proposed reinforcement learning (RL) based methods for scheduling problems (Mao et al., 2016; 2019; Park et al., 2021; Paliwal et al., 2020; Sun et al., 2021; Gagrani et al., 2022; Sun et al., 2024; Li et al., 2024).

**Neural DAG Schedulers.** For DAG scheduling problems, Zhang et al. (2020) develop an approach to generate solutions by using a graph neural network (GNN) to encode state and selecting tasks sequentially. Zhou et al. (2020) develop a method that learns a policy to iteratively refine an existing solution. Wang et al. (2021) introduce a bi-level optimization approach applying RL to add auxiliary edges and proposing heuristics on the modified graph. Typically, these methods require multi-round neural network processing for generating a solution, limiting speed for time-sensitive applications. Jeon et al. (2023) introduce an approach using the Gumbel-TopK trick to generate priorities which are subsequently fed to the list scheduling algorithm for a feasible schedule. While their single-pass network inference enables rapid solution generation, their architecture does not consider compatibility coefficients or pool allocation, remaining challenges in highly heterogeneous settings. Furthermore, the reliance on generation maps like list scheduling in many existing approaches introduces an inherent optimality gap, limiting their ability to consistently find optimal solutions. The skip action in Mao et al. (2016) mitigates this gap, but their design relies on the time-consuming multi-round network processing and is difficult to adapt to the single-pass setting.

**Neural DAG Schedulers for Heterogeneous Environments.** Researchers have also proposed reinforcement learning (RL) based methods for scheduling in heterogeneous environments (Wu et al., 2018; Ni et al., 2020; Grinsztajn et al., 2021; Zhou et al., 2022; Zhadan et al., 2023; Wang et al., 2025). For example, Zhou et al. (2022) propose an approach that selects tasks sequentially by network and assigns them to pools using heuristics. Their method represents compatibility coefficients by averaging them across pools, potentially losing fine-grained information. Similar strategies for embedding compatibility coefficients are proposed in Zhadan et al. (2023); Wang et al. (2025). Other works (Wu et al., 2018; Grinsztajn et al., 2021; Jeon et al., 2023) handle compatibility coefficients using representations like one-hot embedding of task types or fixed-dimensional vectors. However, these representations often depend on a fixed number of task types or pools, limiting their adaptability and flexibility to varying environment structures. Therefore, it remains a significant challenge to fully embed detailed environment information, including complex compatibility constraints, while maintaining adaptability across diverse and dynamically sized heterogeneous configurations.

To address these challenges, we propose an end-to-end reinforcement learning framework for heterogeneous DAG scheduling, particularly addressing problems featuring compatibility coefficients. Our main contributions are as follows. (1) We design an end-to-end reinforcement learning framework to rapidly solve heterogeneous DAG scheduling problems with diverse task-pool compatibilities. Our approach generates the solution with single-pass network inference, resulting in a processing time close to the heuristics; (2) We design a weighted cross-attention network (WeCAN), consisting of weighted cross-attention layers to capture environment information and a longest directed distance graph neural network to adapt the varying tasks dependence. The weighted cross-attention network fully utilizes available environment information and compatibility coefficients for embedding, while preserving adaptability and scalability across varying heterogeneous environment sizes, such as the number of pools and task types. This facilitates accurate evaluation of task features within the specific environment; (3) We perform an analysis of the solution spaces and the generation maps, and propose a criterion for checking their ability to achieve optimality. This analysis highlights the optimality gap inherent in widely used methods such as list scheduling, which can lead to a significant decrease in performance under some characteristic cases. We develop a method to enable the skip action in the single-pass setting, and demonstrate how this method theoretically closes this gap while retaining the computational efficiency, thereby underscoring the importance of skip. We also reveal which cases benefit the most from the skip action; (4) Empirical evaluations on the Computation Graphs and real-world TPC-H benchmarks validate our approach, demonstrating improved performance and significant gains in computational efficiency compared to state-of-the-art methods.

## 2 PRELIMINARIES

### 2.1 HETEROGENEOUS SCHEDULING PROBLEM

The task scheduling problem seeks a schedule that minimizes the objective function among all feasible schedules subject to given constraints. We focus on heterogeneous scheduling problems characterized by compatibility coefficients. A problem instance $\mathcal{P} = (V, E, C, t, \rho, \lambda, K_{acc})$ can be modeled by a DAG $G = (V, E)$ together with a resource pool set $C = \{c(1), ..., c(n_c)\}$. The node set $V$ and the edge set $E$ represent tasks and their precedence relations, respectively. Each node

(task) $v \in V$ has a processing time $t(v)$ and a resource demand vector $\rho(v)$. Each pool in $C$ has a resource capacity vector $\lambda(c)$. The compatibility coefficients $K_{acc}(v, c) \geq 0$ reflect the variations in task execution time across resource pools (Topcuoglu et al., 2002), with the actual execution time of task $v$ on pool $c$ proportional to the inverse of this coefficient. A schedule is represented by a function $x : V \rightarrow \mathbb{R} \times C$, which maps each task $v \in V$ to a pair $(s(v), c(v))$, representing its start time and assigned resource pool, respectively. In this paper, the objective function is the makespan $f$, defined as the latest task completion time. The problem can thus be formulated as:

$$\min_{x=(s,c)} \quad f(x) := \max_{v \in V}[s(v) + t(v)/K_{acc}(v, c(v))],$$

$$\text{s.t.} \quad s(v) + t(v)/K_{acc}(v, c(v)) \leq s(w), \, \forall(v, w) \in E,$$

$$\sum_{v \in F(t,c)} \rho(v) \leq \lambda(c), \, \forall t \geq 0, \, c \in C,$$

$$K_{acc}(v, c(v)) > 0, \, \forall v \in V,$$

$$s(v) \geq 0, \, \forall v \in V.$$

Here, $F(t, v)$ is the set of tasks on $c$ at time $t$. The last constraint ensures the start time nonnegative, while the first three types of constraints are interpreted as follows: (1) dependency constraints, requiring each task to start after all its predecessors complete; (2) resource constraints, ensuring at any time $t$, the total resource demand of tasks running on pool $c$ does not exceed its capacity $\lambda(c)$; and (3) compatibility constraints, requiring that a task $v$ can be assigned to a pool $c$ only if $K_{acc}(v, c) > 0$.

In the homogeneous setting where $K \equiv 1$ and $|C| = 1$, the third constraint is naturally satisfied. In contrast, heterogeneous environments with compatibility constraints pose additional challenges, requiring tasks to run on compatible resource pools and schedulers to adapt to diverse resource characteristics. These scheduling problems can be formulated as mixed integer linear programming (MILP) problems, as described in Appendix A. This formulation establishes a one-to-one correspondence between feasible schedules and MILP feasible solutions, enabling a unified framework for analyzing scheduling scenarios. Compared to the homogeneous case, these settings introduce additional constraints and significantly increase MILP size, posing challenges for effective scheduling.

## 2.2 LEARN TO SCHEDULE WITH MASK AND MAP

Most neural scheduling methods construct feasible schedules by sequentially assigning tasks to pools, rather than directly generating them as function $x : V \rightarrow \mathbb{R} \times C$. For a scheduling problem $X$, the scheduling process is modeled as a Markov decision process (MDP), with the state represented by $s_t$. At each decision step, the scheduler, using a neural network to implement $p_\theta(\pi_t|s_t, \pi_{<t})$, selects an action $\pi_t = (v_t, c_t)$ to execute task $v_t$ on pool $c_t$ with probability $p_\theta(\pi_t|s_t, \pi_{<t})$. The probability of a schedule is the product of action probabilities across all $T(\pi)$ decision steps:

$$p_\theta(\pi) = \prod_{t=1}^{T(\pi)} p_\theta(\pi_t|s_t, \pi_{<t}).$$

Generating a feasible schedule from the action sequence (orders) requires a generation map $S$. Given a problem distribution, the target of learning to schedule is to find a policy $p_\theta$ that minimizes the **loss function** $L(\theta) = \mathbb{E}_X \mathbb{E}_{\pi \sim p_\theta(\cdot|X)} f(S(\pi))$. The most widely used map in both heuristic and neural schedulers is the list scheduling map $S_{list}$. In list scheduling, the current time $t_{cur}$ is initialized to 0, and the scheduler updates masks to prevent selections that violate the dependency or resource constraints. The scheduler repeatedly selects an available action $(v, c)$ by action sequence or other rules, setting $c(v) = c$ and $s(v) = t_{cur}$, until all actions are masked. When all actions are masked, the scheduler advances $t_{cur}$ to the next task completion time, updates the masks, and continues to select actions. During this map, masks for resource constraints are constructed from the remaining resources, while masks for dependency constraints are maintained by a dependency table. In the MDP model, the masks set the probabilities of infeasible actions to zero, ensuring feasibility.

# 3 LEARN TO SCHEDULE FOR HETEROGENEOUS ENVIRONMENT

To effectively schedule tasks across diverse heterogeneous resource environments, we propose the weighted cross-attention network. The overall architecture of our framework is illustrated in Figure

Figure 1: The framework of our architecture. The network produces a action sequence while the generation map gives the feasible schedule.

1, which consists of two parts: once network processing and a generation map for creating feasible schedules. The weighted cross-attention (WeCA) layer in the network processes environmental information, enabling adaptation to various pool features, while maintaining adaptability regarding the number of task types and resource pools. By introducing skip action with once network processing, the generation map forms a surjection capable of representing the optimal solution without multi-round computation, resulting in improved performance in cases involving heavy tasks characterized by long duration and high resource demand. Below, we describe each of the components in detail.

## 3.1 WEIGHTED CROSS-ATTENTION BASED GRAPH ENCODER

**Weighted Cross-Attention:** Each task node $v$ is represented by attribute vector $(t(v), \rho(v))$ and resource pool $c$ by capacity vector $\rho(c)$. Initially, an encoder generates task embeddings $\boldsymbol{f}_v$ and pool embeddings $\boldsymbol{f}_c^c$ using a multilayer perceptron (MLP). These initial embeddings capture only isolated characteristics of tasks and pools. Consequently, tasks with identical attribute vectors receive the same initial embedding, yet their suitability varies across pools due to differing compatibility coefficients $K_{acc}(\cdot, \cdot)$. To capture these contextual characteristics within the heterogeneous environment and ensure that the architecture remains adaptable with respect to the number of task types and pools, we apply the WeCA layer. Based on the Transformer architecture (Vaswani et al., 2017) and skip-connection (He et al., 2016), our WeCA layer executes message passing between tasks and pools:

$$\boldsymbol{q}_v = \boldsymbol{W}^Q \boldsymbol{f}_v, \quad \boldsymbol{K}^c = \boldsymbol{W}^K [\boldsymbol{f}_{c(1)}^c, ..., \boldsymbol{f}_{c(n_c)}^c], \quad \boldsymbol{V}^c = \boldsymbol{W}^V [\boldsymbol{f}_{c(1)}^c, ..., \boldsymbol{f}_{c(n_c)}^c],$$

$$\boldsymbol{g}_v = \boldsymbol{f}_v + \frac{\text{softmax}(\boldsymbol{q}_v^T \boldsymbol{K}^c)}{\sqrt{d}} \text{diag}\{K_{acc}(v, c(1)), ..., K_{acc}(v, c(n_c))\} \boldsymbol{V}^c,$$

where $\{c(1), ..., c(n_c)\}$ is the pool set, $K_{acc}$ is the compatibility coefficient and $d$ is the dimension of $\boldsymbol{q}_v$. This design integrates the compatibility coefficient as the attention bias without imposing fixed dimensionality constraints on the network architecture. Crucially, task $v$ cannot run on pool $c$ if and only if $K_{acc}(v, c) = 0$. Thus, incompatible pool assignments are inherently masked in attention calculation. The WeCA layer evaluates potential pools for each task and aggregates their features, weighted by suitability. A larger $K_{acc}(v, c)$ indicates that task $v$ runs faster on pool $c$, indicating a better compatibility. Therefore, for each task, the WeCA layer effectively gathers and integrates information primarily from its most compatible pools. Unlike approaches relying on fixed-size embeddings, this mechanism maintains adaptability, allowing the framework to evaluate context-dependent resource capacity for each task more realistically within the heterogeneous environment.

Notably, the acceleration coefficient is multiplied outside the softmax normalization function rather than the inside version of logarithmic form, which is described in Appendix G. This placement aids in embedding information about a task's overall compatibility. Consider a scenario of two pools with identical capacity and two tasks $v_1, v_2$ with the same attribute. Suppose $v_1$ is compatible with only one pool, while $v_2$ compatible with both pools. Despite the identical attribute, these tasks differ significantly in their environmental compatibility. If the inside placement were taken, the normalization effect could lead to the same embeddings for both tasks, failing to distinguish their different compatibility profiles. Conversely, the outside placement better reflects a task's overall compatibility within the environment, resulting in a more accurate and distinguishable embedding.

**Longest Directed Distance based Graph Neural Network (LDDGNN):** To embed the dependency, a graph neural network (GNN) is employed. Standard GNN architectures often struggle with long-range dependencies and directed acyclic graphs embedding. Inspired by Graphormer (Ying et al., 2021) and Topoformer (Gagrani et al., 2022), we design the network through attention masks and biases, but based on longest directed distance (LDD) $d_e$ which is defined as the signed length of the longest directed path. Our LDDGNN (detailed in Appendix G) update the node embeddings through $L$ attention layers, each comprising a multi-head attention (MHA) sub-layer and a node-wise MLP:

$$\boldsymbol{q}_v^{l,j} = \boldsymbol{W}_{l,j}^Q \boldsymbol{h}_v^{l-1}, \boldsymbol{k}_v^{l,j} = \boldsymbol{W}_{l,j}^K \boldsymbol{h}_v^{l-1}, \boldsymbol{v}_v^{l,j} = \boldsymbol{W}_{l,j}^V \boldsymbol{h}_v^{l-1},$$

$$\hat{\boldsymbol{h}}_v^l = \boldsymbol{h}_v^{l-1} + \text{concat}_j \big( \sum_{w \in V} [(\boldsymbol{q}_v^{l,j})^T \boldsymbol{k}_w^{l,j} \cdot b_{d_e(v,w)} \cdot M_{v,w}^j] \boldsymbol{v}_w^{l,j} \big),$$

$$\boldsymbol{h}_v^l = \hat{\boldsymbol{h}}_v^l + \text{MLP}_{(l)}(\hat{\boldsymbol{h}}_v^l).$$

Here, $M_{v,w}^j$ is the attention mask of head $j$ for the pair $(v, w)$ which is based on LDD, and $b_{d_e(v,w)}$ is a learnable bias embedding based on the LDD $d_e(v, w)$. This LDD-aware attention mechanism helps capture both directed and undirected dependency structure within the task graph.

## 3.2 DECODER

The schedule is constructed by the action sequence. The decoder should output probabilities (or scores) for all actions. For improving scalability, we employ a non-auto-regressive decoder (comparison with auto-regressive one in Appendix B), where the action probability $p_\theta(\pi_t | s_t, \pi_{<t})$ depends only on the initial state $s_1$ and reduces to $p_\theta(\pi_t | s_1)$. Similarly to the encoder, the decoder utilizes WeCA layers (details in Appendix G) to update the final task embeddings $\boldsymbol{h}_v$ and pool embedding $\boldsymbol{h}_c^c$. Finally, the decoder calculates the score of each action $(v, c)$ by weighted inner product:

$$\hat{\boldsymbol{q}}_v = \hat{\boldsymbol{W}}^Q \boldsymbol{h}_v, \quad \hat{\boldsymbol{k}}_c = \hat{\boldsymbol{W}}^K \boldsymbol{h}_c^c,$$

$$u_{(v,c)} = \hat{\boldsymbol{q}}_v^T \hat{\boldsymbol{k}}_c + \log(K_{acc}(v,c)).$$

At each step, the generation map identifies invalid actions and applies masks to set their score $u_{(v,c)}$ to $-\infty$, then calculates the probability for each action $\pi = (v, c)$ by a normalization:

$$p_\theta(\pi) = \frac{\exp u_\pi}{\sum_\pi \exp u_\pi}. \tag{1}$$

Moreover, to mitigate potential optimality gap, we should introduce the skip action, defined as advancing time $t_{cur}$ to the next task completion time. However, a fixed skip score will lead to endless idling, while generating a dynamic score by network in each step eliminates the single-pass efficiency. For addressing this problem, we use the network to produce skip coefficients $u_a \geq 0, u_b \geq 0$ and $u_c$ which are derived from an MLP with the average of task embeddings $\overline{\boldsymbol{h}}_v$ and the average of pool embeddings $\overline{\boldsymbol{h}}_c^c$ as input, and then calculate skip score as $u_{\pi_{skip}} = u_a(1 - \frac{k}{2n})^{u_b} + u_c$. Here, $k$ is the number of actions taken and $n$ is the number of tasks. This approach fixes the optimality gap and prevents the skip action from overly prioritized, while remaining the single-pass efficiency.

Here we show the full process in Algorithm 1. The following theorem guarantees that the scheduler produces a feasible schedule in finite steps. Moreover, it can produce schedules in any feasible order and then includes the optimal solution. We provide the details of the proof in Appendix A.

**Theorem 1.** *i) Algorithm 1 generates a feasible solution within $2n$ steps where $n$ is the number of nodes. ii) Algorithm 1 assigns positive probabilities to at least one optimal solution and all feasible orders. iii) Without the skip action, statement ii) does not hold for some problem $X$. iv) For each problem $X$, there exist scores $\{u_{(v,c)}\}_{v \in V, c \in C}$ and $u_a, u_b, u_c$ enabling an optimal solution by greedily selecting the action with the highest $p_\theta(\pi)$ in Algorithm 1.*

## 3.3 TRAINING

Given scheduling problems $X$ from a distribution, the goal is to minimize the expectation of makespan $\hat{f}(\pi) = f(S(\pi))$ with our policy $p(\pi|X)$. We use REINFORCE (Williams, 1992) to update the parameters with learning rate $\alpha$:

$$\nabla_\theta L(\theta) = \nabla_\theta \mathbb{E}_X \mathbb{E}_{p_\theta(\pi|X)} \left[ (\hat{f}(\pi) - b(X)) \nabla_\theta \log p_\theta(\pi|X) \right],$$

$$\theta \leftarrow \theta - \alpha \nabla_\theta L(\theta).$$

---

**Algorithm 1** Solution generation through the neural scheduler

---

**Input:** DAG scheduling problem $(G, V, C, \rho, t, \lambda, K_{acc})$ with $n$ tasks and $n_c$ pools.
Initialize the dependency table and current time $t_{cur} = 0$.
Perform WeCAN to get the scores $u_\pi$ for each action $\pi = (v, c)$ and coefficients $u_a, u_b, u_c$.
**while** unfinished node remains **do**
    Mask action $(v, c)$ for all tasks $v$ which have been started or have unfinished dependency.
    Mask action $(v, c)$ by the resource requirement and compatible requirement.
    Calculate skip score $u_{\pi_{skip}} = u_a(1 - \frac{k}{2n})^{u_b} + u_c$, where $k$ is the number of steps taken.
    Mask the skip action if no running tasks on all pools.
    Calculate the probability by (1) and select (sample) an action $\pi$ by the probability.
    **if** the selected action $\pi = (v, c)$ is not the skip action **then**
        Start the task $v$ on pool $c$. Setting $s(v) = t_{cur}$ and $c(v) = c$.
    **else**
        Find the next task completion time $t_{next} > t_{cur}$ and set $t_{cur} = t_{next}$.
    **end if**
    Update the dependency table and current resource.
**end while**

---

Here, $b(X)$ is a baseline for reducing the variance and is taken as average rewards.

## 4 SCHEDULE WITH SKIP TO FIX GAPS

We design a method to introduce the skip action, which is defined as advancing time $t_{cur}$ to the next task completion time, in our single-pass network framework. In this section, we demonstrate and analyze how the skip action enhances scheduling performance in characteristic cases by closing the optimality gap. We also provide a criterion for determining whether a generation map can generate optimal schedules and investigating which cases skip benefits most.

### 4.1 REDUCED SPACE

The scheduling problem can be formulated as an MILP problem. In Section 2, the concept of feasible schedule is introduced. For brevity, we denote the space of all such schedules as the original space $A$. In the MDP formulation, action sequences do not directly generate a feasible schedule; instead, they correspond to schedule orders in a discrete space $B$ and the scores generated by the neural network represent point (in greedy way) or distributions (in sampling way) on $B$, as detailed in Appendix A. We refer to the discrete space $B$ as the reduced space, whose point can be characterized by all task orders and pool allocations. Naturally, there exists a map $T : A \to B$ that associates each feasible schedule with its schedule order in $B$. A feasible schedule order is the legal order lying in the feasible reduced space $B_f = T(A) \subset B$. Obtaining a feasible schedule from a schedule order in $B_f$ requires a generation map to $A$. Most heuristic and neural schedulers employ the list scheduling $S_{list} : B \to A$ described in Section 2, which is effective for generating sub-optimal schedules. However, we prove in Appendix A that, in certain cases, list scheduling cannot yield an optimal schedule. We further find this theoretical gap leads to a significant performance decrease in some cases, as it prioritizes tasks meeting current resource availability. This prioritization creates a preference for delaying resource-intensive tasks and significantly impacts cases with heavy tasks featured by extreme resource demands and running times. Our experimental results in Appendix C further show that as the rate of heavy task increases, the gap also increases.

### 4.2 FINDING OPTIMUM WITH SURJECTION

Our theoretical analysis and illustrative examples show that the inherent optimality gap of $S_{list}$ arises because $TS_{list}$ is neither the identity nor surjective. As a result, $S_{list}$ maps multiple points in $B$ to the same point in $A$, shrinking its image and excluding the optimal solution. To construct a map $S : B_f \to A$ whose image includes the optimal solution, a projection between $A$ and $B_f$ associating each subspace of $A$ with a point in $B_f$ helps. Such maps must satisfy the following requirements.

**Assumption 1.** *(1)* $TS = I$. *This ensures that $T$ and $S$ provide an embedding of feasible reduced space $B_f$ into $A$, while $ST$ serves as a projection.*

*(2) $f(v) \geq f(ST(v)), \forall v$. The map $ST(v)$ minimizes the objective function $f$ within subspace $T^{-1}(T(v))$.*

These properties ensure that $S$ embeds the feasible reduced space as a projection space for $A$, and the following theorem shows that such a map $S$ can be used to find the optimal solution.

**Theorem 2.** *Let $S : B \rightarrow A$ be a generation map satisfying Assumption 1. For any optimal solution $x$, there exists an optimal solution $y \in Image(S)$ and $T(y) = T(x)$.*

We provide the proof and construct a map $S_n$ satisfying Assumption 1 in Appendix A. The map $S_n$ extends list scheduling by relaxing certain constraints to allow waiting. However, although $S_n$ includes the optimal solution, it also produces many inferior solutions scattered across the reduced space. This is because the mapping $S_n$ allows arbitrary idle time, resulting in a large variance of makespan, hindering the training of scheduling policies. Therefore, instead of using $S_n$ directly, we enlarge $B_f$ to include skip actions, lift $S_{list}$ to a map $S$ on the enlarged space, and modify $T$ accordingly; the resulting $(B_f, T, S)$ meet Assumption 1. Moreover, our design clusters most poor solutions in the high-$u_a$, high-$u_c$ region, because excessive skips typically arise from large values of $u_a$ and $u_c$, rather than scattering them across the space; this concentration makes such regions easier to handle during training and reduces variance. Theorem 1 demonstrates that our design in the single-pass setting ensures that $TS$ is a surjection, enabling the generation of the optimal schedule while retaining high inference speed. Our experiments results in Appendix C further validate the effectiveness of this design, revealing that the skip benefits more when the percentage of heavy tasks increases. Therefore, our design enhances the ability to find the optimal schedules, leading to enhanced performance in heavy task cases without sacrificing computational efficiency or significantly increasing the variance in training outcomes by clustering poor solutions.

## 5 NUMERICAL EXPERIMENTS

### 5.1 DATASET AND ENVIRONMENT SETTINGS

**Datasets.** i) *TPC-H dataset:* a dataset that comprises real-world DAG tasks derived from industrial queries. We use the version sorted by Wang et al. (2021), and add additional random memory constraints and task types (each with a group of compatibility coefficients). The problems in TPC-H-30, TPC-H-50, and TPC-H-100 contain 275, 459, and 918 tasks on average. ii) *Computation Graphs dataset:* a synthetic dataset generated using approaches from Jeon et al. (2023), comprising computation graphs for neural networks arising in ML compilers, including layer graphs, Erdős-Rényi graphs, and stochastic block model graphs. Each problem contains 500 tasks. Problems in both datasets are scheduled on three heterogeneous resource pools. The heterogeneous significantly multiple the problem size. Details of the two datasets and their problem size are shown in Appendix D.

**Baselines.** We compare our method with the following baselines: list scheduling algorithms, including critical path (CP), shortest first task (SFT), and most operations remaining (MOPNR); Tetris (Grandl et al., 2014), a dynamic list scheduling heuristic for multi-resource pool scheduling; HEFT (Topcuoglu et al., 2002), a non-list heterogeneous scheduling algorithm; **Two RL baselines**: PPO-BiHyb (Wang et al., 2021), a bi-level neural scheduler with beam search; and One-Shot (Jeon et al., 2023), a one-shot neural scheduler generating schedules sequentially based on list scheduling. For the 4 list scheduling algorithms, we apply three pool-selection rules and select the one with the best makespan.

We evaluate our method using two modes: greedy, which selects actions with the highest probability $p_\theta$ and sampling (S(n)), which generates $n$ samples based on $p_\theta$ within our accelerated environment and selects the schedule with the minimum makespan. For each experiment, we report the makespan, running time or relative improvement over the best heuristic baseline, with further experimental details provided in Appendices D, E, and H.

### 5.2 EXPERIMENTAL RESULTS

On the TPC-H dataset, WeCAN demonstrates up to 18.1% makespan improvement over the best heuristic and 7.7% over the best neural baseline, with superior performance in instances with

Table 1: Experimental results on TPC-H datasets with standard deviation among random seed.

| | TPC-H-30, 3 pools | | TPC-H-50, 3 pools | | TPC-H-100, 3 pools | |
| | MakeSpan | Time | MakeSpan | Time | MakeSpan | Time |
| --- | --- | --- | --- | --- | --- | --- |
| SFT | 27404 | 0.23 | 49172 | 0.78 | 84986 | 3.08 |
| MOPNR | 25052 | 0.30 | 43545 | 0.99 | 77362 | 3.34 |
| CP | 23869 | 0.29 | 41597 | 0.90 | 74364 | 3.35 |
| HEFT | 23177 | 0.18 | 39315 | 0.54 | 70137 | 1.86 |
| Tetris | 23170 | 0.21 | 38654 | 0.62 | 71296 | 2.13 |
| PPO-BiHyb | 21941 | 20.48 | 36333 | 55.74 | 67695 | 179.19 |
| One-Shot-S(256) | $20399 \pm 181$ | 2.26 | $35561 \pm 108$ | 4.16 | $66173 \pm 180$ | 9.85 |
| WeCAN-Greedy | 19578 | 0.15 | 33428 | 0.50 | 62587 | 1.72 |
| WeCAN-S(64) | $19053 \pm 28$ | 1.54 | $32912 \pm 40$ | 2.86 | $61662 \pm 118$ | 5.26 |
| WeCAN-S(256) | $\mathbf{18964 \pm 10}$ | 2.43 | $\mathbf{32814 \pm 47}$ | 4.39 | $\mathbf{61373 \pm 28}$ | 10.43 |

Table 2: Experimental results on Computation Graphs datasets with 500 tasks.

| | Erdős-Rényi | | Layer Graphs | | Stochastic Block | |
| | MakeSpan | Time | MakeSpan | Time | MakeSpan | Time |
| --- | --- | --- | --- | --- | --- | --- |
| SFT | 13317 | 0.81 | 16158 | 0.34 | 14408 | 0.53 |
| MOPNR | 12771 | 1.07 | 14714 | 0.38 | 13148 | 0.68 |
| Tetris | 13084 | 0.52 | 14271 | 0.44 | 13666 | 0.64 |
| CP | 12457 | 1.08 | 14797 | 0.40 | 13388 | 0.74 |
| HEFT | 11098 | 0.55 | 12428 | 0.75 | 11260 | 0.57 |
| PPO-BiHyb | 10795 | 65.51 | 11883 | 73.7 | 10885 | 73.7 |
| One-Shot-S(256) | $11071 \pm 76$ | 4.45 | $12277 \pm 49$ | 3.83 | $11377 \pm 40$ | 4.00 |
| WeCAN-Greedy | 10270 | 0.57 | 11173 | 0.26 | 10539 | 0.41 |
| WeCAN-S(64) | $10115 \pm 10$ | 3.21 | $10862 \pm 29$ | 3.07 | $10074 \pm 18$ | 3.06 |
| WeCAN-S(256) | $\mathbf{10083 \pm 13}$ | 4.94 | $\mathbf{10752 \pm 27}$ | 4.30 | $\mathbf{10019 \pm 12}$ | 4.58 |

300∼500 nodes and robust results for 1,000 nodes (see Table 1). Additionally, our results show that WeCAN excels at learning robust scheduling policies in complex heterogeneous environments, as evidenced by its performance across diverse TPC-H instances. Owing to single-pass neural processing, WeCAN-greedy achieves lower running time than PPO-BiHyb and comparable running time to One-Shot-greedy and heuristic baselines, while delivering superior makespan. This similarity in running time arises because, in heterogeneous environments, the generation map's runtime dominates for both WeCAN and One-Shot, approaching the minimum time required to generate a schedule. Furthermore, we conduct experiments in varying resource environments to evaluate generalization from a fixed training environment. Figure 2 shows that our WeCAN shows robust performance under varying environment fluctuations including pool number, pool type (feature), task number, and task type. This result validates our WeCAN effectively utilizes the environment feature while remaining robust and scalable for environment sizes. We also provide results on large-scale problems and more environment fluctuations in Appendix F.

In the Computation Graphs dataset, WeCAN demonstrates up to 13.4% makespan improvement over the best heuristic and 9.5% percent over the best neural baseline, with superior performance across different types of graphs (see Table 2). Given the prevalence of heterogeneous resource environments in ML compilers, our results demonstrate WeCAN's applicability for efficient scheduling in neural network compilation.

## 5.3 ABLATION STUDY

We conduct ablation experiments to evaluate the contributions of WeCAN's weighted cross-attention layers and longest directed distance graph neural network components. For WeCA layers, we test the inside version, WeCA layers only in decoder, its inside version, and WeCA layers skipped except in

Table 3: Ablation study results on TPC-H datasets with different architectures.

| | TPC-H-30 | | TPC-H-50 | |
| --- | --- | --- | --- | --- |
| | MakeSpan | Improvement | MakeSpan | Improvement |
| Tetris | 23170 | 0.0% | 38654 | 0.0% |
| WeCA + LDDGNN | **19908 ± 48** | **14.0 ± 0.2%** | **34260 ± 52** | **11.4 ± 0.1 %** |
| WeCA-inside + LDDGNN | 20729 ± 55 | 10.5 ± 0.2 % | 34980 ± 30 | 9.5 ± 0.1% |
| WeCA-decoder+ LDDGNN | 20234 ± 41 | 12.7 ± 0.2 % | 34815 ± 156 | 9.9 ± 0.4% |
| WeCA-decoder-inside+ LDDGNN | 21981 ± 195 | 5.1 ± 0.8 % | 36984 ± 72 | 4.3 ± 0.2% |
| WeCA-final-only + LDDGNN | 23066 ± 97 | 0.5 ± 0.3 % | 40308 ± 358 | -4.2 ± 0.9% |
| WeCA + GAT(forward) | 20747 ± 21 | 10.5 ± 0.1 % | 35224 ± 14 | 8.9 ± 0.1% |
| WeCA + GAT(bi-direction) | 20873 ± 7 | 9.9 ± 0.0 % | 35177 ± 20 | 9.0 ± 0.1% |

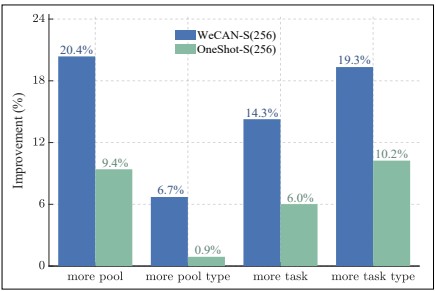

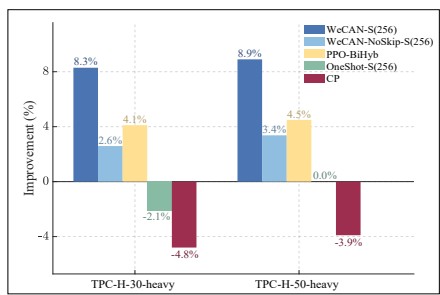

Figure 2: Evaluations of models on TPC-H-30 with different environment fluctuations under fixed training conditions. The percent of improvement over best heuristics are labeled.

Figure 3: Ablation study results on TPC-H datasets with heavy tasks. The percent of improvement over best heuristics HEFT are labeled.

the final decoder layer. For LDDGNN, we test a standard GAT (GAT forward) and a bidirectional GAT (GAT bidrection). All network variants share the same layer count and hidden dimensions, with fewer WeCA layers offset by additional LDDGNN layers. We train and test each model on TPC-H-30 and TPC-H-50. For each of 10 test problems, we generate 256 samples, computing mean makespan and relative improvement to the best heuristic (Tetris). Table 3 shows that replacing either component increases makespan and decreases relative improvement. The modified WeCA layers yields higher makespan, with skipping WeCA layers causing significantly worse performance, highlighting WeCA layers' importance in capturing task-resource pool compatibility. Both GAT variants yield higher makespan than LDDGNN, demonstrating the superiority of LDDGNN.

We conduct experiments to evaluate the skip-action mechanism on TPC-H with "heavy tasks" characterized by high resource demand and long processing time. We modify the TPC-H-30 and TPC-H-50 datasets by randomly replacing $1\%$ tasks with "heavy tasks". Figure 3 shows that HEFT (not list scheduling) achieves smaller makespan than the best list scheduling approach (CP). WeCAN with the skip action achieves lower makespan than its non-skipping variant and all other approaches. This result validates the optimality gap of list scheduling. It also reveals that the skip-action mechanism mitigates the optimality gap and increases the performance in cases with "heavy tasks".

## 6 CONCLUSION

This paper presents WeCAN, an end-to-end reinforcement learning framework for heterogeneous DAG scheduling with task-pool compatibility. Weighted cross-attention layers enable WeCAN to fully utilize environment information while adapting to diverse problem sizes. Introducing skip-action in the single-pass setting closes optimality gap of list scheduling and improves performance in heavy task cases. Our theoretical analysis further highlights its importance, especially when heavy-task proportions are large. Evaluations on TPC-H and Computation Graphs datasets demonstrate WeCAN's effectiveness for heterogeneous DAG scheduling with task-resource compatibility. Extending our WeCAN to address more complicated settings will be an interesting future research direction.

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

# A SUPPLEMENT THEORETICAL RESULTS

## A.1 MILP FORMULATIONS OF THE SCHEDULING PROBLEM

A heterogeneous scheduling problem $\mathcal{P} = (V, E, C, t, \rho, \lambda, K_{acc})$ can be formulated as

$$\min_{x=(s,c)} \quad f(x) := \max_{v \in V}[s(v) + t(v)/K_{acc}(v, c(v))],$$

$$\text{s.t.} \quad s(v) + t(v)/K_{acc}(v, c(v)) \leq s(w), \ \forall (v, w) \in E,$$

$$\sum_{v \in F(t,c)} \rho(v) \leq \lambda(c), \ \forall t \geq 0, \ c \in C,$$

$$K_{acc}(v, c(v)) > 0, \ \forall v \in V,$$

$$s(v) \geq 0, \ \forall v \in V.$$

where $F(t, c)$ is defined as

$$F(t, c) := \{v \in V | s(v) \leq t; \ s(v) + t(v)/K_{acc}(v, c(v)) > t; \ c(v) = c\}.$$

For the homogeneous scheduling problem $(V, E, t, \rho, \lambda)$ with $r$ types of resources, where $K \equiv 1$ and $|C| = 1$, the third constraint is naturally satisfied. $V$ is the task set with $|V| = n$ and $E$ is the edge set. Let constant $t_i$ be the processing time of task $i$, $\rho_l^i$ be the $l$-th type resource demand of task $i$, and $\lambda_l$ be the capacity of the $l$-th type resource. Let constant $C_1^{\text{up}}$ be a sufficiently large number and can be set to $\sum_{i \in V} t_i$. Let variable $s_i$ denote the start time of task $i$, variable $t_{max}$ denote the makespan. Let variable $u_{ij}$ equal 0 if and only if task $i$ starts before task $j$ starts, variable $w_{ij}$ equal 0 if and only if task $i$ starts after or at the same time as task $j$ finishes, variable $x_{ij}$ equal 1 if and only if task $j$ is running when task $i$ starts. The corresponding mixed integer linear programming (MILP) formulation is given by (2).

$$\min t_{max}$$

$$\text{s.t.} \ s_i + t_i \leq t_{max}, \forall i \in V,$$

$$s_i + t_i \leq s_j, \forall (i, j) \in E,$$

$$s_i < s_j + C_1^{\text{up}} u_{ij}, \forall i, j \in V,$$

$$s_j \leq s_i + C_1^{\text{up}}(1 - u_{ij}), \forall i, j \in V,$$

$$s_i < s_j + t_j + C_1^{\text{up}}(1 - w_{ij}), \forall i, j \in V,$$

$$s_j + t_j \leq s_i + C_1^{\text{up}} w_{ij}, \forall i, j \in V,$$

$$u_{ij} + w_{ij} \leq 1 + x_{ij}, \forall i, j \in V,$$

$$u_{ij} + w_{ij} \geq 2x_{ij}, \forall i, j \in V,$$

$$\rho_l^i + \sum_{j \neq i} x_{ij} \rho_l^j \leq \lambda_l, \forall i \in V, l \in \{1, ..., r\},$$

$$s_i \geq 0, t_{max} \geq 0, \ u_{ij}, w_{ij}, x_{ij} \in \{0, 1\}, \forall i, j \in V. \tag{2}$$

For the heterogeneous scheduling problem $(V, E, C, t, \rho, \lambda, K_{acc})$ with $r$ types of resources, where $C$ is the pool set and $K_{acc}$ is the compatibility coefficient matrix. Let variables $s_i, t_{max}, u_{ij}, w_{ij}, x_{ij}$ and constants $t_i, \rho_l^i$ share the same meaning as in the homogeneous case. Let variable $v_i^k$ equal 1 if and only if task $i$ is assigned to pool $k$, variable $y_{ijk}$ equal 1 if and only if task $j$ is running on pool $k$ when task $i$ starts. Let constant $C_0^{\text{up}}$ be a sufficiently large bound and can be set to $2 \max\{1/K_{acc}(i, k) : K_{acc}(i, k) \neq 0, \ \forall i, k\}$. Let constant $a_{ik} = \min\{1/K_{acc}(i, k), C_0^{\text{up}}\}$, constant $C_1^{\text{up}}$ be a sufficiently large number and can be set to $\sum_{i \in V, k \in C} t_i a_{ik} + \sum_{i \in V, l=1...,r} \rho_l^i$. Let constant $\lambda_l^k$ be the $l$-th type resource capacity of pool $k$. The MILP formulation of the heterogeneous scheduling problem is formulated by (3). The MILP formulation for heterogeneous scheduling involves substantially more intricate constraints and a greater number of variables and constraints

compared to the homogeneous case, significantly increasing the difficulty of solving the problem.

$$
\begin{aligned}
&\min t_{max}, \\
&\text{s.t. } s_i + t_i \sum_k v_i^k a_{ik} \leq t_{max}, \forall i \in V, \\
&\quad s_i + t_i \sum_{k \in C} v_i^k a_{ik} \leq s_j, \forall (i,j) \in E, \\
&\quad s_i < s_j + C_1^{\text{up}} u_{ij}, \forall i,j \in V, \\
&\quad s_j \leq s_i + C_1^{\text{up}}(1 - u_{ij}), \forall i,j \in V, \\
&\quad s_i < s_j + t_j \sum_{k \in C} v_j^k a_{jk} + C_1^{\text{up}}(1 - w_{ij}), \forall i,j \in V, \\
&\quad s_j + t_j \sum_{k \in C} v_j^k a_{jk} \leq s_i + C_1^{\text{up}} w_{ij}, \forall i,j \in V, \\
&\quad u_{ij} + w_{ij} \leq 1 + x_{ij}, \forall i,j \in V, \\
&\quad u_{ij} + w_{ij} \geq 2x_{ij}, \forall i,j \in V, \\
&\quad x_{ij} + v_j^k \geq 2y_{ijk}, \forall i,j \in V, k \in C, \\
&\quad x_{ij} + v_j^k \leq 1 + y_{ijk}, \forall i,j \in V, k \in C, \\
&\quad \sum_k v_i^k = 1, \forall i \in V, k \in C, \\
&\quad 1 - v_i^k + C_0^{\text{up}} - a_{ik} > 0, \ \forall i \in V, k \in C, \\
&\quad \rho_l^i + \sum_{j \neq i} y_{ijk} \rho_l^j \leq C_1^{\text{up}}(1 - v_i^k) + \lambda_l^k, \forall i \in V, k \in C, l \in \{1,...,r\}, \\
&\quad s_i \geq 0, t_{max} \geq 0, \ u_{ij}, w_{ij}, x_{ij}, y_{ijk}, v_i^k \in \{0,1\}, \forall i,j \in V, k \in C.
\end{aligned} \tag{3}
$$

For a given solution $z = \{s_i, t_{max}, u_{ij}, w_{ij}, x_{ij}, y_{ijk}, v_i^k\}_{i,j,k}$, we define

$$
\phi(z) = x, \text{ where } x(i) = (s(i), c(i)) = (s_i, \{k : v_i^k = 1\}).
$$

The original space $A$ is defined as the image of $\phi$ taking domain as the set of all feasible solutions of the MILP.

**Definition 1.** $A = \{\phi(z) : z \text{ is a feasible MILP solution}\}$.

Due to the definition of the variables, we immediately have that $\phi$ builds a one-to-one correspondence between the space of all feasible MILP solutions and the original space $A$.

**Proposition 1.** $\phi$ *is a bijection from the space of all feasible MILP solutions to* $A$.

A.2  DEFINITIONS AND PROPERTIES

Here, we provide formal definitions for the reduced action space and related concepts used in our heterogeneous DAG scheduling framework, as introduced in Section 4.2. We define the reduced space, feasible solutions, reliance, and deadlock, which support the analysis of list scheduling's optimality gap and WeCAN's single-pass skip action. Additionally, we reformulate Algorithm 1 and describe the projection map that achieves optimal schedules.

**Definition 2.** *The reduced space is defined as*

$$
B = \{\{d(v), c(v)\}_{v \in V}; c(v) \in C, \{d(v) : c(v) = c\} = \mathbb{Z}^+(|\{v : c(v) = c\}|), \forall c \in C\},
$$

*where* $d(v) \in \mathbb{Z}^+$ *is the execution order of task* $v \in V$ *on its resource pool* $c(v) \in C$. *The elements in reduced space* $B$ *are called schedule orders. We define the natural map* $T : A \to B$ *as* $T(\{s(v), c(v)\}_{v \in V}) = \{d(v), c(v)\}_{v \in V}$ *where* $d(v)$ *is the rank of task* $v$'s *start time* $s(v)$ *among tasks assigned to* $c(v)$. *The feasible reduced space, denoted by* $B_f$, *is defined as the image of* $A$ *under* $T$, *i.e.,* $B_f = T(A)$.

**Definition 3.** *A schedule order $\{(d(v), c(v))\}_{v \in V}$ is feasible, if $\{(d(v), c(v))\}_{v \in V} \in B_f$, i.e., there exists a feasible solution $x \in A$, such that $T(x) = \{(d(v), c(v))\}_{v \in V}$.*

To analyze properties of the feasible reduced space $B_f$, we introduce reliance and deadlock, which capture scheduling dependencies and infeasibility.

**Definition 4.** *In a scheduling problem $(G, V, C, \rho, t, \lambda, K_{acc})$, we say $w$ depends on $v$ if $(v, w) \in E$, indicating $v$ must complete before $w$ starts.*

**Definition 5.** *Given a schedule order $\{(d(v), c(v))\}_{v \in V} \in B$, we say that $w$ relies on $v$ if there exists a sequence of tasks $v_0 = v, v_1, ..., v_l = w$ such that, for any $i = 0, ..., l - 1$, at least one of the following holds: (1) $(v_i, v_{i+1}) \in E$; (2) $c(v_i) = c(v_{i+1})$ and $d(v_i) < d(v_{i+1})$.*

**Definition 6.** *A schedule order $\{(d(v), c(v))\}_{v \in V}$ has a deadlock, if there exists $v$ such that $v$ relies on itself.*

We next formulate Algorithm 2 to incorporate the projection map, as discussed in Section 4.2. The projection map $S_n$, defined in Section 4.2, maps schedule orders to feasible schedules by allowing waiting, enabling optimal solutions unattainable by list scheduling (see Figure 5).

**Definition 7.** *The projection map $S_n : B_f \rightarrow A$ is defined as the map specified by Algorithm 2. It maps feasible schedule orders in the reduced space $B_f$ to feasible solutions in $A$.*

We show that the projection map $S_n$ is well-defined in $B_f$ by the following proposition.

**Proposition 2.** *For any schedule order $\{(d(v), c(v))\}_{v \in V} \in B$ with no deadlock and meeting $\rho(v) \leq \lambda(c(v))$ and $K_{acc}(v, c(v)) > 0$ for all $v \in V$, Algorithm 2 generates a feasible schedule $\{(s(v), c(v))\}_{v \in V} \in A$ in at most $3n \cdot n_c$ loops, preserving the schedule order, i.e., $T(\{(s(v), c(v))\}_{v \in V}) = (d(v), c(v))\}_{v \in V}$.*

*Proof.* Denote $V = \{1, \ldots, n\}$ and $C = \{c_1, \ldots, c_{n_c}\}$. We prove that Algorithm 2 produces a feasible solution for the schedule order satisfying the conditions.

First, the algorithm ensures feasibility. The variable $t_{dep}(v)$ tracks the completion times of tasks $u$ such that $(u, v) \in E$, enforcing dependency constraints. Resource and compatibility checks, i.e., $\rho(v) \leq \lambda_{cur}(c(v))$ and $K_{acc}(v, c(v)) > 0$, ensure resource availability and task-pool compatibility. Thus, if Algorithm 2 terminates with $I_{\text{illegal}} = 0$, the output $\{(s(v), c(v))\}_{v \in V}$ is a feasible schedule in $A$.

Second, we show that Algorithm 2 does not exit with $I_{\text{illegal}} = 1$ for an schedule order satisfying the conditions. Suppose a schedule order $\{(d(v), c(v))\}_{v \in V} \in B$ has no deadlock and meets $\rho(v) \leq \lambda(c(v))$, $K_{acc}(v, c(v)) > 0$ for all $v$, yet the algorithm exits with $I_{\text{illegal}} = 1$. Since resource and compatibility conditions hold, an illegal exit occurs only when $n_{\text{count}} \geq n_c$, indicating a cycle through all pools. For each pool $c \in C$, let $u(c)$ be the task in $S$ with $c(u(c)) = c$ and minimal $d(u(c))$. When $n_{\text{count}} \geq n_c$, each $u(c)$ depends on some task $w \in S$, and thus relies on $w$. Since $w$ relies on $u(c(w))$, we have $u(c)$ relies on $u(c(w))$. With finite pools, this implies a cyclic reliance, e.g., $u(c_1) \rightarrow u(c_2) \rightarrow \cdots \rightarrow u(c_1)$, forming a deadlock. This contradicts the assumption of no deadlock, so Algorithm 2 cannot exit with $I_{\text{illegal}} = 1$.

Third, we bound the number of iterations. In each full iteration, one of three events occurs: (1) a task $v$ starts, (2) a task $v$ finishes (updating $t_{cur}$ via $t_f$), or (3) $t_{cur}(c(v))$ is updated to $t_{dep}(v)$. Each task starts and finishes at most once, contributing $n$ starts and $n$ finishes. Time updates to $t_{dep}(v)$ are bounded by the number of task, as $t_{dep}(v)$ reflects dependency resolutions. Since $n_{\text{count}} < n_c$ (from the contradiction), at least one full iteration per $n_c$ loops avoids the continue command, triggering an event. Thus, at most $3n$ events occur, and the algorithm terminates in at most $3n \cdot n_c$ loops.

Finally, the algorithm preserves the schedule order. Each task $v$ starts when it has the smallest $d(v)$ among unallocated tasks in pool $c(v)$. Start times $s(v)$ are assigned in $d(v)$-order within each pool, ensuring $T(\{(s(v), c(v))\}_{v \in V}) = (d(v), c(v))\}_{v \in V}$.

Thus, $S_n$ is well-defined, producing a feasible schedule in $A$ that preserves the schedule order. $\square$

Figure 4 illustrates how $S_n$ maps a feasible order to a feasible schedule, preserving its schedule order. This map not only connects the feasible reduced space and the original space, but also reveals the structure of the feasible reduced space by the following proposition.

---

**Algorithm 2** Projection Map $S_n$

---

**Input:** Scheduling problem $(V, E, C, \rho, t, \lambda, K_{acc})$ with $n$ tasks $V = \{1, \ldots, n\}$ and $n_c$ resource pools $C = \{c_1, \ldots, c_{n_c}\}$, and a schedule order $\{(d(v), c(v))\}_{v \in V} \in B_f$.
Initialize current pool $c = c_1$, cycle counter $n_{count} = 0$, legality flag $I_{illegal} = 0$, unallocated task set $S = V$, initial resources $\lambda_{cur}(c) = \lambda(c)$, earliest start times $t_{dep}(v) = 0$, and current pool times $t_{cur}(c) = 0$, for all $c \in C$, $v \in V$. Define next pool as $c_{next}(c_i) = c_{((i \mod n_c)+1)}$.
**while** $S \neq \emptyset$ **do**
  // Select task with smallest order on current pool
  Find task $v \in S$ such that $c(v) = c$ and $d(v)$ is minimized.
  **if** there exists task $u \in S$ such that $(u, v) \in E$ **then**
    // Check dependencies
    $n_{count} \leftarrow n_{count} + 1$
    **if** $n_{count} \geq n_c$ **then**
      // Detect potential deadlock
      $I_{illegal} \leftarrow 1$
      **break**
    **end if**
    $c \leftarrow c_{next}(c)$ // Move to next pool
    **continue**
  **end if**
  **if** not ($\rho(v) \leq \lambda(c(v))$ and $K_{acc}(v, c(v)) > 0$) **then**
    // Check resource and compatibility
    $I_{illegal} \leftarrow 1$
    **break**
  **end if**
  $n_{count} \leftarrow 0$ // Reset cycle counter
  **if** $\rho(v) \leq \lambda_{cur}(c(v))$ and $t_{cur}(c(v)) \geq t_{dep}(v)$ **then**
    // Start task if resources and time allow
    Start task $v$, update $S \leftarrow S \setminus \{v\}$.
    Set start time $s(v) = t_{cur}(c(v))$.
    Update resources: $\lambda_{cur}(c(v)) \leftarrow \lambda_{cur}(c(v)) - \rho(v)$.
    Update earliest start times of all successors of task $v$: $t_{dep}(w) \leftarrow \max\{t_{dep}(w), t_{cur}(c(v)) + t(v)/K_{acc}(v, c(v))\}$ for all $w$ with $(v, w) \in E$.
  **else**
    // Wait for resources or dependencies
    Compute next event time: $t_f = \min_w\{t_{end}(w) : t_{end}(w) \geq t_{cur}(c(v)), c(w) = c(v), t_{end}(w) = s(w) + t(w)/K_{acc}(w, c(w)), w \in V \setminus S\}$, or $\infty$ if no such $w$ exists.
    Update pool time: $t_{cur}(c(v)) \leftarrow \min(t_{dep}(v), t_f)$.
    Update resources: $\lambda_{cur}(c(v)) \leftarrow \lambda_{cur}(c(v)) + \sum_{w \in S_{up}} \rho(w)$, where $S_{up} = \{w \in V \setminus S : c(w) = c(v), t_{end}(w) = t_{cur}(c(v))\}$.
  **end if**
**end while**

---

**Proposition 3.** *A schedule order $\{(d(v), c(v))\}_{v \in V}$ is feasible, if and only if it has no deadlock and satisfies that $\rho(v) \leq \lambda(c(v))$ and $K_{acc}(v, c(v)) > 0$ hold for each $v \in V$.*

*Proof.* The necessity is obvious. The sufficiency arises from Proposition 2. Suppose $\{(d(v), c(v))\}_{v \in V} \in B$ has no deadlock and satisfies $\rho(v) \leq \lambda(c(v))$, $K_{acc}(v, c(v)) > 0$ for any $v \in V$. By Proposition 2, Algorithm 2 generates a feasible schedule $x = S_n(\{(d(v), c(v))\}_{v \in V}) \in A$ such that $T(x) = \{(d(v), c(v))\}_{v \in V}$. Thus, $\{(d(v), c(v))\}_{v \in V} \in B_f = T(A)$. This finishes the proof. $\qquad\square$

The above proposition characterizes $B_f$ as the set of schedule orders without deadlocks and with valid resource and compatibility constraints. It gives a judgment of whether a schedule order $x \in B$ locates in $B_f$. Now recall that $S_n$ defined above already meets $TS_n = I$, which is the first assumption in Assumption 1. Now we prove that the second assumption in Assumption 1 also holds for map $S_n$.

**Proposition 4.** $f(S_n T(x)) \leq f(x)$ *holds for any feasible schedule $x \in A$.*

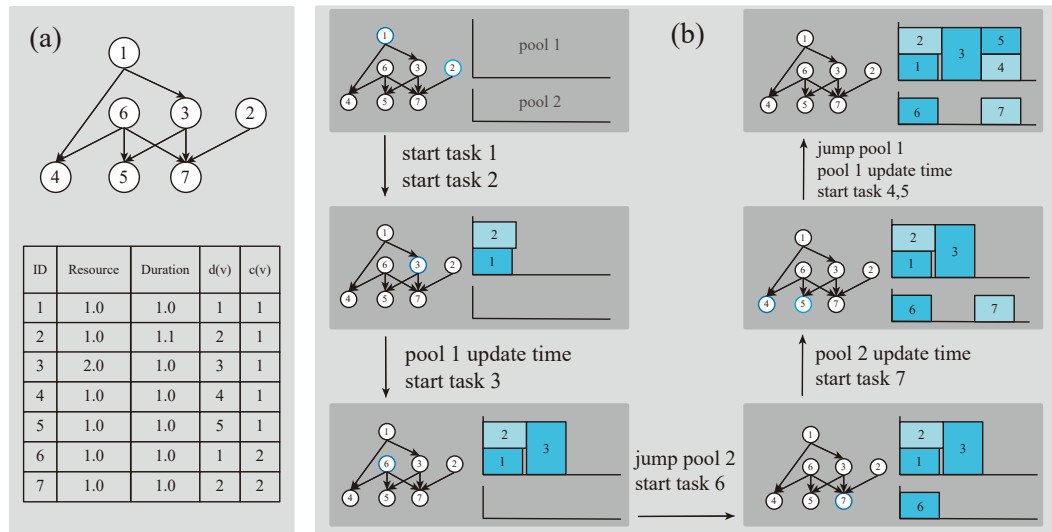

Figure 4: Example illustrating the generation process of $S_n$ for a heterogeneous DAG scheduling problem. (a) Problem instance with tasks, resource pools, and dependency constraints. Two pools provide a single resource type, with capacities 2 and 1. (b) The process of $S_n$, cycling through pools to allocate tasks in order, switching pools when dependencies require, and enabling waiting to preserve the schedule order, as proven in Proposition 2.

*Proof.* It suffices to show that for any $x \in A$ and $y = T(x)$, it holds that $f(S_n(y)) \leq f(x)$. Suppose there exists $y = T(x)$ such that $f(S_n(y)) > f(x)$. Let $y = \{(d(v), c(v))\}_{v \in V} \in A$, $S_n(y) = \{(s(v), c(v))\}_{v \in V} \in A$, and $x = \{(s_x(v), c_x(v))\}_{v \in V} \in A$. Since $y = T(x)$, we immediately have $c_x(v) = c(v)$ for all $v \in V$. The makespan condition $f(S_n(y)) > f(x)$ implies:

$$\max_{v \in V}\{s(v) + t(v)/K_{acc}(v, c(v))\} > \max_{v \in V}\{s_x(v) + t(v)/K_{acc}(v, c(v))\}.$$

Thus, there exists $v \in V$ such that $s(v) + t(v)/K_{acc}(v, c(v)) > s_x(v) + t(v)/K_{acc}(v, c(v))$, i.e., $s(v) > s_x(v)$. Thus we can define the non-empty set $W_l = \{v \in V : s(v) > s_x(v)\}$.

Select $v_0 \in W_l$ with the smallest $s(v_0)$ and smallest $d(v_0)$ among those with same $s(v_0)$, and let $c_0 = c(v_0)$, $k = d(v_0)$. Let $v_1, \ldots, v_{k-1}$ be tasks with $d(v_i) = i$ and $c(v_i) = c_0$. Since $v_0$ has the smallest $s(v)$ in $W_l$, for any $i = 1, ..., k-1$ it holds that $s_x(v_i) \geq s(v_i)$. Therefore, we have $s(v_0) > s_x(v_0) \geq s_x(v_{k-1}) \geq s(v_{k-1})$. By the fact $s(v_0) > s(v_{k-1})$ and $\{(s(v), c(v))\}_{v \in V}$ is generated by Algorithm 2, $s(v_0)$ is the earliest time $t \geq t_{dep}(v_0)$ such that $\rho(v_0) \leq \lambda_{cur}(c_0, t)$.

However, since $s_x(v_0) < s(v_0)$ and $s_x(v_0) \geq t^x_{dep}(v_0) \geq t_{dep}(v_0)$ (as $x$ is feasible), consider the resource availability in $x$. For $i = 1, \ldots, k-1$, $v_i \notin W_l$ and then $s_x(v_i) \geq s(v_i)$, implying that task $v_i$ in $x$ starts no earlier than in $S_n(y)$. Define $\lambda^x_{cur}(c_0, t)$ as the available resource vector in $x$ at time $t$ before $v_0$ starts. Since $x$ schedules tasks with the same resources, $\rho(v_0) \leq \lambda^x_{cur}(c_0, s_x(v_0)) \leq \lambda_{cur}(c_0, s_x(v_0))$, as fewer or equal resources are used in $x$ up to $s_x(v_0)$. Thus, $s_x(v_0)$ satisfies $t \geq t_{dep}(v_0)$ and $\rho(v_0) \leq \lambda_{cur}(c_0, s_x(v_0))$, contradicting the minimality of $s(v_0)$ in Algorithm 2.

Therefore, no such $y = T(x)$ exists with $f(S_n(y)) > f(x)$. Therefore, $f(S_n T(x)) \leq f(x)$ for all $x \in A$, which finishes the proof. $\qquad\square$

We proved that the projection map $S_n$ satisfies both conditions of Assumption 1. Thus, $S_n$ acts as a projection map, as $S_n T$ maps any feasible schedule $x \in A$ to a locally optimal schedule in $A$ among all schedules sharing the same schedule order of $x$. There exist optimal schedules residing in $S_n(B_f) \subset A$, where $S_n(B_f)$ is equivalent to $B_f$ via the bijections $T : S_n(B_f) \to B_f$ and $S_n : B_f \to S_n(B_f)$. Consequently, the feasible reduced space $B_f$ serves as a criterion for checking whether a existing approach, with generation map $S$ from action space to feasible schedule, always includes optimal schedules by verifying whether $TS$ is a surjection to $B_f$.

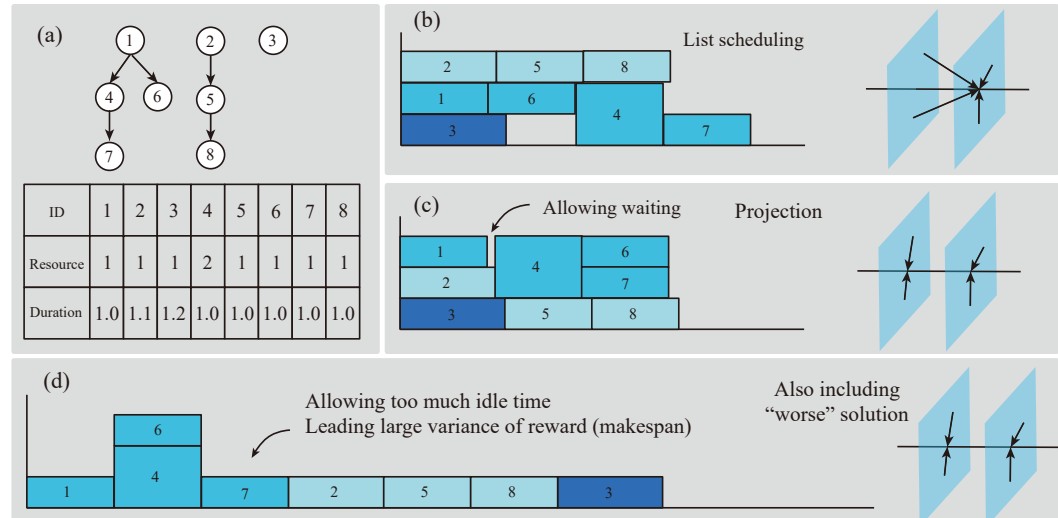

Figure 5: A counterexample showing list scheduling excludes the optimal solution for a DAG scheduling problem. (a) Problem instance with only one pool offering one type of resource with capacity 3. (b) The best (only) solution from list scheduling. (c) Optimal solution which can be generated via projection map $S_n$ by allowing waiting. (d) Worse solution by $S_n$, showing high makespan variance from unrestricted waiting.

We now define the action sequence and the score space from the neural network and explain how list scheduling connects these spaces to the original space $A$.

**Definition 8.** *Let $n = |V|$ and $n_c = |C|$. An action sequence $\{(v_i, c_i)\}_{i=1}^n$, where $v_i \in V$ and $c_i \in C$, comprises distinct tasks, i.e., $v_i \neq v_j$ for all $i \neq j$. Each action sequence corresponds to a point in the reduced space $B$ as follows: for each $v \in V$, there exists $i$ such that $v_i = v$; set $c(v) = c_i$, and let $d(v)$ be the count of $\{j \leq i : c_j = c(v)\}$. An extended action sequence is obtained by inserting several skip actions into an action sequence.*

**Definition 9.** *Let $n = |V|$ and $n_c = |C|$. The score space $B_{score}$ comprises all functions $d : V \times C \to \mathbb{R}$, representing neural network scores for ranking task-pool pairs. The extended score space $B_{ex-score}$ comprises all functions $d : V \times C \cup \{u_a, u_b, u_c\} \to \mathbb{R}$.*

Each action sequence corresponds to a point in the reduced space $B$, whereas neural network scores correspond to a point in the score space $B_{score}$ or $B_{ex-score}$. To translate action sequences or scores into feasible schedules, a mapping from $B$, $B_{score}$ or $B_{ex-score}$ to the original space $A$ is required. List scheduling, as described in Section 2.2 of the main text, can be extended to these spaces. It repeatedly selects choosing unmasked actions until all actions are masked and advance to the next task completion time. The extension differs in the rule for selecting unmasked actions, which varies by domain. For a point $\{(d(v), c(v))\}_{v \in V} \in B$ in the reduced space, list scheduling selects the action $(v, c)$ with the smallest $d(v)$ among those with $c(v) = c$, breaking ties by choosing the smallest pool index $c$. For a point $d$ in $B_{score}$ or $B_{ex-score}$, list scheduling selects the action $(v, c)$ with the smallest $d(v, c)$ (for $B_{ex-score}$, also calculate $d(\pi_{skip}) = u_a(1 - \frac{k}{2n})^{u_b} + u_c$ dynamical). This provides a flexible method for generating feasible schedules. However, we will show that list scheduling map $S_{list}$ from $B$ or $B_{score}$ cannot always include optimal schedules, as $TS_{list}$ is not surjective to $B_f$, risking suboptimal and ineffective schedules.

### A.3 A COUNTEREXAMPLE REVEALING NON-OPTIMUM PROPERTIES OF LIST SCHEDULING

We present a counterexample showing that the image of the list scheduling mapping $S_{list}$ may exclude the optimal feasible schedule. In panel (a) of Figure 5, three DAGs with eight tasks are scheduled on one resource pool with capacity $\lambda(c) = 3$. Panel (b) shows the best feasible schedule produced by list scheduling, which is also the only schedule in $S_{list}$'s image. Panel (c) shows the optimal

feasible schedule. In the list scheduling schedule, task 4 is critical to the makespan. At time 1.0, list scheduling selects task 5 or 6 due to masking constraints, resulting in a longer makespan. However, if the scheduler waits for task 4 at time 1.0, as enabled by WeCAN's skip action, the optimal schedule is achieved, reducing the makespan by about 25%, promoting effective resource allocation.

In this problem, list scheduling cannot produce the optimal feasible schedule because $TS_{list}$'s image does not include the optimal schedule order. As shown in panel (c) of Figure 6, the optimal feasible schedule $x \in A$ corresponds to a schedule order $T(x) \in B_f$, but $S_{list}$ maps $T(x)$ to a suboptimal schedule $S_{list}T(x) \in A$ with a different order, i.e., $T(x) \neq TS_{list}T(x)$. This occurs because list scheduling requires immediate resource availability and prohibits skip actions, prioritizing tasks with lower resource demands and delaying resource-intensive tasks. To include optimal schedules, a map $S$ should ensure $TS$ is surjective to $B_f$. Thus, $S$ should allow skip to schedule tasks until their resource demands are met, as in $S_n$ (panel (b) of Figure 6) or by including skip actions in the decision space, as in Algorithm 1 (panel (d) of Figure 6).

Panel (d) of Figure 5 illustrates the limitations of using $S_n$ for training a scheduling policy. To include all feasible schedule orders, $S_n$ may introduce long idle time in worst-case schedules due to waiting for reliance. Although $S_n$ includes optimal schedules, it also produces inferior solutions scattered across the reduced space, resulting in a higher average makespan and increased variance. Lower mean and higher variance in the reward function (negative makespan) hinder the convergence of reinforcement learning by providing a poor starting point and increased sample variance. Compared to using $S_n$ directly, we expand the decision space to include all feasible schedule orders by incorporating skip actions in the single-pass design (Algorithm 1). This design cluster most poor solutions along the axis $u_a$ and $u_c$, only moderately increasing variance while closing the optimality gap.

### A.4   PROOFS OF THEOREMS

Here we provide the proofs of the theorems in the main text.

**Theorem 1.** *i) Algorithm 1 generates a feasible solution within $2n$ steps where $n$ is the number of nodes. ii) Algorithm 1 assigns positive probabilities to at least one optimal solution and all feasible orders. iii) Without the skip action, statement ii) does not hold for some problem $X$. iv) For each problem $X$, there exist scores $\{u_{(v,c)}\}_{v \in V, c \in C}$ and $u_a, u_b, u_c$ enabling an optimal solution by greedily selecting the action with the highest $p_\theta(\pi)$ in Algorithm 1.*

*Proof.* **i)** The dependency mask ensures that tasks start only after their dependencies are completed, satisfying dependency constraints. Actions satisfying the resource mask meet resource requirements. Since both constraints are satisfied, the generated feasible schedule is valid. Each action corresponds to a task's start or finish, and with $n = |V|$ tasks, the number of actions is at most $2n$.

**ii)** To prove that any feasible schedule order $\{(d(v), c(v))\}_{v \in V} \in B_f$ has a positive probability under Algorithm 1, denote $V = \{1, \ldots, n\}$. First, we show that for any partial action sequence $\pi_{1:t}$ preserving the schedule order, there exists an action $\pi_{t+1}$ such that $p(\pi_{t+1}|\pi_{1:t}) > 0$ and $\pi_{1:t+1}$ preserves the schedule order.

If there exist tasks running on some pools, let $\pi_{t+1}$ be the skip action. Since unfinished tasks exist, the skip action is unmasked, with $p(\pi_{t+1}|\pi_{1:t}) > 0$. The skip action does not alter the schedule order, so $\pi_{1:t+1}$ preserves the schedule order.

If no tasks are running, all pools have full resource capacity, i.e., $\lambda_{cur}(c) = \lambda(c)$. Since $\{(d(v), c(v))\}_{v \in V} \in B_f$, Proposition 3 ensures $K_{acc}(v, c(v)) > 0$ and $\rho(v) \leq \lambda(c(v)) = \lambda_{cur}(c(v))$ for all $v$. Thus, pool $c(v)$ can schedule task $v$. Suppose all actions preserving the order are masked by the dependency mask. For each pool $c_0 \in C$, define $u(c_0)$ as the unstarted task with the smallest $d(v)$ such that $c(v) = c_0$. If all such tasks are masked, each $u(c_0)$ depends on an unstarted task $w(u(c_0))$, which relies on or equals $u(c(w(u(c_0))))$. With $|C| = n_c$ finite pools, this forms a cyclic reliance, implying a deadlock. By Proposition 3, this contradicts the feasibility of $\{(d(v), c(v))\}_{v \in V}$. Thus, an unmasked action preserving the order exists, with $p(\pi_{t+1}|\pi_{1:t}) > 0$.

Since the action sequence terminates within $2n$ steps, there exists a feasible schedule $x \in A$ with positive probability such that $T(x) = \{(d(v), c(v))\}_{v \in V}$.

Next, let $x_0 \in A$ be an optimal feasible schedule. By Proposition 4, $f(S_nT(x_0)) \leq f(x_0)$, so $x = S_nT(x_0) = \{(s(v), c(v))\}_{v \in V} \in A$ is also optimal. In Algorithm 2, each task $v$ starts at $t_{cur}(c(v))$, initialized to 0 and updated to $\min(t_{dep}(w), t_f)$, where $t_{dep}(w)$ and $t_f$ are both task completion times. Thus, in $x$, each task starts at 0 or a task's completion time. To show Algorithm 1 assigns positive probability to $x$, it suffices to prove that for any partial sequence $\pi_{1:t}$ matching $x$, there exists $\pi_{t+1}$ such that $p(\pi_{t+1}|\pi_{1:t}) > 0$ and $\pi_{1:t+1}$ matches $x$.

Define $t_{cur}$ as the current time after $\pi_{1:t}$, and let $U \subset V$ be the set of unstarted tasks. Let $v \in U$ have the smallest $s(v)$ in $x$. Since $\pi_{1:t}$ matches $x$, we have $s(v) \geq t_{\text{cur}}$.

If $s(v) = t_{cur}$, since $x$ is feasible, task $v$ on pool $c(v)$ satisfies dependency ($t_{cur} = s(v) \geq t_{dep}(v)$) and resource constraints ($\rho(v) \leq \lambda_{cur}(c(v))$, $K_{acc}(v, c(v)) > 0$). Thus, action $(v, c(v))$ is unmasked, with $p((v, c(v))|\pi_{1:t}) > 0$, and $\pi_{1:t+1}$ matches $x$.

If $s(v) > t_{cur}$, since $x = S_nT(x_0)$, either $\rho(v) > \lambda_{cur}(c(v), t_{cur})$ or $t_{cur} < t_{dep}(v)$. By Proposition 3, $\rho(v) \leq \lambda(c(v))$. Thus, $\lambda_{cur}(c(v), t_{cur}) < \lambda(c(v))$ (tasks are running on $c(v)$) or $t_{cur} < t_{dep}(v)$ (dependencies are running). Hence, there exist running tasks, so the skip action is unmasked, with $p(\pi_{\text{skip}}|\pi_{1:t}) > 0$. Since $s(v)$ is a task completion time, the skip action advances $t_{cur}$ to $t_f \leq s(v)$, and $\pi_{1:t+1}$ matches $x$.

In both cases, an unmasked action satisfies the requirement. Thus, Algorithm 1 assigns positive probability to an optimal schedule $x$.

iii) The counter-example in Appendix A.3 (Figure 5, problem $X$) shows that statement ii) fails without skip actions. Without skip action, Algorithm 1 cannot produce the optimal schedule order.

iv) Statement ii) shows that an optimal feasible schedule $x \in A$ is generated by an extended action sequence $\{\pi_t\}_{t=1}^{T(x)}$. For any coefficients $u_a > 0$, $u_b > 0$, and $u_c \in \mathbb{R}$, the score for skip action $\pi_{\text{skip}}$ at step $k$ as $u_{\pi_{\text{skip}}} = u_a \left(1 - \frac{k}{2n}\right)^{u_b} + u_c$, is a decreasing function. For non-skip action $\pi_t = (v_t, c_t)$ in the sequence, assign $u_{\pi_t} = \frac{1}{2}\left(\left(1 - \frac{t-1}{2n}\right)^{u_b} + \left(1 - \frac{t}{2n}\right)^{u_b}\right)$. For other non-skip actions $\pi = (v, c)$ not in the sequence, assign $u_\pi = u_c - 1$. With these scores, at each step $t$, the action $\pi_t$ is unmasked and has the highest score in Algorithm 1. Thus, greedily selecting the action with the highest $p_\theta(\pi)$ generates $x$.

$\square$

Theorem 1 states that the skip action in Algorithm 1 addresses list scheduling's optimality gaps by enabling optimal feasible schedules while not sacrificing computational efficiency. Various approaches generate feasible schedules by first producing a point in a discrete decision space, such as the reduced space $B$, score space $B_{score}$, or another space, and then applying a map $S$ to obtain a feasible schedule in the schedule space $A$. To include optimal schedules, the map's image should cover sufficient schedule orders, with $TS$ being surjective to $B_f$ ensuring that composing $S$ with $S_nT$ as $S_nTS$ includes optimal schedules. As shown in Figure 6, the projection map $S_n$ (panel (b)) satisfies this condition, including optimal schedules but introducing high variance in makespan. The list scheduling map $S_{list}$ (panel (c)) violates this condition, excluding optimal schedules but maintaining low variance. Our Algorithm 1's map $S$ (panel (d)) satisfies the condition, including optimal schedules while maintaining low variance by clustering poor solutions, supporting WeCAN's effective and optimal scheduling.

**Theorem 2.** *Let $S : B \to A$ be a generation map satisfying Assumption 1. For any optimal solution $x$, there exists an optimal solution $y \in Image(S)$ and $T(y) = T(x)$.*

*Proof.* Let $y = S(T(x))$. Obviously $y \in Image(S)$. And $T(y) = T(S(T(x)) = TS(T(x)) = I(T(x)) = T(x)$. Since $f(y) = f(ST(x)) \leq f(x)$, $y$ is also an optimal solution. This finishes the proof. $\square$

In Theorem 1, we proved that the extended action space, together with the image of the extended maps, contains both an optimal solution and all feasible orders. The extension is obtained by enlarging the reduced decision space $B_f$, which originally corresponds to quotient space of action sequences without skip, to a richer space that also includes skip actions. By merging equivalent classes in this

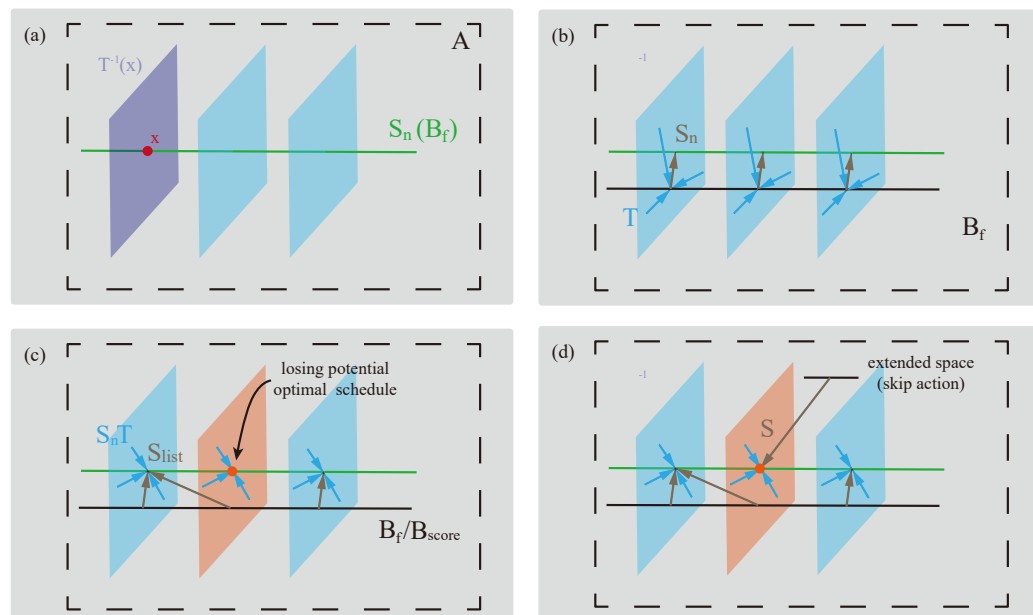

Figure 6: Generation approaches produce a point in a decision space (e.g., feasible reduced space $B_f$ or score space $B_{score}$) and apply a mapping $S$ to obtain a feasible schedule in the schedule space $A$. The surjectivity of $TS$ to $B_f$ serves as a criterion for including optimal schedules. (a) Mappings $S_n$ and $T$ embed $B_f$ as a subspace $S_n(B_f) \subset A$ containing optimal schedules, with $T^{-1}$ partitioning $A$ into pieces, where $S_nT(x)$ is the optimal schedule in each piece. (b) The projection map $S_n$ links $B_f$ to $S_n(B_f)$, satisfying $TS_n = I$ and including optimal schedules. (c) List scheduling $S_{list}$ fails to ensure $TS_{list}$ is surjective, excluding optimal schedules for some problems. (d) Algorithm 1's map $S$ incorporates skip actions, ensuring $TS$ is surjective and including optimal schedules while remaining low variance of makespan.

extended action space, that is, actions that yield the same feasible schedule, we obtain a quotient space that represents all distinct feasible schedules under the new action set.

In this construction, the generation map $S_{list}$ is extended to a new map $S$ that remains valid over the enlarged domain, while the transformation map $T$ is adjusted to map schedules consistently into the extended quotient space. Following the proofs of Theorem 1 and Proposition 4, we observe that the resulting triplet $(B_f, T, S)$ now satisfies Assumption 1. Importantly, this extension preserves optimal solutions while also expanding the feasible action space.

Moreover, the extended space has a desirable structural property: poor solutions, which arise from excessive skipping, are not scattered arbitrarily but become concentrated in the high-$u_a$, high-$u_c$ region of the space. This clustering means that suboptimal behaviors occupy identifiable and avoidable regions, while the rest of the space remains tractable for exploration. Consequently, our design of single-pass skip actions provide not only an enlarged feasible domain but also a natural projection framework that retains optimality, improves the structure of the solution space, and facilitates more stable training dynamics.

## A.5 ROLE OF LOCAL SEARCH

Although the projection map $S_n$ cannot be directly used for learning due to its high variance in makespan (Figure 6), it enables an alternative two-step approach: sampling and refinement, with the latter called local search. In the first step, a sampler generates suboptimal feasible schedules, and in the second step, local search identifies potential optimal schedules in their neighborhoods. This approach mitigates optimality gaps in maps like $S_{list}$ by leveraging projection map $S_n$ in local search, aligning with WeCAN's goal of effect and optimal scheduling.

To illustrate this process, we propose a local search strategy using the projection map $S_n$. For a feasible schedule $x \in A$ from the sampler, the natural map $T$ yields its schedule order $T(x) \in B_f$. The neighborhood $N(T(x)) \subset B_f$ comprises feasible schedule orders obtained by inserting a task into another position in a pool. The local search algorithm selects the optimal schedule in $S_n(N(T(x)))$, completing one search step. Theorem 3 guarantees the connectivity of $B_f$, ensuring any feasible schedule can reach an optimal schedule through finite task insertions.

**Theorem 3.** *For any feasible schedule $x \in A$, there exists an optimal schedule $x_0 \in A$ reachable by applying finite steps of task insertions in the feasible reduced space $B_f$. That is, there exist feasible schedules $x_1, \ldots, x_m \in A$, with $x_0 = x$, such that $T(x_i) \in N(T(x_{i+1}))$ for $i = 0, \ldots, m-1$.*

*Proof.* We prove this by induction on the number of tasks $n = |V|$. Note that this is equivalent to that any two feasible schedules are connected by finite steps of task insertions. For the base case, when $n = 1$, a single task has $k < n_c = |C|$ feasible schedule orders in $B_f$, each assignable to a pool $c \in C$ with $K_{acc}(v, c) > 0$ and $\rho(v) \leq \lambda(c)$. Any schedule order can be transformed into another by one insertion (reassigning the task to another pool), so the result holds.

For the inductive step, assume the result holds for $n-1$ tasks. Let $T(x) = \{(d(v), c(v))\}_{v \in V} \in B_f$ and $T(x_0) = \{(d_0(v), c_0(v))\}_{v \in V} \in B_f$ be the schedule orders of a feasible schedule $x \in A$ and an optimal schedule $x_0 \in A$, respectively. Since the DAG is acyclic, there exists a task $v_0 \in V$ with no dependent tasks (i.e., no $v \to v_0$ edges). Define $y_1 = \{(d_1(v), c_1(v))\}_{v \in V} \in B_f$ by inserting $v_0$ at the end of pool $c(v_0)$ in $T(x)$, and $y_2 = \{(d_2(v), c_2(v))\}_{v \in V} \in B_f$ by inserting $v_0$ at the end of pool $c_0(v_0)$ in $T(x_0)$. Since no tasks depend on $v_0$, inserting $v_0$ does not introduce deadlocks. By Proposition 3, since $T(x) \in B_f$, we have $K_{acc}(v_0, c(v_0)) > 0$ and $\rho(v_0) \leq \lambda(c(v_0))$, ensuring $y_1 \in B_f$. Similarly, $y_2 \in B_f$.

Thus, $T(x) \in N(y_1)$ and $y_2 \in N(T(x_0))$. Define $z_1 = \{(d_1(v), c_1(v))\}_{v \neq v_0}$ and $z_2 = \{(d_2(v), c_2(v))\}_{v \neq v_0}$, representing schedule orders in $B_f^{(n-1)}$, the feasible reduced space for the DAG scheduling problem excluding $v_0$. By Proposition 3, $z_1, z_2 \in B_f^{(n-1)}$. By the inductive hypothesis, there exist $z_3, \ldots, z_m \in B_f^{(n-1)}$, with $z_1 = z_m$, such that $z_i \in N(z_{i+1})$ for $i = 2, \ldots, m-1$. For each $z_i$, construct $y_i = \{(d_i(v), c_i(v))\}_{v \in V}$ by adding $v_0$ at the end of pool $c(v_0)$ to $z_i$. Since $z_i \in B_f^{(n-1)}$ has no deadlocks and no tasks depend on $v_0$, $y_i \in B_f$ by Proposition 3. By the definition of the neighborhood, $z_i \in N(z_{i+1})$ implies $y_i \in N(y_{i+1})$.

Thus, the sequence $y_1, y_2, \ldots, y_m$ connects $T(x)$ to $T(x_0)$, with $y_1 \in N(T(x))$, $y_i \in N(y_{i+1})$, and $T(x_0) \in N(y_2)$. Applying $S_n$ yields feasible schedules $x_i = S_n(y_i) \in A$, satisfying $T(x_i) = y_i$, and the sequence $x_0, x_2, \ldots, x_m = x_1, x$ reaches $x_0$, proving the connectivity of $B_f$ and finishing the proof.

$\square$

This local search approach with projection iteratively improves schedules by exploring the original space piece by piece, with Theorem 3 guaranteeing its feasibility in approaching an optimal solution. Figure 7 illustrates how local search with projection map $S_n$ also fixes the inherent optimality gap in list scheduling (as illustrated in Panel (c) of Figure 6). Panel (a) shows that the optimum is excluded by the list scheduling map $S_{list}$ through mapping it to a schedule with differing order. Panel (b) shows that although $S_{list}$ excludes the optimal solution, applying the local search process $LS$ on $T(S_{list}(y))$ and then performing the projection map $S_n$ also helps to get the optimal solution $S_n(LS(T(S_{list}(y))))$. This provides an alternative way for overcoming the optimality gaps. Experimental results validating the effectiveness of local search are shown in Appendix F.4.

## B   BENEFIT FROM ONCE NETWORK PROCESSING (NON-AUTO-REGRESSIVE) MODEL

Compared to the auto-regressive models. We use the once network processing (non-auto-regressive) model because it takes the following two benefits and does not bring noticeable performance loss.

- **Computation-speed gains from non-auto-regressive decoding:** Our single-pass architecture rapidly generate solutions with just one GNN forward, enabling rapid schedule

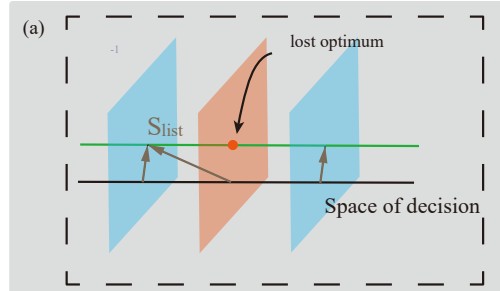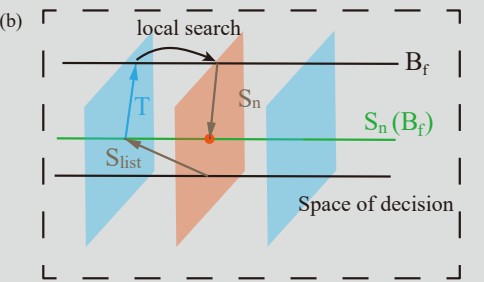

Figure 7: Local search with projection. (a) The optimal schedule is excluded by the list scheduling map $S_{list}$ due to a different schedule order. (b) Local search $LS$ on the schedule order $T(S_{list}(x))$, followed by the projection map $S_n$, recovers the optimal schedule $S_n(LS(T(S_{list}(x))))$.

generation even on CPU-only servers. As shown in Appendix J and Table 20, over 90 % of inference time is spent in the generation-map step (which any scheduler must perform), so our total runtime approaches the theoretical lower bound. By contrast, an auto-regressive decoder would rerun the GNN for each of the $n$ tasks, on TPC-H-30 this inflates total cost by $\approx 10\times$ on GPU, and on TPC-H-100 by $\approx 100\times$ on CPU, making it orders of magnitude slower.

- **Reduced training GPU memory with non-autoregressive decoding:** During training, backpropagation must keep all forward-pass activations in memory which dominates the GPU memory costs of DAG scheduling. In evaluation process, activations can be immediately freed. But in trainning process, they must be kept after the solution generated and reward calculated. An auto-regressive decoder runs the full network at each of the $n$ scheduling steps, retaining $n$ sets of activations, so you quickly run out of GPU memory and must use tiny batch sizes (e.g. on TPC-H-30/50) and cannot scale to larger DAG scheduling problem. In contrast, our single-pass design invokes the network exactly once per problem, storing only one set of activations. This lets us train on much larger problems (up to $\approx 1000$ tasks) with practical batch sizes.

Unlike combinatorial problems such as TSP where each decision reshapes the remaining solution, in DAG scheduling each choice mainly affects which tasks are running (and which masks are active). Once a task finishes, it no longer influences later decisions beyond updating the dependency mask. Because any selectable task already has all its parents completed, available tasks are independent except for shared resource contention. Intuitively, this matches real-world practice: you plan your next jobs based on currently running workloads, not on jobs that have already finished. Moreover, the following two experiments results shows that non-auto-regressive model does not lead to noticeable performance loss.

**Evidence from local search:** If early decisions had a strong ripple effect, perturbing the action order would yield substantial improvements. We implemented a local search that reinserts any action at any position and keeps the best reorder. As shown in Appendix F.3, 100 local search steps improve makespan by only 2% on TPC-H-30 and 1% on TPC-H-50, demonstrating that early scheduling choices exert only limited long-range influence on final performance.

**Minimal gains from auto-regressive models in DAG scheduling:** Before adopting the non-auto-regressive (NAR) model, we start our works by implementing an auto-regressive (AR) version that encodes both current and future resource availability (i.e., running tasks) for each pool. Since pools may have variable numbers of running tasks but embeddings must be fixed-size, we represent each pool's state with quantile information. Let $K$ be a integer-value hyper-parameters. For pool $c$, $t_{cur}$ denotes the current time, $r_{cur}$ the current resource capacity vector, and $r_f$ the full capacity. At each step, pool $c$ is embedded into $(r+1) \cdot (K+1)$-dimensional dynamical feature vector $(t_1, r_1, ..., t_{K+1}, r_{K+1})$, where $r$ is the number of resource types. Here, $t_i$ is the earliest time the

resource type 1 will be released to reach $r_{cur}^{(1)} + \frac{i-1}{K}(r_f^{(1)} - r_{cur}^{(1)})$ if no more task is coming, and $r_i$ is the corresponding resource vector at $t_i$. Notably, we have $t_1 = t_{cur}$, $r_1 = r_{cur}$, $t_{K+1}$ is the maximal finish time for tasks running on pool $c$, and $r_{K+1} = r_{ful}$, so this embedding captures the current resource state, the full resource capacity, and critical resource states over time.

The AR model is identical to the NAR model except at the decoder: at each step, the decoder extracts these dynamic features for every pool, processes them with an MLP for pool embeddings, then applies the WCA layer to update both task embeddings and pool–task scores. By taking $K = 5$, we evaluated the AR model on the TPC-H dataset, taking the maximal batch size (20 for TPC-H-30 and 8 for TPC-H-50) not out of GPU-memory.

| | TPC-H-30 | | TPC-H-30 | | TPC-H-100 | |
|---|---|---|---|---|---|---|
| | time | time | time | time | time | time |
| WeCAN-Greedy(NAR) | 19578 | 0.15 | 33428 | 0.50 | 62587 | 1.72 |
| WeCAN-S(256)(NAR) | 18964±10 | 2.43 | 32814±47 | 2.86 | 61373±28 | 10.43 |
| WeCAN-AR-Greedy | 19386 | 1.96 | 33786 | 3.39 | (out of GPU-memory) | * |
| WeCAN-AR-S(256) | 18910±26 | 3.57 | 33201±75 | 7.05 | (out of GPU-memory) | * |

Table 4: Performance comparison on TPC-H workloads.

From the results, the AR model achieves only less than 1% better makespan on TPC-H-30 and even 1% worse performance on TPC-H-50 (this may due to the small batch size due to larger GPU memory costs), but requires substantially more inference time (more than 10 times of NAR) and training GPU memory. This trade-off is the main reason we adopted the non-auto-regressive model, single-pass design.

## C  CONTRIBUTIONS ON SKIP ACTIONS OVER PREVIOUS WORKS

Although skip actions have been taken in a few previous works (Mao et al., 2016; Grinsztajn et al., 2021; Zhadan et al., 2023), their architectural designs require either a fixed task type or a fixed number of resource pools, which limits their adaptability to diverse environments (e.g., when a new task type or resource pool is added, the trained network can no longer be applied). Our approach preserves this adaptability. Furthermore, these prior methods are built on actor–critic architectures and require $n$-round network processing to solve a problem instance with $n$ tasks, leading to significantly longer inference times. Their designs generate a dynamic skip score at each round of processing, which makes them unsuitable for the single-pass setting: computing each dynamic skip score with a forward pass greatly increases inference cost, while relying on a static score cannot close the optimality gap. Consequently, existing methods do not introduce a feasible way to enable skip actions under single-pass settings. In contrast, our framework provides a principled method to incorporate skip actions in the single-pass setting, and we prove that it closes the optimality gap while preserving the efficiency of single-pass inference. Moreover, as revealed in 4.2, our approach clusters poor solutions and reduces the training variance.

Moreover, most previous works (Wu et al., 2018; Park et al., 2021; Wang et al., 2021; Zhou et al., 2022; Jeon et al., 2023; Sun et al., 2024; Wang et al., 2025) do not allow skip actions whenever ready tasks exist. These approaches inherit the limitations of list scheduling and thus its inherent optimality gap. Our theoretical analysis identifies this gap, showing that it can lead to significant performance degradation in heavy-task scenarios. The analysis demonstrates that this gap is non-negligible and highlights the importance of corrective mechanisms such as skip actions. We also propose a criterion to evaluate this optimality gap and analyze its advantages (significantly better performance in heavy task cases) and limitations (larger training variance). Moreover, we reveal the influence of skip actions under varying proportions of heavy tasks.

We evaluate our approach on cases containing heavy tasks. Starting from the TPC-H-30 and TPC-H-50 benchmarks, we construct new test instances by replacing different percentages of DAG tasks with tasks that have high resource demands and long processing times. This modification yields a new benchmark designed to assess the ability of algorithms to address optimality gaps. Details of the benchmark construction are provided in Appendix D. Figure 8 reports the relative improvements

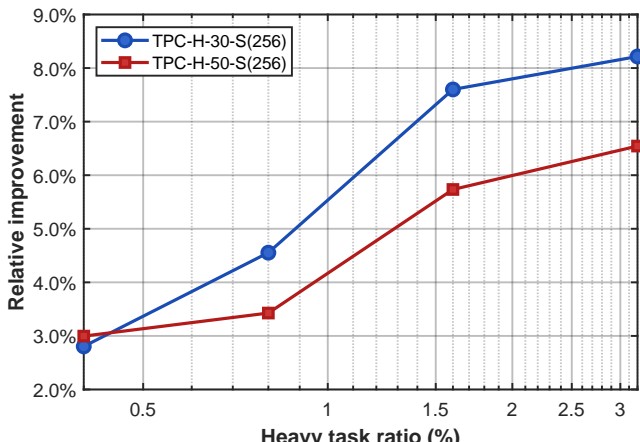

Figure 8: Relative improvement of enabling skip action in single-pass settings under cases with varying percentage of heavy tasks.

obtained by incorporating skip actions on this benchmark. The results show that the benefit of skip actions grows as the proportion of heavy tasks increases. These findings provide practical guidance for determining when skip actions should be incorporated into real-world scheduling applications.

## D   DATASET

### D.1   TPC-H DATASET

In the TPC-H dataset[1], each DAG contains between 2 and 18 nodes, with 9.18 nodes on average to be scheduled. The dataset used in Wang et al. (2021)[2] only includes one type of resource demand, with an average value of 125.8, a minimum of 1, and a maximum of 593. To better reflect heterogeneous real-world scenarios, we extend this dataset by introducing a second resource dimension (representing memory usage) and assigning a random type to each task to indicate its primary computational load (e.g., CPU, IO, etc.). The second resource demand is sampled uniformly from $\{30, 40, 50\}$, and task types are sampled uniformly from $\{0, 1, 2\}$.

We construct training and test datasets under three different problem-size settings, denoted as TPC-H-30, TPC-H-50, and TPC-H-100. Under each setting, a problem instance consists of 30, 50, or 100 DAGs, respectively. The training datasets under these three settings contain 2,464, 3,850, and 7,700 DAGs in total. The corresponding test datasets consist of 10 problem instances built from unseen DAGs. For the test datasets under the TPC-H-50 and TPC-H-100, TPC-H-150 settings, the DAG structures and the first-type resource demands are identical to those used in Wang et al. (2021); we augment them by introducing a second resource dimension and task-type annotations. Since the TPC-H-30 and TPC-H-200 setting was not considered in the prior work, we generate both the DAGs and the associated resource demands from scratch, following the same generation protocol used by Wang et al. (2021).

We schedule tasks on 3 pools: pool 1 is of type 0, and pools 2 and 3 are of type 1. The first resource capacities of the three pools are 600, 800, and 500, respectively, and the second resource capacities are 260, 240, and 240. We define the compatibility coefficients as follows: for pool type 0, the coefficients are $(1.0, 1/0.8, 1/0.7)$ for task types 0, 1, and 2, respectively; for pool type 1, the coefficients are $(0, 1/1.0, 1/1.1)$.

---

[1]http://tpc.org/tpch/default5.asp
[2]https://github.com/Thinklab-SJTU/PPO-BiHyb/tree/main/dag_data/tpch

To study scenarios involving heavy tasks (Figure 3 in the main text), we modify the features of certain tasks. Specifically, we identify non-root tasks with durations exceeding 1000 seconds, randomly select half of them, and assign them a resource demand of $(300, 210)$ and task type 0. Approximately 1.6% of all tasks are replaced in this manner, resulting in an average of 4.8 and 8.0 heavy tasks per problem instance in the TPC-H-30 and TPC-H-50 settings, respectively. To accommodate these heavy tasks, the first resource capacities of the three pools are increased to 1200, 3000, and 800, while the second resource capacities are set to 260, 260, and 240. The compatibility coefficients are kept unchanged.

To evaluate the ability of algorithms to address optimality gaps, we construct a new benchmark (Figure 8) based on TPC-H-30 and TPC-H-50. Specifically, we identify all non-root DAG tasks and replace 0.4%, 0.8%, 1.6% and 3.2% of them with tasks that have high resource demands $(300, 210)$, are of type 0, and require long processing times 2000. The environment settings and compatibility coefficients remain the same as those described in the previous paragraph (Figures 3 and Table 8).

To evaluate the generalization ability of our model, we train it on the original environment and then test its performance across the following eight distinct fluctuating environmental settings.

- **More pools:** (a) adding an extra pool with capacity $(500, 240)$ and type 1; (b) adding an extra pool with capacity $(200, 100)$ and type 0; (i) adding three copies for each pool.

- **More pool types:** (c) modifying the type of pool 3 to a different new pool type with compatibility coefficients $(1/2.0, 1/1.0, 1/1.1)$; (d) modifying the type of pool 3 to a different new pool type with compatibility coefficients $(0, 1/1.2, 1/0.8)$.

- **More task types:** (e) reclassifying half of the type 2 tasks as a new task type, which has a compatibility coefficient of $1/1.4$ with pool type 0 and $1/0.8$ with pool type 1; (f) reclassifying half of the type 2 tasks as another new task type, which has a compatibility coefficient of $1/1.0$ with pool type 0 and $1/2.0$ with pool type 1, and reclassifying the rest half as the new type in (e).

- **Different pool capacity:** (g) increasing the first resource capacities for the three pools to $800, 1100, 500$ and second capacities to $340, 320, 240$; (h) decreasing the first resource capacities for the three pools to $600, 800, 500$ and second capacities to $180, 180, 240$.

The scenarios labeled "more pool", "more pool type" and "more task type" in Figure 2 of the main text correspond to environments (a), (c), and (e) described above, respectively.

### D.2 COMPUTATION GRAPHS

We use the three types of computation graphs introduced in Jeon et al. (2023). The first type, the layered graph, is a synthetic structure designed to resemble the computational graphs commonly found in certain classes of neural networks. The other two types are established families of random undirected graphs, which we convert into DAGs following the same transformation procedure described by Jeon et al. (2023). We generate graphs with specific configurations based on Jeon et al.'s methodology. For the layered graph, we use the parameters $\sigma_N = 0.75, \rho_E = 0.2, \rho_S = 0.14$. For the Erdős-Rényi graph, the edge probability is set to $p = 0.05$. For the stochastic block model, we use the parameters $p_{in} = 0.3$ and $p_{out} = 0.005$. For each instance, we generate 10 graphs of size 50, resulting in a total of 500 task nodes per problem.

Task durations are generated following the approach of Jeon et al. (2023). The duration $t(v)$ for task v is calculated as:

$$t(v) = round(100 \times m(v)) + 1, \tag{4}$$

where $m(v)$ is sampled from a Gaussian mixture model (GMM) and projected onto the non-negative domain. The GMM consists of four Gaussian components, each with equal mixture weight. The means of the components are $(0.5, 1, 3, 5)$, and their corresponding standard deviations are $(0.5, 1, 1, 1)$. For resource requirements, each task's processor demand is uniformly sampled from the set $\{2, 4, 8, 16\}$, and memory demand is selected from $\{1, 2, 3\}$, also with uniform probability. Task type are sample from $\{0, 1, 2\}$ with probability $1/6, 1/6, 2/3$, respectively. Task scheduling is performed with three resource pools: pool 1 is of type 0, while pools 2 and 3 are of type 1. The first resource capacities of the three pools are 16, 12, 64, respectively, and the second resource capacities are 15, 20, 50. The compatibility coefficients of pools with type 0 are $(1.0, 1/0.8, 1/1.2)$ and those for pools with type 1 are $(0, 1/1.2, 1/0.8)$.

### D.3 PROBLEM SIZE OF BENCHMARKS

We summarize the key statistics regarding the average problem size of the benchmarks. Table 5 reports the size of the action set, the number of MILP variables, the number of MILP constraints, and the number of nonzeros in the MILP constraints. The two homogeneous lines with a single pool correspond to the largest benchmark instances tested in prior works (Wang et al., 2021; Jeon et al., 2023). The heterogeneous cases correspond to the benchmarks evaluated in our study. As shown in the table, the heterogeneous settings substantially enlarge the problem size, making the benchmarks we test considerably more challenging than those in previous work.

|  | Action set | variables | constraints | nonzeros |
|---|---|---|---|---|
| Homogeneous | $n$ | $3n^2$ | $6n^2$ | $19n^2$ |
| TPC-H-150, 1 pool | 1500 | $6.8 \times 10^6$ | $1.4 \times 10^7$ | $4.3 \times 10^7$ |
| TPC-H-300, 1 pool | 3000 | $2.7 \times 10^7$ | $5.4 \times 10^7$ | $1.7 \times 10^8$ |
| Heterogeneous | $nn_c$ | $(3 + n_c)n^2$ | $(6 + 2n_c)n^2$ | $(18 + 8n_c + rn_c)n^2$ |
| Computation-Graph, 3 pool | 1500 | $1.5 \times 10^6$ | $3 \times 10^6$ | $1.2 \times 10^7$ |
| TPC-H-100, 3 pool | 3000 | $6.0 \times 10^6$ | $1.2 \times 10^7$ | $4.8 \times 10^7$ |
| TPC-H-200, 3 pool | 6000 | $2.4 \times 10^7$ | $4.8 \times 10^7$ | $2 \times 10^8$ |
| TPC-H-200, 12 pool | 24000 | $6.0 \times 10^7$ | $1.2 \times 10^8$ | $5.5 \times 10^8$ |

Table 5: Problem size of Benchmarks

## E BASELINE ALGORITHMS

### E.1 HEURISTIC BASELINES

We here provide implementation details of heuristic baselines. For clarity, we denote the average processing time $\bar{t}(v)$ for task $v$ as the average of its actual processing times across all compatible resource pools $c$ where $K_{acc}(v, c) > 0$.

- **Shortest first task scheduling (SFT).** The scheduler calculates the average processing time for each task and selects the unmasked task with the smallest value.

- **Most operations remaining (MOPNR).** For each task $v$, we define the number of remaining operations as the number of tasks that directly or indirectly depend on $v$. At each step, the scheduler selects the unmasked task with the largest number of remaining operations.

- **Critical path scheduling (CP).** A directed path originating from task $v$ is any path leading from $v$ to one of its descendants $w$ in the DAG, i.e., a sequence $v = v_0, v_1, \ldots, v_k = w$ in which $(v_i, v_{i+1}) \in E$ for all $i = 0, \ldots, k - 1$. The length of a path is defined as the sum of the average processing times of all tasks along it. The critical path of a task $v$ refers to the longest directed path originating from $v$. The scheduler selects the unmasked task with the largest critical path length.

- **Tetris**(Grandl et al., 2014). A heuristic method for multi-pool cases where jobs are arranged as a Tetris game on the two-dimension space of makespan and resource. The Tetris score between task $v$ and pool $c$ is defined to be the normalized inner product between their resource feature. Let $\rho(v)$ be the resource demand vector of task $v$ and $\lambda(c), \lambda_{cur}(c)$ be the full resource vector and the current resource vector of pool $c$. The Tetris score is defined to be $s_{tetris}(v, c) = (\rho(v)/\lambda(c)) \cdot (\lambda_{cur}(c)/\lambda(c)) = \sum_{k=1}^{r} (\rho^k(v)/\lambda^k(c)) \cdot (\lambda_{cur}^k(c)/\lambda^k(c))$. The scheduler keeps choosing the unmasked task with the highest Tetris score.

- **HEFT**(Topcuoglu et al., 2002). A dependency-aware scheduling algorithm designed for heterogeneous DAG scheduling. It maintains a timeline for each pool and inserts the unmasked task with the highest critical path length into the timeline gap where it can complete at the earliest possible time among all feasible pools.

For each list scheduling based heuristic algorithm (i.e., all except HEFT), the pool assignment throughout the scheduling process is determined by one of the following three rules:

- **Earliest finishing time (EFT).** For a given task $v$, the scheduler selects the pool $c$ that allows the earliest feasible completion, which, in the context of resource compatibility, corresponds to the pool with the largest $K_{acc}(v, c)$ value.

- **Tetris score.** For a given task $v$, the scheduler selects the compatible pool $c$ such that the Tetris score between $v$ and $c$ is the highest.

- **Balance choosing.** For a given task $v$, the scheduler selects the pool $c$ that achieves a balance between fast execution and good resource fit. Specifically, it selects the pool with the largest combined score defined as $s_{tetris}(v, c) \cdot K_{acc}(v, c)$.

Each list scheduling based heuristic (SFT, MOPNR, CP, Tetris) is evaluated under all three pool assignment rules, resulting in three complete schedules. The smallest makespan is reported as the final result, and run time is the sum obtained across the three runs.

### E.2 Neural baselines

We provide the implementation details of two neural baselines. Both employ the earliest finishing time as the rule for pool selection.

- **PPO-BiHyb(Wang et al., 2021).** The network architecture, as well as the training and inference settings, remain consistent with the original source code, except for the node features and the underlying heuristic algorithm. Specifically, the node features include the average processing time (averaged over all pool types) and the real processing times for each individual pool type. We observe that HEFT achieves the best performance among all heuristic baselines on most datasets. Therefore, we adopt HEFT as the underlying heuristic algorithm in the bilevel framework of PPO-BiHyb.

- **OneShot(Jeon et al., 2023).** As the source code of Topoformer is unavailable, Graphormer with additional attention masks is used as the encoder instead. Each task has a feature vector $(\bar{t}, \rho_1, \rho_2)$ of dimension 3, where $\bar{t}$ is the average processing time. This design enables OneShot to handle varying environment structures, facilitating the comparison of its generalization ability with that of our WeCAN. The training settings align with those of our WeCAN, except for an additional $\ell_2$ regularization term in the loss function mentioned in their paper (Jeon et al., 2023).

## F  Additional results

### F.1 Generalization results on large-scale TPC-H dataset

For testing the generalization ability of WeCAN on large-scale problems, we present the experiment results on TPC-H-150 ($\approx 1500$ tasks) and TPC-H-200 ($\approx 1000$). The model is trained in TPC-H-30 with the original 3 pool environment. Table 6 shows results under the original 3 pool environment and Table 7 shows results under 12 pool fluctuating environment (i). Details of the settings are shown in Appendix D. Our results shows that our WeCAN outperforms the best heuristics $\sim 100\%$ and the best learning methods $\sim 5\%$, while remaining comparable running time in greedy mode. This result shows WeCAN has nice scalability to large-scale problems.

Table 6: Full generalization results on large-scale TPC-H datasets

| | TPC-H-150, 3 pools | | | TPC-H-200, 3 pools | | |
| | MakeSpan | Imporvement | Time | MakeSpan | Imporvement | Time |
|---|---|---|---|---|---|---|
| SFT | 129996 | -20.05% | 5.06 | 164320 | -18.14% | 9.27 |
| MOPNR | 119948 | -10.77% | 6.20 | 153326 | -10.23% | 11.4 |
| CP | 112422 | -3.82% | 5.73 | 149163 | -7.24% | 10.2 |
| Tetris | 109079 | -0.73% | 3.65 | 140438 | -0.97% | 6.09 |
| HEFT | 108285 | 0.00% | 3.26 | 139094 | 0.00% | 4.98 |
| PPO-BiHyb | 102414 | 5.42% | 411.81 | 136495 | 1.86% | 538.90 |
| One-Shot-S(256) | 101690±408 | 6.09±0.38% | 15.10 | 131941±255 | 5.14±0.18% | 23.50 |
| WeCAN-Greedy | 96225 | 11.13% | 2.65 | 124345 | 10.60% | 4.97 |
| WeCAN-S(256) | **95491±36** | **11.82±0.03%** | 19.10 | **124140±54** | **10.75±0.04%** | 29.70 |

Table 7: Full generalization results on large-scale TPC-H datasets with on fluctuating environmen (i)

| | TPC-H-150, 12 pools | | | TPC-H-200, 12 pools | | |
| | MakeSpan | Imporvement | Time | MakeSpan | Imporvement | Time |
|---|---|---|---|---|---|---|
| SFT | 35611 | -29.90% | 11.83 | 44727 | -28.40% | 23.54 |
| MOPNR | 32056 | -16.93% | 16.03 | 40217 | -15.45% | 31.75 |
| CP | 28212 | -2.91% | 14.69 | 36361 | -4.38% | 28.20 |
| Tetris | 30552 | -11.44% | 4.80 | 37737 | -8.33% | 8.51 |
| HEFT | 27415 | 0.00% | 3.11 | 34834 | 0.00% | 5.21 |
| PPO-BiHyb | 26866 | 2.00% | 489.82 | 34427 | 1.17% | 778.14 |
| One-Shot-S(256) | 25590 | 6.66% | 16.83 | 32951 | 5.40% | 26.15 |
| WeCAN-Greedy | 24761 | 9.68% | 3.48 | 31864 | 8.53% | 6.09 |
| WeCAN-S(256) | 24321 | 11.29% | 18.36 | 31421 | 9.80% | 30.40 |

## F.2 FULL RESULTS ON HEAVY-TASK CASE OF TPC-H DATASET

We present full experimental results on the TPC-H dataset with heavy tasks (as described in Appendix D and shown in Figure 3) in Table 8. The results indicate that the heuristic algorithm HEFT significantly outperforms the four list scheduling based algorithms, including SFT, MOPNR, CP, and Tetris, since it is not a list scheduling based method and is therefore not subject to the inherent optimality gap associated with such approaches. The neural baseline PPO-BiHyb reaches about 4% improvement compared to HEFT since its implementation utilizes HEFT as the underlying heuristic algorithm. One-Shot and the variant of WeCAN without the skip action perform similarly to HEFT, as both follow the generation map induced by list scheduling and thereby inherit the associated optimality gap. In contrast, the skip action in our WeCAN effectively overcomes this limitation, resulting in an improvement around 8% over HEFT.

## F.3 ADDITIONAL RESULTS ON GENERALIZATION ABILITY

We present full experimental results evaluating the generalization ability of WeCAN. In Tables 9 to 16, we report the performance of models trained in a fixed TPC-H environment and evaluated under eight environment fluctuations (a)-(h), as defined in Appendix D. Additionally, in Table 17 we report the performance of models trained on TPC-H-30 while tested on TPC-H-50 and TPC-H-100, which contain more tasks. In this table, WeCAN-Train30 and OneShot-Train30 indicate the models trained on TPC-H-30 while WeCAN and OneShot denote models trained and tested under the same task setting. The TPC-H-30 results in Tables 9, 11, 13 and 17 correspond to the "more pool", "more pool type", "more task type" and "more task" bars in Figure 2 of the main text, respectively.

The results show that our WeCAN consistently outperforms OneShot and heuristic baselines across all evaluation scenarios. Moreover, it maintains stable performance under both fluctuating environments and varying task numbers, demonstrating strong generalization ability.

Table 8: Full results on TPC-H datasets with heavy tasks.

| | TPC-H-30, $n \approx 300$ | | TPC-H-50, $n \approx 500$ | |
| | MakeSpan | Improvement | MakeSpan | Improvement |
|---|---|---|---|---|
| SFT | 30017 | -11.8% | 43808 | -10.0% |
| MOPNR | 28713 | -6.92% | 41721 | -4.76% |
| Tetris | 29324 | -9.19% | 44296 | -11.2% |
| CP | 28144 | -4.80% | 41378 | -3.90% |
| HEFT | 26856 | 0.00% | 39825 | 0.00% |
| PPO-BiHyb | 25753 | 4.11% | 38039 | 4.49% |
| OneShot-S(256) | $27425 \pm 39$ | $-2.12 \pm 0.15\%$ | $39826 \pm 33$ | $0.00\% \pm 0.08\%$ |
| WeCAN-No-Skip-Greedy | 27138 | -1.05% | 39457 | 0.92% |
| WeCAN-No-Skip-S(256) | $26159 \pm 18$ | $2.59 \pm 0.06\%$ | $38485 \pm 19$ | $3.37 \pm 0.05\%$ |
| WeCAN-With-Skip-Greedy | 25176 | 6.25% | 37652 | 5.46% |
| WeCAN-With-Skip-S(256) | $\mathbf{24630 \pm 20}$ | $\mathbf{8.29 \pm 0.07\%}$ | $\mathbf{36282 \pm 56}$ | $\mathbf{8.90 \pm 0.14\%}$ |

Table 9: Experimental results on TPC-H datasets on fluctuating environment (a). Here TPC-H-30 corresponds to the "more pool" bars in the Figure 3 of main text.

| | TPC-H-30, 3 pools | | TPC-H-50, 3 pools | | TPC-H-100, 3 pools | |
| | MakeSpan | Imporvement | MakeSpan | Imporvement | MakeSpan | Imporvement |
|---|---|---|---|---|---|---|
| SFT | 25556 | -16.8% | 43289 | -14.9% | 77494 | -14.2% |
| MOPNR | 23080 | -5.44% | 40724 | -8.07% | 71166 | -4.85% |
| Tetris | 21889 | 0.00% | 37682 | 0.00% | 67923 | -0.07% |
| CP | 22431 | -2.48% | 39041 | -3.61% | 68278 | -0.60% |
| HEFT | 22622 | -3.35% | 38643 | -2.55% | 67872 | 0.00% |
| OneShot-S(256) | 19830 | 9.40% | 34250 | 9.11% | 62744 | 7.56% |
| WeCAN-Greedy | 18167 | 17.0% | 32244 | 14.4% | 58764 | 13.4% |
| WeCAN-S(64) | 17496 | 20.0% | 31416 | 16.6% | 57732 | 14.9% |
| WeCAN-S(256) | **17430** | **20.4%** | **31157** | **17.3%** | **57404** | **15.4%** |

Table 10: Experimental results on TPC-H datasets on fluctuating environment (b).

| | TPC-H-30, 3 pools | | TPC-H-50, 3 pools | | TPC-H-100, 3 pools | |
| | MakeSpan | Imporvement | MakeSpan | Imporvement | MakeSpan | Imporvement |
|---|---|---|---|---|---|---|
| SFT | 23741 | -27.5% | 41476 | -30.1% | 74061 | -26.3% |
| MOPNR | 21869 | -17.4% | 38129 | -19.7% | 67597 | -15.5% |
| Tetris | 20000 | -7.39% | 32237 | -1.16% | 58796 | -0.45% |
| CP | 21089 | -13.2% | 34598 | -8.57% | 62874 | -7.42% |
| HEFT | 18624 | 0.00% | 31866 | 0.00% | 58531 | 0.00% |
| OneShot-S(256) | 17330 | 6.95% | 29807 | 6.46% | 55739 | 4.77% |
| WeCAN-Greedy | 17119 | 8.08% | 29251 | 8.21% | 55314 | 5.50% |
| WeCAN-S(64) | 16578 | 11.0% | 28580 | 10.31% | 53891 | 7.93% |
| WeCAN-S(256) | **16059** | **11.3%** | **28476** | **10.6%** | **53401** | **8.76%** |

Table 11: Experimental results on TPC-H datasets on fluctuating environment (c). Here TPC-H-30 corresponds to the "more pool type" bars in the Figure 3 of main text.

| | TPC-H-30, 3 pools | | TPC-H-50, 3 pools | | TPC-H-100, 3 pools | |
| | MakeSpan | Imporvement | MakeSpan | Imporvement | MakeSpan | Imporvement |
|---|---|---|---|---|---|---|
| SFT | 26626 | -28.4% | 44708 | -25.1% | 80460 | -16.8% |
| MOPNR | 24792 | -19.6% | 41589 | -16.4% | 75168 | -9.16% |
| Tetris | 22829 | -10.1% | 36691 | -2.70% | 68863 | 0.00% |
| CP | 22519 | -8.58% | 38745 | -8.45% | 72774 | -5.68% |
| HEFT | 20732 | 0.00% | 35727 | 0.00% | 68996 | -0.19% |
| OneShot-S(256) | 20545 | 0.90% | 36091 | -1.02% | 69410 | -0.79% |
| WeCAN-Greedy | 19816 | 4.42% | 32893 | 7.93% | 65760 | 4.51% |
| WeCAN-S(64) | 19449 | 6.19% | 32696 | 8.48% | 65326 | 5.14% |
| WeCAN-S(256) | **19340** | **6.71%** | **32675** | **8.54%** | **65199** | **5.32%** |

Table 12: Experimental results on TPC-H datasets on fluctuating environment (d).

| | TPC-H-30, 3 pools | | TPC-H-50, 3 pools | | TPC-H-100, 3 pools | |
| | MakeSpan | Imporvement | MakeSpan | Imporvement | MakeSpan | Imporvement |
|---|---|---|---|---|---|---|
| SFT | 27303 | -21.8% | 47400 | -22.6% | 84195 | -21.2% |
| MOPNR | 24964 | -11.4% | 43010 | -11.3% | 77059 | -11.0% |
| Tetris | 23203 | -3.51% | 38885 | -0.61% | 71034 | -2.29% |
| CP | 23129 | -3.18% | 41465 | -7.29% | 73481 | -5.82% |
| HEFT | 22416 | 0.00% | 38648 | 0.00% | 69443 | 0.00% |
| OneShot-S(256) | 19938 | 11.1% | 35018 | 9.39% | 65561 | 5.59% |
| WeCAN-Greedy | 19146 | 14.6% | 33282 | 13.9% | 61271 | 11.8% |
| WeCAN-S(64) | 18716 | 16.5% | 32526 | 15.8% | 60285 | 13.2% |
| WeCAN-S(256) | **18569** | **17.2%** | **32417** | **16.1%** | **60051** | **13.5%** |

Table 13: Experimental results on TPC-H datasets on fluctuating environment (e). Here TPC-H-30 corresponds to the "more task type" bars in the Figure 3 of main text.

| | TPC-H-30, 3 pools | | TPC-H-50, 3 pools | | TPC-H-100, 3 pools | |
| | MakeSpan | Imporvement | MakeSpan | Imporvement | MakeSpan | Imporvement |
|---|---|---|---|---|---|---|
| SFT | 27254 | -19.7% | 49040 | -27.4% | 83922 | -23.0% |
| MOPNR | 24809 | -8.97% | 43135 | -12.1% | 77623 | -13.7% |
| Tetris | 23629 | -3.78% | 38484 | 0.00% | 71358 | -4.56% |
| CP | 23488 | -3.16% | 41564 | -8.00% | 72609 | -6.40% |
| HEFT | 22768 | 0.00% | 40136 | -4.29% | 68243 | 0.00% |
| OneShot-S(256) | 20435 | 10.2% | 35093 | 8.81% | 64709 | 5.18% |
| WeCAN-Greedy | 18908 | 17.0% | 33950 | 11.8% | 61759 | 9.50% |
| WeCAN-S(64) | 18456 | 18.9% | 33179 | 13.8% | 60752 | 11.0% |
| WeCAN-S(256) | **18365** | **19.34%** | **32994** | **14.3%** | **60278** | **11.7%** |

Table 14: Experimental results on TPC-H datasets on fluctuating environment (f).

| | TPC-H-30, 3 pools | | TPC-H-50, 3 pools | | TPC-H-100, 3 pools | |
|---|---|---|---|---|---|---|
| | MakeSpan | Imporvement | MakeSpan | Imporvement | MakeSpan | Imporvement |
| SFT | 29738 | -14.5% | 50118 | -24.3% | 94002 | -20.5% |
| MOPNR | 26427 | -1.71% | 45473 | -12.8% | 86306 | -10.6% |
| Tetris | 25983 | 0.00% | 40322 | 0.00% | 78031 | 0.00% |
| CP | 26427 | -1.71% | 43760 | -8.52% | 85981 | -10.2% |
| HEFT | 26707 | -2.79% | 41042 | -1.78% | 83114 | -6.51% |
| OneShot-S(256) | 22573 | 13.1% | 36077 | 10.5% | 69004 | 11.6% |
| WeCAN-Greedy | 22864 | 12.0% | 36088 | 10.5% | 68934 | 11.7% |
| WeCAN-S(64) | 21847 | 15.9% | 34539 | 14.3% | 68299 | 12.5% |
| WeCAN-S(256) | **21801** | **16.1%** | **34378** | **14.7%** | **67722** | **13.2%** |

Table 15: Experimental results on TPC-H datasets on fluctuating environment (g).

| | TPC-H-30, 3 pools | | TPC-H-50, 3 pools | | TPC-H-100, 3 pools | |
|---|---|---|---|---|---|---|
| | MakeSpan | Imporvement | MakeSpan | Imporvement | MakeSpan | Imporvement |
| SFT | 20842 | -20.1% | 36076 | -24.4% | 63677 | -20.3% |
| MOPNR | 18417 | -6.09% | 32614 | -12.4% | 57660 | -8.93% |
| Tetris | 18417 | -6.09% | 30438 | -4.92% | 54054 | -2.11% |
| CP | 17790 | -2.48% | 29502 | -1.70% | 53009 | -0.14% |
| HEFT | 17360 | 0.00% | 29010 | 0.00% | 52935 | 0.00% |
| OneShot-S(256) | 15592 | 10.19% | 27420 | 5.48% | 50939 | 3.77% |
| WeCAN-Greedy | 15982 | 7.94% | 27050 | 6.75% | 50361 | 4.86% |
| WeCAN-S(64) | 15013 | 13.5% | 26159 | 9.83% | 48882 | 7.66% |
| WeCAN-S(256) | **13498** | **22.3%** | **25990** | **10.4%** | **48711** | **7.98%** |

Table 16: Experimental results on TPC-H datasets on fluctuating environment (h).

| | TPC-H-30, 3 pools | | TPC-H-50, 3 pools | | TPC-H-100, 3 pools | |
|---|---|---|---|---|---|---|
| | MakeSpan | Imporvement | MakeSpan | Imporvement | MakeSpan | Imporvement |
| SFT | 33138 | -19.4% | 54852 | -16.7% | 100653 | -16.4% |
| MOPNR | 29829 | -7.45% | 50931 | -8.40% | 92989 | -7.50% |
| Tetris | 29179 | -5.11% | 48891 | -4.06% | 91940 | -6.29% |
| CP | 27760 | 0.00% | 46985 | 0.00% | 86498 | 0.00% |
| HEFT | 28119 | -1.29% | 47602 | -1.31% | 88638 | -2.47% |
| OneShot-S(256) | 25349 | 8.68% | 44122 | 6.09% | 82797 | 4.28% |
| WeCAN-Greedy | 24525 | 11.65% | 41563 | 11.54% | 79378 | 8.23% |
| WeCAN-S(64) | 23791 | 14.3% | 40778 | 13.21% | 78293 | 9.49% |
| WeCAN-S(256) | **23628** | **14.9%** | **40705** | **13.4%** | **78009** | **9.81%** |

Table 17: Experimental results on TPC-H datasets. OneShot-Train30 and WeCAN-Train30 refer to models trained in TPC-H-30 while OneShot and WeCAN refer to model trained in problems with corresponding size (TPC-H-50 or TPC-H-100)

|  | TPC-H-50 | | TPC-H-100 | |
| --- | --- | --- | --- | --- |
|  | MakeSpan | Imporvement | MakeSpan | Imporvement |
| SFT | 49172 | -27.2% | 84986 | -21.2% |
| MOPNR | 43545 | -12.7% | 74364 | -10.3% |
| CP | 41597 | -7.62% | 74364 | -6.03% |
| Tetris | 38654 | 0.00% | 71269 | -1.65% |
| HEFT | 36326 | -1.71% | 70137 | 0.00% |
| OneShot-Train30-S(256) | 36326 | 6.02% | 66777 | 4.79% |
| OneShot-S(256) | 35561 | 8.00% | 66173 | 5.65% |
| WeCAN-Train30-Greedy | 33654 | 12.9% | 63034 | 10.1% |
| WeCAN-Greedy | 33428 | 13.5% | 62587 | 10.8% |
| WeCAN-Train30-S(64) | 33265 | 13.9% | 61894 | 11.8% |
| WeCAN-S(64) | 32927 | 14.8% | 61706 | 12.0% |
| WeCAN-Train30-S(256) | 33137 | 14.3% | 61812 | 11.9% |
| WeCAN-S(256) | **32885** | **14.9%** | **61364** | **12.5%** |

## F.4    RESULTS ON LOCAL SEARCH

In Appendix A.5, we mentioned that local search with the projection map $S_n$ can serve as an alternative approach to overcome the optimality gap, as illustrated in Figure 7. Here we present the corresponding experimental results on the TPC-H dataset with heavy tasks. The experimental setup is the same as in Table 8, with details provided in Appendix D, except that we use only one random seed due to the high computational cost of local search. The local search approach is described in Appendix A.5. We evaluate its performance on two versions of WeCAN: one that incorporates the skip action, and one that does not. For both settings, we report local search results using 50 and 100 search steps.

As shown in Table 18, local search yields substantially greater improvements when applied to WeCAN without the skip action. This is because WeCAN with the skip action has already addressed the optimality gap inherent in list scheduling, thereby leaving limited room for further enhancement through local search. In contrast, applying local search to the configuration without the skip action helps reduce the remaining suboptimality and achieve robust performance under heavy-task conditions. However, these gains come at the cost of significantly increased runtime, which is primarily due to repeated calls to the generation map that dominate the computational cost. In summary, although local search with the projection map $S_n$ is a viable method for mitigating the optimality gap, incorporating the skip action offers a more computationally efficient alternative and achieves comparable robustness in heavy-task scenarios.

Table 18: Experimental results on TPC-H-heavy datasets.

|  | TPC-H-30 | | TPC-H-50 | |
| --- | --- | --- | --- | --- |
|  | MakeSpan | Time | MakeSpan | Time |
| HEFT | 26856 | 0.19 | 39825 | 0.53 |
| WeCAN-NoSkip-Greedy | 27138 | 0.17 | 39457 | 0.47 |
| WeCAN-NoSkip-S(256) | 26175 | 2.38 | 38450 | 5.37 |
| WeCAN-NoSkip-S(256)+LocalSearch(50) | 24946 | 92.80 | 37726 | 275.57 |
| WeCAN-NoSkip-S(256)+LocalSearch(100) | 24793 | 192.56 | 37519 | 526.22 |
| WeCAN-Greedy | 25176 | 0.13 | 37652 | 0.36 |
| WeCAN-S(256) | 24611 | 2.43 | 36177 | 6.12 |
| WeCAN-S(256)+LocalSearch(50) | 24122 | 87.87 | 35777 | 240.68 |
| WeCAN-S(256)+LocalSearch(100) | 24092 | 181.34 | 35751 | 465.11 |

## G MODEL ARCHITECTURE OF WECAN

### G.1 MODEL DETAILS OF WHOLE ARCHTECTURE

The network architecture, illustrated in the main text, processes task and pool features as follows. Each task is initially represented by a 3-dimensional feature vector $(t, \rho_1, \rho_2)$, and each pool by a feature vector $(\lambda_1, \lambda_2)$ (here we assume two types of resource demands). These are then embedded using separate multi-layer perceptrons (MLPs) into higher-dimensional task features $\boldsymbol{f}_v$ and pool features $\boldsymbol{f}_c^c$, respectively, both matching the encoder's dimension.

Subsequently, a weighted cross-attention layer and the longest directed distance graph neural network (LDDGNN), detailed in the main text, are applied to update the task features to $\boldsymbol{h}_v^L$. Following the LDDGNN, a linear transformation reduces the task feature dimension to match the decoder dimension. Subsequently, two weighted cross-attention layers are employed to update the task feature $\boldsymbol{h}_v$ and pool feature $\boldsymbol{h}_c^c$:

$$\boldsymbol{q}_c^1 = \boldsymbol{W}_1^Q \boldsymbol{f}_c^c, \quad \boldsymbol{K}^1 = \boldsymbol{W}_1^K[\boldsymbol{h}_{v(1)}^L, ..., \boldsymbol{h}_{v(n)}^L], \quad \boldsymbol{V}^1 = \boldsymbol{W}_1^V[\boldsymbol{h}_{v(1)}^L, ..., \boldsymbol{h}_{v(n)}^L],$$

$$\boldsymbol{h}_c^c = \boldsymbol{f}_c^c + \frac{\mathrm{softmax}((\boldsymbol{q}_c^1)^T \boldsymbol{K}^1)}{\sqrt{d}} \mathrm{diag}\{K_{acc}(v(1), c), ..., K_{acc}(v(n), c)\} \boldsymbol{V}^1, \tag{5}$$

$$\boldsymbol{q}_v^2 = \boldsymbol{W}_2^Q \boldsymbol{h}_v^L, \quad \boldsymbol{K}^2 = \boldsymbol{W}_2^K[\boldsymbol{h}_{c(1)}^c, ..., \boldsymbol{h}_{c(n_c)}^c], \quad \boldsymbol{V}^2 = \boldsymbol{W}_2^V[\boldsymbol{h}_{c(1)}^c, ..., \boldsymbol{h}_{c(n_c)}^c],$$

$$\boldsymbol{h}_v = \boldsymbol{h}_v^L + \frac{\mathrm{softmax}((\boldsymbol{q}_v^2)^T \boldsymbol{K}^2)}{\sqrt{d}} \mathrm{diag}\{K_{acc}(v, c(1)), ..., K_{acc}(v, c(n_c))\} \boldsymbol{V}^2. \tag{6}$$

Finally, a output layer, as detailed in the main text, computes the action scores. The skip coefficients are determined by an MLP that takes the mean-pooled task features($\frac{1}{|V|} \sum_{v \in V} \boldsymbol{h}_v$) and mean-pooled pool features ($\frac{1}{|C|} \sum_{c \in C} \boldsymbol{h}_c^c$) as input. All three weighted cross-attention layers utilize a multi-head attention mechanism: multiple independent attention heads compute attention scores, and their outputs are concatenated to form the updated features.

The network ultimately produces scores $\hat{u}_\pi$ for each action. To prevent the numerical issue, these scores are clipped to the range $[-C_{clip}, C_{clip}]$ (where $C_{clip} = 15$) using the $\tanh$ function:

$$u_\pi = C_{clip} \cdot \tanh\left(\frac{\hat{u}_\pi}{C_{clip}}\right).$$

### G.2 LONGEST DIRECTED DISTANCE BASED GRAPH NEURAL NETWORK (LDDGNN)

In DAG scheduling, edges $e = (v_1, v_2) \in E$ coexisting with another directed path from $l$ from $v_1$ to $v_2$ are redundant; removing them preserves scheduling constraints. This equivalence allows us to focus on the eliminated graph formed by removing all these redundant edges. We defined the Lognest directed distance (LDD) $d_e(v, w)$ as the directed length of the longest directed path between $v$ and $w$. The edges of the eliminated graph can be identified through LDD $d_e(v, w) = \pm 1$, enabling using LDD to implicitly represent this structure without explicit construction. Furthermore, to capture broader connectivity, we extend LDD to $d_e(v, w) = +\infty$ for undirected connected but not directed connected node pairs and $d_e(v, w) = -\infty$ for disconnected node pairs. The LDD is leveraged to define both the attention masks and attention biases within the MHA sub-layer. Eight distinct types of attention mask selectively perform attention between node pairs based on LDD values.

The LDDGNN leverages the longest directed distances (LDD) within the DAG. Its node embedding update is defined as:

$$\boldsymbol{q}_v^{l,j} = \boldsymbol{W}_{l,j}^Q \boldsymbol{h}_v^{l-1}, \; \boldsymbol{K}^{l,j} = \boldsymbol{W}_{l,j}^K[\boldsymbol{h}_{v(1)}^{l-1}, ..., \boldsymbol{h}_{v(n)}^{l-1}], \; \boldsymbol{V}^{l,j} = \boldsymbol{W}_{l,j}^V[\boldsymbol{h}_{v(1)}^{l-1}, ..., \boldsymbol{h}_{v(n)}^{l-1}],$$

$$\hat{\boldsymbol{h}}_v^l = \boldsymbol{h}_v^{l-1} + \mathrm{concat}_j\left[\mathrm{softmax}\left((\boldsymbol{q}_v^{l,j})^T \boldsymbol{K}^{l,j} + [b_{d_e(v,v(1))}, ..., b_{d_e(v,v(n))}] + [M_{v,v(1)}^j, ..M_{v,v(n)}^j]\right) \boldsymbol{V}^{l,j}\right],$$

$$\boldsymbol{h}_v^l = \hat{\boldsymbol{h}}_v^l + \mathrm{MLP}_{(l)}(\hat{\boldsymbol{h}}_v^l),$$

where $V = \{v(1), ..., v(n)\}$ is the set of nodes and $\boldsymbol{h}_v^0 = \boldsymbol{g}_v$. Note that the softmax function, included here for completeness, is omitted in high-level descriptions in the main text for presentation clarity.

The LDDGNN incorporates attention masks $M_{v,w}^j$ and bias terms $b_{d_e(v,w)}$ both derived from the LDD. Let $N_1 = \mathbb{Z} \cup \{-\infty, +\infty\}$, $N_2 = \{1\}$, $N_3 = \{-1\}$, $N_4 = \{2\}$, $N_5 = \{-2\}$, $N_6 = [3, +\infty) \cap \mathbb{Z}$, $N_7 = (-\infty, -3] \cap \mathbb{Z}$, $N_8 = \{+\infty\}$. Given the number of attention heads, $n_{head}$ (a multiple of 8), the mask $M_{v,w}^j$ ($j = 0, ..., n_{head} - 1$) for each attention head is calculated by

$$M_{v,w}^j = \begin{cases} 0 & \text{if } d_e(v,w) \in N_{\lfloor 8j/n_{head} \rfloor + 1} \\ -\infty & \text{otherwise} \end{cases}. \tag{7}$$

For the bias calculation, a hyperparameter $D_{max} = 500$ is employed. First, the folded longest directed distance $d_f(v,w)$ is calculated:

$$d_f(v,w) = \begin{cases} d_e(v,w) & \text{if } d_e(v,w) \in [-D_{max} + 1, D_{max} - 1] \\ D_{max} - 1 & \text{if } d_e(v,w) \in [D_{max}, +\infty) \\ -D_{max} + 1 & \text{if } d_e(v,w) \in (-\infty, -D_{max}] \\ D_{max} & \text{if } d_e(v,w) = +\infty \\ -D_{max} & \text{if } d_e(v,w) = -\infty \end{cases}. \tag{8}$$

Then the bias $b_{d_e(v,w)}$ is obtained by selecting the $(d_f(v,w) + D_{max} + 1)$-th parameter from a learnable embedding table of size $2D_{max} + 1$. This mechanism is equivalent to applying a one-hot encoding to the folded distance and then passing it through a linear layer.

### G.3 INSIDE VERSION OF WEIGHTED CROSS-ATTENTION

As mentioned in the main text, there exists an "inside" version of the weighted cross-attention layer. This version updates the task features as follows:

$$\boldsymbol{q}_v = \boldsymbol{W}^Q f_v, \quad \boldsymbol{K}^c = \boldsymbol{W}^K [\boldsymbol{f}_{c(1)}^c, ..., \boldsymbol{f}_{c(n_c)}^c], \quad \boldsymbol{V}^c = \boldsymbol{W}^V [\boldsymbol{f}_{c(1)}^c, ..., \boldsymbol{f}_{c(n_c)}^c],$$

$$\boldsymbol{g}_v = \boldsymbol{f}_v + \frac{1}{\sqrt{d}} \text{softmax}\left(\boldsymbol{q}_v^T \boldsymbol{K}^c + [\log K_{acc}(v, c(1)), ..., \log K_{acc}(v, c(n_c))]\right) \boldsymbol{V}^c,$$

where $C = \{c(1), ..., c(n_c)\}$ is the pool set, $K_{acc}$ is the compatibility coefficient and $d$ is the dimension of $\boldsymbol{q}_v$. The distinction between this "inside" form and the previously described "outside" form lies primarily in the placement of the normalization step within the softmax computation (which is composed of an exponentiation followed by normalization).

## H IMPLEMENTATION DETAILS

### H.1 MODEL HYPERPARAMETERS

We employ our WeCAN model with an encoder dimension of $d_{encoder} = 512$ and a decoder dimension of $d_{decoder} = 128$. A single MLP is used to embed the initial features of tasks, and a separate MLP is used for resource pools. Specifically, each MLP consists of a linear layer that maps from 3-dimensional (for tasks) or 2-dimensional (for pools) input features to the shared embedding dimension $d_{encoder} = 512$, followed by a GELU activation. In the encoder, weighted cross-attention layers use $d_{encoder} = 512$ for $q_v$, $k_v$, and $v_c$. In the decoder, they use $d_{decoder} = 128$ for $q_c^1$, $k_v^1$, $v_v^1$, $q_v^2$, $k_v^2$, and $v_c^2$. All weighted cross-attention layers use 8 attention heads. The longest directed distance graph neural network adopts an embedding dimension of $d_{encoder} = 512$, with 8 layers and 16 attention heads (with each task type corresponding to 2 heads). The coefficients $u_a$, $u_b$, and $u_c$ are computed by an MLP consisting of two hidden layers with a hidden dimension of 64, using GELU and Sigmoid as activation functions.

### H.2 TRAINING DETAILS

We train our model using the ADAM optimizer with a learning rate of $10^{-4}$. The training batch size is set to 64 by default. For TPC-H-100, a smaller batch size of 32 is adopted due to GPU memory limitations. The model is trained for 800 batches. The training process does not include any information about the test dataset. Training durations for different configurations are summarized in Table 19.

We consider two baseline strategies for REINFORCE: (1) the average reward of all samples; (2) the reward of a greedy rollout obtained using a separate model that is periodically synchronized with the current policy. In practice, we find that the average reward baseline leads to faster and smoother convergence. Based on this observation, we adopt it as the default baseline for training.

As detailed in Appendix J, the generation map module is the primary computational bottleneck during both training and inference. To improve efficiency, we implement a PyTorch-based version of the generation map that runs on GPU. This GPU-based implementation is used during training and also in sample-based inference, where large batch sizes or many samples are processed. For greedy inference, however, we use a CPU-based version of the generation map, as GPU acceleration provides no runtime benefit in this setting.

Table 19: Experimental results about training time of WeCAN-Greedy.

| Experiment | Total time | Time per batch |
|---|---|---|
| TPC-H-30 | 6.6h | 0.5min |
| TPC-H-50 | 14.5h | 1.1min |
| TPC-H-100 | 26h | 2min |
| Erdős-Rényi graph | 19h | 1.4 min |
| Layer Graphs | 15.5h | 1.2 min |
| Stochastic Block | 13.3h | 1 min |
| TPC-H-30-heavy | 6.5h | 0.5min |
| TPC-H-50-heavy | 14h | 1.1min |

Table 20: Experimental results about inference time (s).

| Experiment | GPU-inference | | | CPU-inference | | |
|---|---|---|---|---|---|---|
| | Total | Generation map | Network | Total | Generation map | Network |
| TPC-H-30 | 0.154 | 0.140 | 0.014 | 0.201 | 0.116 | 0.085 |
| TPC-H-50 | 0.480 | 0.461 | 0.019 | 0.587 | 0.442 | 0.145 |
| TPC-H-100 | 1.72 | 1.71 | 0.017 | 1.92 | 1.53 | 0.391 |
| Erdős-Rényi graph | 0.533 | 0.512 | 0.020 | 0.548 | 0.393 | 0.155 |
| Layer Graphs | 0.256 | 0.238 | 0.018 | 0.372 | 0.214 | 0.158 |
| Stochastic Block | 0.361 | 0.342 | 0.019 | 0.527 | 0.367 | 0.160 |

## H.3 INFERENCE DETAILS

Since the skip action has a limited impact in regular TPC-H instances (i.e., those without heavy tasks) and tends to increase the variance of makespan, we disable the skip action across all experiments on regular TPC-H datasets. This allows for more stable and reliable evaluation of model performance and component contributions. Five random seeds: 2000, 2001, 2002, 2003, 2004 are used for calculating the mean and standard deviation of results in Tables 1 to 8. A single fixed random seed of 2000 are used for the results in Tables 9 to 18.

## I COMPUTER RESOURCES

All experiments are conducted on an internal cluster with NVIDIA Tesla A100 (80 GB), 4×Intel Xeon Silver 4310 2.10GHz (12 cores) and 192GB memory.

## J DETAILS ON INFERENCE TIME

In Table 20 we provide a detailed breakdown of the inference time. We observe that network computation accounts for less than 10% of the total inference time, while the generation map dominates the overall runtime. This explains why the total inference time cannot be significantly

lower than heuristic baselines, despite using a fast network. Prior methods that rely on multi-round network inference scaling with the number of tasks incur substantially higher inference times than those dictated by the generation map alone. In contrast, our approach requires only a single forward pass of the network, enabling substantially faster inference overall. Additionally, we report inference time when using only CPU (i.e., without GPU acceleration). As shown in Table 20, WeCAN-Greedy on CPU achieves evaluation times only slightly higher than on GPU, demonstrating that our method remains practical and efficient in scheduling scenarios where only CPU resources are available.

## K  COMPARISON AND BROADER IMPACT

To incorporate task-resource compatibilities, we compare four embedding strategies: (1) averaging task processing time over compatible pools; (2) one-hot encoding of task types; (3) embedding the full compatibility coefficients into task features with a fixed dimension; (4) our proposed weighted cross-attention (WeCA) layers.

Our approach (4) consistently outperforms approach (1) while offering broader applicability than approaches (2) and (3). Specifically, approach (2) fails to generalize when the number of task types changes or when tasks have distinct compatibility coefficients. Approach (3) struggles to handle cases where the number of resource pools changes dynamically. In contrast, both approaches (1) and (4) are capable of handling such dynamic settings with a single trained model, but our approach (4) achieves significantly better performance than (1). However, under fixed environments with a known number of task types and resource pools, approaches (2) and (3) may achieve performance comparable to WeCAN.

Our WeCAN model is designed to address heterogeneous DAG scheduling problems with compatibility constraints. Nonetheless, real-world heterogeneous scheduling often involves additional considerations such as inter-task communication costs, an aspect not currently modeled in WeCAN or reflected in our datasets. Extending the WeCAN framework to address such scenarios remains an important direction for future research.

Here, we discuss the broader impact of our WeCAN framework. Organizations adopting WeCAN for task scheduling may gain efficiency, reducing costs and boosting productivity. Society benefits from optimized resource use, potentially lowering energy consumption and environmental impact in applications like cloud computing. WeCAN's optimization may lead to inequitable resource or task allocations, favoring certain pools or workers. This risks unfair access or workloads, particularly in diverse environments. Companies should integrate fairness metrics into WeCAN's policy to ensure equitable scheduling and mitigate disparities.

## L  LICENSES AND FURTHER INFORMATION ON USED ASSETS

The usage of TPC-H dataset is under clause 9 of the End-User License Agreement (EULA) of TPC[3]: "THE TPC SOFTWARE IS AVAILABLE WITHOUT CHARGE FROM TPC". All assets from Graphormer (Ying et al., 2021) are used under the terms of the MIT License.

## M  THE USE OF LARGE LANGUAGE MODELS

Large Language Models (LLM) is used only for polishing the writing.

---

[3]http://tpc.org/TPC_Documents_Current_Versions/txt/TPC-EULA_v2.2.0.txt

