# OpenReview forum: "Reinforcement Learning for Heterogeneous DAG Scheduling with Weighted Cross-Attention"
_ICLR.cc/2026/Conference — Submitted to ICLR 2026_

### Official Review · Reviewer_DcSo · 2025-10-25

**Soundness:** 3
**Presentation:** 3
**Contribution:** 3
**Rating:** 6
**Confidence:** 3

**Summary:**

This paper introduces WeCAN, a novel, end-to-end reinforcement learning framework for scheduling Directed Acyclic Graphs (DAGs) in heterogeneous environments. The core problems it addresses are (1) adapting to diverse resource pools and task-pool compatibility coefficients, and (2) overcoming the inherent "optimality gap" of fast, single-pass list-scheduling methods. The main contributions are a new **Weighted Cross-Attention (WeCA) layer** to flexibly encode task-pool compatibility and a novel method to enable a dynamic **skip action** in a single-pass, non-auto-regressive decoder. The authors provide theoretical justification for their approach and demonstrate state-of-the-art performance in terms of schedule quality (makespan) and inference speed on real-world and synthetic benchmarks.

**Strengths:**

1.  **Problem Significance:** The paper tackles an NP-hard problem that is highly relevant to modern distributed systems, cloud computing, and data centers. The focus on *heterogeneity* (diverse pools and task compatibility) and *adaptability* (generalizing to new numbers of tasks/pools) addresses key practical challenges that are often abstracted away.

2.  **Novelty of WeCA Layer:** The proposed Weighted Cross-Attention (WeCA) layer is a novel and effective mechanism for this problem. [cite_start]Unlike prior works that either average compatibility coefficients (losing information) or use fixed-size one-hot encodings (losing adaptability), WeCA integrates these coefficients directly into the attention mechanism[cite: 351, 354, 359]. This allows the model to learn context-dependent task features while remaining scalable and adaptable to varying numbers of task types and pools. [cite_start]The ablation study (Table 3) confirms its superiority over alternative designs[cite: 701, 778].

3.  **Novelty of Single-Pass Skip Action:** This is arguably the paper's strongest technical contribution. The authors correctly identify that fast, list-scheduling-based methods suffer from an "optimality gap" because they cannot "wait" (skip) for a better scheduling opportunity. While skip actions exist (e.g., Mao et al., 2016), they are typically used in multi-round, auto-regressive models, which are slow. [cite_start]This paper introduces a method to enable a *dynamic* skip action within a *single-pass, non-auto-regressive* framework [cite: 440-444, 508]. [cite_start]This is achieved by having the network output three coefficients ($u_a, u_b, u_c$) that define a dynamic score for skipping, which elegantly solves the challenge of enabling "waiting" without sacrificing single-pass efficiency[cite: 442].

4.  **Theoretical Justification:** The paper provides a solid theoretical analysis in Section 4 to motivate the skip action. [cite_start]By defining the "original space" of all feasible schedules and a "reduced space" of schedule orders, it formally argues that the standard list scheduling map ($S_{list}$) is not surjective and can therefore miss the optimal solution[cite: 529, 1607, 1777]. [cite_start]It then proves that their proposed map (Algorithm 1), which includes the skip action, is surjective and can generate an optimal schedule (Theorem 1) [cite: 448-450, 1673-1674]. This provides a strong foundation for *why* the skip action is necessary.

5.  **Strong Empirical Results:** The experimental validation is comprehensive and convincing.
    * [cite_start]**Performance:** WeCAN outperforms all heuristic and SOTA RL baselines (PPO-BiHyb, One-Shot) on both TPC-H and Computation Graph datasets (Tables 1 & 2), achieving up to an 18.1% makespan improvement[cite: 621, 629, 645, 690].
    * [cite_start]**Efficiency:** The greedy version (WeCAN-Greedy) is extremely fast, with inference times comparable to simple heuristics and orders of magnitude faster than multi-round RL methods like PPO-BiHyb[cite: 629, 684].
    * **Skip Action Ablation:** The ablation study on "heavy task" datasets (Table 8, Fig. 3) is a key result. [cite_start]It shows that WeCAN-with-Skip significantly outperforms WeCAN-No-Skip and all list-scheduling-based methods, empirically proving the value of the skip action in closing the optimality gap [cite: 772, 781-785, 2549].
    * [cite_start]**Generalization:** The WeCA design is validated in generalization experiments (Fig. 2, Appx. F.3), where a model trained on one environment setting shows robust performance when tested on environments with more pools, new pool types, or new task types [cite: 686-688, 770, 2536].

**Weaknesses:**

1.  **Clarity of Section 4:** While the theoretical analysis in Section 4 is a strength, its presentation is very dense and difficult to parse. The definitions of "reduced space B", "original space A", and the maps $T$, $S_{list}$, and $S_n$ [cite: 512-515, 1223-1229, 1290-1291] could be explained more intuitively. The core idea—that list scheduling is overly greedy and can't "wait" for a better task to become available—is clear, but the formalism is challenging.

2.  **Skip Score Formulation:** The formulation for the skip score, $u_{\pi_{skip}}=u_{a}(1-\frac{k}{2n})^{u_{b}}+u_{c}$[cite: 442], feels somewhat ad-hoc. The paper justifies it as preventing the skip action from being "overly prioritized", but it would be beneficial to include more justification or an ablation study on this specific functional form. How sensitive is the model to this design choice versus, for example, a simpler linear decay?

3.  **Ablation on LDDGNN:** The LDDGNN component, while shown to be effective (beating GAT in Table 3) [cite: 701, 780], is presented as a secondary contribution [cite: 422-429]. It's not entirely clear how much of the performance gain comes from this sophisticated GNN versus the two primary contributions (WeCA and the skip action). An ablation with a simpler GNN (e.g., GIN or GCN), combined with WeCA and the skip mechanism, would help isolate the true impact of the main proposals.

4.  **"Heavy Task" Scenarios:** The "heavy task" ablation (Table 8) is excellent, but the method for generating these tasks (randomly replacing 1% or more of tasks) [cite: 2281-2282, 2284-2285] feels synthetic. The paper's impact could be strengthened by discussing how this scenario maps to real-world workloads (e.g., are 1-3% heavy tasks common?) or by testing on a real-world dataset known to have this "heavy-task" property. Figure 8[cite: 2132], which shows the benefit of skip scaling with the heavy task ratio, is very informative and might be better placed in the main paper.

**Questions:**

See above

---

> ### Author Response · Authors · 2025-11-22
>
> We are grateful to the reviewer for the exceptionally thorough summary and positive evaluation of our work. We sincerely appreciate the insightful feedback, which provides clear and valuable directions for improving the paper's clarity and impact. To address these suggestions, we have conducted new ablations on our skip-score design, and in our revision, we will improve the clarity of our theoretical framework and move the recommended figure into the main text.
>
> ## Response to Weakness 1
>    We thank the reviewer for the suggestions and will improve our presentation. To clarify the terminology, the "original space" refers to the set of all possible feasible schedules. In contrast, the "feasible reduced space" consists of the processing orders and pools (simply refer as schedule orders) of these schedules. The "reduced space" consists of all arbitrary schedule orders, regardless of their feasibility. $T$ can be seen as a map that extracts the schedule orders from a given feasible schedule. Conversely, maps like $S_{list}$ (list scheduling) and $S_n$ are methods that construct a final schedule from a given schedule orders. We will incorporate these clearer definitions into the revised manuscript to improve readability.

---

> ### Author Response · Authors · 2025-11-22
>
> ## Response to Weakness 2
>    While our method of introducing skip actions into the single-pass framework is theoretically validated to ensure optimal solutions can be found, we acknowledge that our specific scoring function may not be the only empirically effective design. We present two key intuitions behind our proposed skip score formulation:
>    1. Our skip score formulation, $u_a(1-\frac{k}{2n})^{u_b} + u_c$ (where $u_a,u_b\ge 0$),  is deliberately designed to be a decreasing function of $k$. This is because according to our theoretical results, a non-decreasing term cannot restore the surjectivity and guarantee the inclusion of an optimal solution in greedy mode. With a non-decreasing skip score, once a task's score is lower than the skip score, it may never be selected. Consequently, such a task will only be selected when no other tasks are running and the skip action is masked, leading to low resource utilization.
>    Consider the simple example in Figure 5 of our main paper in the greedy mode where the scheduler always selects the task or skip action with the highest score. Here 8 tasks need to be processed in a resource pool with only one resource type and resource capacity 3. All tasks have processing time 1 except task 2 with 1.1 and task 3 with 1.2, and all tasks have resource demand 1 except task 4 has resource demand 2. In this simple example, the optimal solution is depicted in subfigure (c), also as following:
>    [----1----]--[----4----][----6----]
>    [----2------][----4----][----7----]
>    [-----3-------][----5----][----8----]
>    This optimal solution needs skips "--" between task 1 and task 4 when task 1 finishes ($t=1$). Given that at this time task 5 is the only avaliable task, when skip is not allowed the task 5 will be directly processed and resulting in the sub-optimal schedule in subfigure (b), also depicted as follows:
>    [----2------][----6----][----8----]
>    [----1----][----5----][----4----]
>    [-----3-------]______[----4----][----7----]
>    Thus, for generating the optimal solution, a skip is needed when task 1 finishes ($t = 1$), and at this time the skip score needs to be larger than the score of task 5. For a non-decreasing skip score, the skip score will always be larger than the score of task $5$ when $t\ge 1$. Therefore, task $5$ will be always delayed (since its score is less than skips) until the skip is masked when task 6 and 7 finish (no task is running), leading an even worse schedule as follows:
>    [----1----]--[----4----][----7----][----5----][----8----]
>    [----2------][----4----]
>    [-----3-------][----6----]
>    This illustration shows that, for a non-decreasing skip score, the scheduler still cannot yield an optimal solution in such a simple example. Therefore, the skip score should be a decreasing function.
>    2. The $(1-\frac{k}{2n})$ serves as normalization for $k$. As established in Theorem 1, the algorithm is guaranteed to terminate within $2n$ steps, ensuring $k \le 2n$. Consequently, this term decreases from 1 to 0. Our results in Table R11 show that removing this normalization (i.e., replacing $(1-\frac{k}{2n})$ with $-k$, and $u_{skip} = -u_a k + u_c, (u_a\ge 0)$) leads to a performance decrease.
>    3. As suggested by reviewer DcSo, we also evaluated a linear form ($u_a(1-\frac{k}{2n})+u_b$) of the skip scores. The results in Table R11 show the linear form achieves performance comparable to our proposed design.
>
> **Table R11: Ablation Study on Skip Score Formulations.**
> |  | **tpch-30-heavy** | |
> | :--- | :--- | :--- |
> | | **Cost** | **Improvement** |
> | No-skip | 26175 | 0.00% |
> | Original-skip | 24611 | 5.97% |
> | Linear | 24692 | 5.67% |
> | Un-normalized | 24924 | 4.78% |

---

> ### Author Response · Authors · 2025-11-22
>
> ## Response to Weakness 3
> In our ablation study (Table 5 in the main text, also Table R12 as following), we already compare our LDDGNN backbone against two versions of a standard GAT (forward and bidirectional).
>
> **Table R12: Ablation Study on TPC-H.**
> |  | **tpch-30** | | **tpch-50** | |
> | :--- | :--- | :--- | :--- | :--- |
> | | **Cost** | **Improvement**| **Cost** | **Improvement** |
> | Tetris | 23170 | 0.0% |  38654 | 0.0% |
> | WeCA + LDDGNN| 19908 $\pm$ 48| 14.0 $\pm$ 0.2% |34260 $\pm$ 52 |11.4 $\pm$ 0.1 %|
> | WeCA + GAT(forward) |20747 $\pm$ 21 |10.5 $\pm$ 0.1 % |35224 $\pm$ 14 |8.9 $\pm$ 0.1%|
> | WeCA + GAT(bi-direction) |20873 $\pm$ 7 |9.9 $\pm$ 0.0 % |35177 $\pm$ 20 |9.0 $\pm$ 0.1%|
>
> ## Response to Weakness 4
>  Although we have not found an existing dataset including many heavy tasks, this scenario is prevalent in the real world. Cloud computation platforms often provide heterogeneous resources for users. While most users submit common computation tasks, a portion of users train Large Language Models, requiring long training times and significant resource demands. Consequently, heterogeneous DAG scheduling problems involving heavy tasks are likely widely encountered in the real world. We thank the reviewer for this insightful suggestion and for highlighting the importance of Figure 8. We agree that Figure 8 would be more impactful in the main text, and we will move this figure from the appendix to the main body of the paper, replacing the original Figure 3.

---

> > ### Comment · Reviewer_DcSo · 2025-11-26
> >
> > Thank you for the further clarification. I will remain with this score.

---

### Official Review · Reviewer_dXJx · 2025-10-31

**Soundness:** 3
**Presentation:** 3
**Contribution:** 2
**Rating:** 4
**Confidence:** 4

**Summary:**

The paper presents a Scheduling method for Directed Acyclic Graphs in heterogenous environments. The heterogeneity is presenting in compatibility coefficients and resource pools. The presented method WeCan, leverages Weighted Cross Attention Based Graph Encoding, and is shown to outperform the baselines in given benchmarks.

**Strengths:**

- Claims are supported by extensive empirical analysis.

- Outperforms existing benchmarks.

**Weaknesses:**

- The need for retraining in different scales leads to question on how well the model can scale up.

- While the empirical results show that the model is able to outperform heuristics, it is unclear how good the performance is with respect of exact solvers such as Gurobi or SCIP.

- It is unclear how different constraints would effect the performance of the model.

**Questions:**

What are the advantages of the presented method compared to Graph Scheduling methods based on Attention [1, 2, 3]?

Skip action in Multi-Agent Task Allocation and Scheduling has been done in sequential MARL based methods [2]. How does the proposed method compare to existing methods that leverage skip action?

How does the proposed method differ from Heterogenous Task Allocation and Scheduling proposed in [3]? What are the advantages and disadvantages of the proposed method.

Most sequential decision making methods represent the problems in a Markovian Decision Process.

[1] Y. Hu, Y. Yao, and W. S. Lee, “A reinforcement learning approach for optimizing multiple traveling salesman problems over graphs,” _Knowledge-Based Systems_, vol. 204, p. 106244, Sept. 2020, doi: [10.1016/j.knosys.2020.106244](https://doi.org/10.1016/j.knosys.2020.106244).

[2] Z. Wang and M. Gombolay, “Stochastic Resource Optimization over Heterogeneous Graph Neural Networks for Failure-Predictive Maintenance Scheduling,” _ICAPS_, vol. 32, pp. 527–536, June 2022, doi: [10.1609/icaps.v32i1.19839](https://doi.org/10.1609/icaps.v32i1.19839).

[3] B. Altundas, Z. Wang, J. Bishop, and M. Gombolay, “Learning Coordination Policies over Heterogeneous Graphs for Human-Robot Teams via Recurrent Neural Schedule Propagation,” in _2022 IEEE/RSJ International Conference on Intelligent Robots and Systems (IROS)_, Kyoto, Japan: IEEE, Oct. 2022, pp. 11679–11686. doi: [10.1109/IROS47612.2022.9981748](https://doi.org/10.1109/IROS47612.2022.9981748).

---

> ### Author Response · Authors · 2025-11-22
>
> We thank the reviewer for the valuable questions. In this response, we provide results demonstrating our model's scalability, explain the infeasibility of using exact solvers, and detail the key differences between our work and the cited papers regarding problem formulation and methodology.
>
> ## Response to Weakness 1
>    Our network is scalable and does not require retraining for different problem scales. We conducted evaluations involving training on different sizes solely to ensure a fair comparison with previous baselines. As the results show, a model trained on one size generalizes well to larger scales.
>
>    **Table R8: Generalization for different problem scales.**
>    | Evaluation cases | tpch-30-model | tpch-50-model | tpch-100-model |
>    | :--- | :---: | :---: | :---: |
>    | tpch-30  | 18960 | 18878 | 19138 |
>    | tpch-50  | 33137 | 32885 | 33280 |
>    | tpch-100 | 61812 | 61014 | 61364 |
>
>    Notably, these results show that when evaluated on the large scale (tpch-100), the model trained on a smaller size (tpch-50) even achieves a better performance.
>
> ## Response to Weakness 2
>    The reason we do not directly use the exact solvers is that, as pointed out in Appendix D, the corresponding MILP formulation involves approximately $10^6$ to $10^8$ variables and constraints.
>
>    **Table R9: Problem size of the MILP formulation.**
>    |  | **MILP scales** | | |
>    | :--- | :--- | :--- | :--- |
>    | | **Variables** | **Constraints** | **Nonzeros** |
>    | Computation-Graph, 3 pool | $1.5\times10^6$ | $3\times10^6$ | $1.2\times10^7$ |
>    | TPC-H-100-3 pools | $6\times10^6$ | $1.2\times10^7$ | $4.8\times10^7$ |
>    | TPC-H-200-3 pools | $2.4\times10^6$ | $4.8\times10^7$ | $2\times10^8$ |
>
>    Therefore, exact solvers like Gurobi cannot find even a feasible solution within the given time limit (3600s).
>
> ## Response to Weakness 3
>    Our generalization results validate the effectiveness of our model under different constraints. The generalization results in Figure 2 and Appendix F.3 show that our model maintains robust performance across different environmental changes (more pools, more pool types, more task types and different pool capacities). These changes in settings alter both the numerical values within these constraints and the total number of constraints.  Our experiments show our approach has robust performance under different constraints.

---

> ### Author Response · Authors · 2025-11-22
>
> ## Response to Question 1 and 3
>    Previous works [1] and [2] focus on different combinatorial optimization problems which do not include settings such as compatibility between two sets of objects (tasks and pools). for those problems, standard cross-attention is a powerful architecture for embedding relational information. Our approach focuses on the Heterogeneous DAG scheduling problem and provides an effective and adaptive way of embedding compatibility into our weighted cross-attention architecture.
>
>    Although previous HGN-based works such as [3,4] also provide a way for addresing the compatibility, they often treat compatibility (actual running time) as a simple edge attribute. This attribute is typically concatenated with node features for further computation within a graph convolution or attention mechanism, as illustrated by some common formulations:
>    $$ f_i \gets \sum_{e=(i,j) \in E} \text{MLP}([x_i || x_j || e_{ij}]), $$
>    $$ f_i \gets \sum_{e=(i,j) \in E} e_{ij} W_V[x_j||e_{ij}], \quad \text{where } e_{ij} \text{ is part of the attention score } \alpha_{ij}. $$
>    Here, $x_i, x_j$ are node embeddings and $e_{ij}$ is the edge attribute. When compatibility is integrated within the MLP or the attention score calculation, it is typically processed in the same manner as common features. For heterogeneous DAG scheduling, where these coefficients heavily influence the problem structure, a more direct representation of their structural role can be more effective. WeCAN is designed to explicitly preserve this structural importance by leveraging the non-negative compatibility coefficients as a post-softmax re-weighting factor, applying them outside the standard cross attention normalization:
>    $$q_v = W^Q f_v, \quad K^c = W^K [f_{c(1)}^c,...,f_{c(n_c)}^c], \quad V^c = W^V [f_{c(1)}^c,...,f_{c(n_c)}^c],$$
>    $$g_v = f_v + \text{softmax}\left(\frac{q_v^T K^c}{\sqrt{d}}\right) \cdot \text{diag}\{K_{acc}(v,c(1)), ..., K_{acc}(v,c(n_c))\} \cdot V^c.$$
>    This unique design allows a task to directly absorb information from its most compatible pools while naturally suppressing contributions from incompatible ones. As demonstrated in our ablation study, this external re-weighting mechanism is more powerful for capturing the problem's structure than designs that embed compatibility within the softmax normalization.
>    To further validate the effectiveness of WeCAN, we have added two new baselines [3,4] leveraging two types of HGNs for handling compatibility. The results of two HGN-based approaches in Table R10 show that WeCAN significantly outperforms these HGN-based approaches on the heterogeneous DAG scheduling problem. Therefore, WeCAN offers a more effective and interpretable mechanism for representing compatibility, leading to superior performance in the heterogeneous DAG scheduling problem.
>
>    **Table R10: Performance of HGN-based Approach on TPC-H Dataset.**
>    |  | **tpch-30** | | **tpch-50** | | **tpch-100** | |
>    | :--- | :--- | :--- | :--- | :--- | :--- | :--- |
>    | | **Cost** | **Improvement** | **Cost** | **Improvement** | **Cost** | **Improvement** |
>    | HEFT | 23177 | 0.00% | 39315 | 0.00% | 70137 | 0.00% |
>    | HeGAT | 21226 | 8.41% | 36643 | 6.80% | 69342 | 1.13% |
>    | HGN-Two-Stage | 21080 | 9.04% | 37295 | 5.13% | 69818 | 0.45% |
>    | WeCAN-Greedy | 19578 | 15.53% | 33428 | 14.97% | 62587 | 10.76% |
>    | WeCAN-S(256) | 18964 | 18.18% | 32814 | 16.54% | 61373 | 12.50% |
>
>    Previous work [3] focuses on a different scheduling domain compared to ours. In the setting of [3], an agent can only perform one task at a time, while in our settings there may be multiple tasks (with different end time) running at the same resource pool. The setting in [3] is closer to the JSSP or CVRP. Moreover, the approach in [3] requires a network forward pass for each decision step, while our approach requires only a single forward pass to solve an entire problem instance.

---

> ### Author Response · Authors · 2025-11-22
>
> ## Response to Question 2
>    Previous work [2] focuses on a different field of scheduling compared to ours. In the setting of [2], tasks represent maintenance for airplanes. These tasks are not mandatory (i.e. if the maintenance cost is higher than the hourly income) or can be performed several times. In contrast, the tasks in our models are the computation tasks that must be executed exactly once. Therefore, the skip in the setting of [2] is more like giving up a current opportunity, whereas a skip in our setting is more like delaying a task. Furthermore, the time is discrete in the setting of [2] and a skip means jumping from time $t$ to $t+1$ to obtain new observations. Conversely, the time is continuous in our setting and the skip means advancing to the next time instance when some task finishes. Consequently, the definition and semantics of the skip differ significantly between our setting and that of [2].
>    Moreover, we have discussed our contributions regarding the skip action compared to previous work in the scheduling field in Appendix C.

---

> ### Author Response · Authors · 2025-11-22
>
> ## References
> [1] Y. Hu, Y. Yao, and W. S. Lee, "A reinforcement learning approach for optimizing multiple traveling salesman problems over graphs," Knowledge-Based Systems, vol. 204, p. 106244, Sept. 2020, doi: 10.1016/j.knosys.2020.106244.
>
> [2] Z. Wang and M. Gombolay, "Stochastic Resource Optimization over Heterogeneous Graph Neural Networks for Failure-Predictive Maintenance Scheduling," ICAPS, vol. 32, pp. 527-536, June 2022, doi: 10.1609/icaps.v32i1.19839.
>
> [3] B. Altundas, Z. Wang, J. Bishop and M. Gombolay, "Learning Coordination Policies over Heterogeneous Graphs for Human-Robot Teams via Recurrent Neural Schedule Propagation," 2022 IEEE/RSJ International Conference on Intelligent Robots and Systems (IROS), Kyoto, Japan, 2022, pp. 11679-11686, doi: 10.1109/IROS47612.2022.9981748.
>
> [4] W. Song, X. Chen, Q. Li and Z. Cao, "Flexible Job-Shop Scheduling via Graph Neural Network and Deep Reinforcement Learning," in IEEE Transactions on Industrial Informatics, vol. 19, no. 2, pp. 1600-1610, Feb. 2023, doi: 10.1109/TII.2022.3189725.

---

### Official Review · Reviewer_PDB6 · 2025-11-01

**Soundness:** 2
**Presentation:** 2
**Contribution:** 1
**Rating:** 2
**Confidence:** 4

**Summary:**

This paper proposes **WeCAN**, a reinforcement learning framework for *one-shot DAG scheduling* in heterogeneous environments. The method uses a single forward pass of a neural network to generate task-pool priority scores, which are then converted into schedules using heuristic-based generation. The claimed key components include weighted cross-attention layers to incorporate compatibility coefficients $K_{acc}$ and a longest directed distance graph neural network (LDDGNN) for structural encoding.

While the paper claims contributions in network architecture and skip-action design, the actual novelty appears limited compared to existing one-shot scheduling approaches like [1] Jeon et al. (2023), and the experimental evaluation lacks breadth in datasets and baselines. I would rate this paper **3/10** for weak novelty and limited evaluation (temporarily rating as “2” since 3 is not selectable).

----
[1] Jeon et al., *Neural DAG Scheduling via One-Shot Priority Sampling*, ICLR 2023.

**Strengths:**

* Provides a scheduling framework similar to HEFT, but implemented with a learned model.
* The integration of skip actions in a single-pass inference setting shows performance gains in many scenarios.
* The paper clearly formulates the heterogeneous DAG scheduling problem with compatibility constraints and provides a detailed MILP formulation.

**Weaknesses:**

* **Limited novelty**: The core approach of one-shot priority generation closely follows [1] Jeon et al. (2023), with incremental improvements mainly in the attention mechanism and skip action design. The weighted cross-attention essentially amplifies compatibility coefficients in softmax, while LDDGNN adds a distance-based bias to GNN computations. The term “featuring task–resource compatibility” is highlighted as a key property, but most recent scheduling works already include this.
* **Insufficient theoretical contribution**: Section 4 provides a formalization of the optimality gap in list scheduling but does not offer new theoretical insights or convergence guarantees for RL-based scheduling. The analysis feels more like a formalization of existing intuition rather than a substantive theoretical advancement.
* **Incomplete experimental evaluation**:
  - Missing comparison with important benchmarks like Pegasus workflow traces and Flexible Job Shop Scheduling (FJSP) instances, which are standard in scheduling literature.
  - Baseline selection omits recent heuristic methods (e.g., PEFT, IPPTS) and learning-based schedulers (e.g., L2D), limiting the persuasiveness of performance claims.
* **Presentations**:
  - Input features for task and pool embeddings are not thoroughly described.
  - Figures (especially Fig. 1) are unclear, and several key definitions (e.g., generation map, skip score design, input features) are not well explained.
  - The claimed synthetic dataset reference ([1] Jeon et al., 2023) is inaccurate, as that dataset is not publicly available.

**Questions:**

1. The Introduction cites older works under “Neural DAG Schedulers for Heterogeneous Environments.” Please include more recent literature that addresses adaptability in dynamic heterogeneous settings (e.g., recent papers from top AI/Systems venues).
1. Given that the scheduling problem is formulated as a MILP, why not compare with off-the-shelf solvers (e.g., Gurobi, CPLEX) on smaller instances to benchmark optimality gaps?
1. The skip action score is defined as $u_{\pi_{skip}} = u_a(1 - \frac{k}{2n})^{u_b} + u_c$. What is the intuition behind this specific form? How does it prevent over-prioritization of skip actions, and were alternative formulations explored?
1. Figure 1 is hard to interpret — please clarify the full pipeline and the meaning of the “generation map.”
1. What raw features are used to construct the task and pool embeddings?
1. The paper states that the “Computation Graphs” dataset was generated following Jeon et al. (2023), but that dataset was not released; please clarify your data generation process.
1. This work appears as an incremental extension of [1] Jeon et al. (2023), which evaluated on broader domains (JSP, TPC-H, computation graphs). Please evaluate the proposed method on the Pegasus datasets and the FJSP instances.
1. The baselines are incomplete — recent heuristic methods such as PEFT and IPPTS should be compared, along with neural methods (e.g., L2D, other workflow NCO models) to highlight performance and inference-time trade-offs.

---

> ### Author Response · Authors · 2025-11-22
>
> We thank the reviewer for the feedback. To address the concerns raised about our experimental evaluation, we have now benchmarked WeCAN against several new heuristic and HGN-based baselines. We also provide detailed responses to clarify the theoretical novelty of our work, the motivation for our design choices, and the key differences between our approach and prior methods.
>
> ## Response to Weakness 1 \& Question 7 (Part A)
>    Our approach differs from the Jeon et al. (2023) [1] in several key aspects.
>    - Our probability model is fundamentally different from that of Jeon et al. (2023) [1]. Our approach originates from our auto-regressive (AR) model (Appendix B), whose core constraint-enforcing mechanism (i.e. action masking mentioned in Section 2.2 of the main text) is adapted for our main single-pass, non-auto-regressive (NAR) framework. Specifically, our model's probability calculation is defined over a space of valid action sequences. By applying feasibility masks during generation, we ensure that only sequences adhering to our three types of constraints are produced. Each output from our model is therefore a valid sequence, and the model computes its probability directly. In contrast, Jeon et al.'s one-shot approach defines probabilities over a much larger space of raw generation orders, which permits arbitrary permutations of tasks. This means that to obtain the real probability of a single feasible schedule, their model would need to account for multiple distinct generation orders that are mapped by list scheduling to that same schedule (corresponding to a valid sequence). Consequently, the probability mass for what is a single target sequence in our model is "split" across several source sequences in theirs. Therefore, the very set of outcomes for which the probabilities are computed is different, leading to a distinct probability formulation.

---

> > ### Author Response · Authors · 2025-11-22
> >
> > ## Response to Weakness 4 \& Questions 4, 5, 6
> >    - The two main types of input features are shown in lines 186-187 in the main text. The initial task feature is $(t(v),\rho(v))$ where $t$ is the average processing time and $\rho$ is the resource demand vector. The initial pool feature is $(\lambda)$ where $\lambda$ is the resource capacity vector. We will revise the manuscript to present the description of input features in a dedicated paragraph, and add these detail description in the Appendix for better clarity.
> >    - The generation map is a map from the action space to the origin space (feasible schedules) and differs in different approaches. The widely used list scheduling map is a special type of generation map. The projection map $S_n$ described in the Appendix, and our generation map $S$, as specified in Algorithm 1, are also examples of generation maps. The explanation of the skip action design and its intuition is provided in our response to Question 3. We will revise the manuscript to clarify the description of the generation map, provide more intuition for the skip scores, and refine our main figures accordingly.
> >    - The reviewer is correct that the dataset from Jeon et al. (2023) [1] is not publicly available. To clarify, our statement "the dataset was generated following Jeon et al. (2023)" means that we implemented the data generation process based on the pseudo-code provided in a paper [8] that Jeon et al. (2023) [1] also referenced. The parameter settings are described in Appendix D of our paper. We will adjust the description to "we follow the pseudo-code of Gagrani et al. (2022) and Jeon et al. (2023) to generate the dataset". The details of the pseudo-code we follow are provided in Appendix A of the paper [8] referred to by Jeon et al. (2023). We implemented the data generator ourselves based on their pseudo-code. We plan to open-source our data generation code upon publication of the paper.
> >
> >    We thank the reviewer for the helpful suggestions and will improve our writing for greater clarity.
> >
> > ## Response to Question 1
> >    We will update our related work section to include and discuss additional papers that handle compatibility, such as [2,3], and clarify their relation to our work.
> >
> > ## Response to Question 2
> >    The reason we do not directly use the exact solvers is that, as pointed out in Appendix D, the corresponding MILP formulation involves approximately $10^6$ to $10^8$ variables and constraints.
> >
> >    **Table R6: Problem size of the MILP formulation.**
> >    |  | **MILP scales** | | |
> >    | :--- | :--- | :--- | :--- |
> >    | | **Variables** | **Constraints** | **Nonzeros** |
> >    | Computation-Graph, 3 pool | $1.5\times10^6$ | $3\times10^6$ | $1.2\times10^7$ |
> >    | TPC-H-100-3 pools | $6\times10^6$ | $1.2\times10^7$ | $4.8\times10^7$ |
> >    | TPC-H-200-3 pools | $2.4\times10^6$ | $4.8\times10^7$ | $2\times10^8$ |
> >
> >    Therefore, exact solvers like Gurobi cannot find even a feasible solution within the given time limit (3600s). For smaller problem instances that can be solved by Gurobi, the resources in our experimental setting become abundant, causing the problem to reduce to a degenerate case where scheduling is trivial.

---

> ### Author Response · Authors · 2025-11-22
>
> ## Response to Weakness 1 \& Question 7 (Part B)
>
>    - WeCAN introduces a new approach for handling the compatibility coefficients.
>       - As discussed in our related work section, prior methods addressing compatibility in dynamic heterogeneous settings usually fall into two categories: either averaging compatibility information, which leads to information loss, or embedding the full compatibility matrix into task features. The latter approach binds the network's input dimension to the number of resource pools, thus losing adaptability to new environments (e.g., an increased number of pools).
>
>       - While heterogeneous graph networks (HGNs) can leverage full compatibility information while retaining adaptability, previous HGN-based works often treat compatibility (actual running time) as a simple edge attribute. This attribute is typically concatenated with node features for further computation within a graph convolution or attention mechanism, as illustrated by some common formulations:
>       $$ f_i \gets \sum_{e=(i,j) \in E} \text{MLP}([x_i || x_j || e_{ij}]), $$
>       $$ f_i \gets \sum_{e=(i,j) \in E} e_{ij} W_V[x_j||e_{ij}], \quad \text{where } e_{ij} \text{ is part of the attention score } \alpha_{ij}. $$
>        Here, $x_i, x_j$ are node embeddings and $e_{ij}$ is the edge attribute. When compatibility is integrated within the MLP or the attention score calculation, it is typically processed in the same manner as common features. For heterogeneous DAG scheduling, where these coefficients heavily influence the problem structure, a more direct representation of their structural role can be more effective. WeCAN is designed to explicitly preserve this structural importance by leveraging the non-negative compatibility coefficients as a post-softmax re-weighting factor, applying them outside the standard cross attention normalization:
>        $$q_v = W^Q f_v, \quad K^c = W^K [f_{c(1)}^c,...,f_{c(n_c)}^c], \quad V^c = W^V [f_{c(1)}^c,...,f_{c(n_c)}^c],$$
>        $$g_v = f_v + \text{softmax}\left(\frac{q_v^T K^c}{\sqrt{d}}\right) \cdot \text{diag}\{K_{acc}(v,c(1)), ..., K_{acc}(v,c(n_c))\} \cdot V^c.$$
>        This unique design allows a task to directly absorb information from its most compatible pools while naturally suppressing contributions from incompatible ones. As demonstrated in our ablation study, this external re-weighting mechanism is more powerful for capturing the problem's structure than designs that embed compatibility within the softmax normalization.
>        To further validate the effectiveness of WeCAN, we have added two new baselines [2,3] leveraging two types of HGNs for handling compatibility. The results of two HGN-based approaches in Table R4 show that WeCAN significantly outperforms these HGN-based approaches on the heterogeneous DAG scheduling problem. Therefore, WeCAN offers a more effective and interpretable mechanism for representing compatibility, leading to superior performance in the heterogeneous DAG scheduling problem.
>
>    - WeCAN introduces powerful tools to reveal the existing optimality gap and provides theoretical insights for reducing the optimality gap. Further details are provided in our response to Weakness 2.
>
> **Table R4: Performance of HGN-based Approach on TPC-H Dataset.**
> |  | **tpch-30** | | **tpch-50** | | **tpch-100** | |
> | :--- | :--- | :--- | :--- | :--- | :--- | :--- |
> | | **Cost** | **Improvement** | **Cost** | **Improvement** | **Cost** | **Improvement** |
> | HEFT | 23177 | 0.00% | 39315 | 0.00% | 70137 | 0.00% |
> | HeGAT | 21226 | 8.41% | 36643 | 6.80% | 69342 | 1.13% |
> | HGN-Two-Stage | 21080 | 9.04% | 37295 | 5.13% | 69818 | 0.45% |
> | WeCAN-Greedy | 19578 | 15.53% | 33428 | 14.97% | 62587 | 10.76% |
> | WeCAN-S(256) | 18964 | 18.18% | 32814 | 16.54% | 61373 | 12.50% |

---

> ### Author Response · Authors · 2025-11-22
>
> ## Response to Weakness 2
>    WeCAN introduces a powerful framework that not only reveals existing optimality gaps in prior work but also offers theoretical insights on how to reduce them. In Section 4, we establish a formal framework for analyzing the action spaces relevant to the DAG scheduling problem. This framework reveals that the action spaces constructed by many previous works (e.g., [1,4,5]) inherently exclude optimal solutions, leading to an inherent optimality gap. And our framework points out when this gap will lead to a significant performance loss (the cases with heavy tasks). This is a critical insight that has not been formally addressed in most prior work. We provide a framework for revealing the key reasons why the previous action spaces exclude the optimal solutions: the combination of the generation map and the natural map does not form a surjection into the feasible reduced space. This analytical result, in turn, provides clear guidance on how to bridge this gap, motivating methods such as skip actions and local search with projection maps. The core principle for closing this gap, as identified by our framework, is to restore surjectivity. Guided by this theory, we can distinguish between sufficient approaches (like our skip-design or a time-consuming local search) and insufficient ones (like a skip mechanism with a constant score). In this way, our theoretical results provide valuable insights that directly guide our algorithmic design.
>
> ## Response to Weakness 3 \& Question 8
>    - We have already included comprehensive experimental evaluations: evaluations on TPC-H and Computational Graph, evaluations on larger problem scales, varying types of environment changes and different rates of heavy tasks, ablation of architecture and skip actions, and evaluations on local search. Our approach focuses on the general setting of Heterogeneous DAG scheduling: dependencies are nonlinear, each task has several types of resource demands and each pool can have several tasks running in parallel. FJSP represents a highly simplified instance of our problem. In FJSP, where task dependencies are typically linear, a machine is a unary resource, meaning it processes only one task at a time with exclusive access. This effectively homogenizes resource demand, as each task requires the full capacity of a single machine, rendering concepts like "heavy tasks" irrelevant. Consequently, scheduling in FJSP reduces to a matter of temporal sequencing, bypassing the multi-dimensional resource capacity checks present in our approach. Therefore, directly applying our model, which is architected for heterogeneous DAG scheduling, would be inappropriate as its mechanisms for resource management would become redundant. For Pegasus, we have not found a clean and open-source dataset online. The open-source workflow trace contains numerous redundant files and we are in the process of sorting a clean dataset from the workflow traces.
>    - We have added PEFT and IPPTS as baselines [6,7]. The results in Table R5 show that PEFT achieves a similar performance to HEFT, and IPPTS achieves better performance than both. Our approach, however, significantly outperforms both of these new heuristic baselines. Note that compared to heuristics like CP and HEFT, PEFT and IPPTS rely on a specific assumption that all tasks can start on any pool. Thus, although they can be applied to the current instances, they can not be directly applied in some specific instances which our general problem setting covers (or some problems in JSSP).
>    **Table R5: Performance of Heuristic on TPC-H Dataset.**
>       |  | **tpch-30** | | **tpch-50** | | **tpch-100** | |
>       | :--- | :--- | :--- | :--- | :--- | :--- | :--- |
>       | | **Cost** | **Improvement** | **Cost** | **Improvement** | **Cost** | **Improvement** |
>       | HEFT | 23177 | 0.00% | 39315 | 0.00% | 70137 | 0.00% |
>       | PEFT | 22930 | 1.07% | 39881 | -1.44% | 69832 | 0.44% |
>       | IPPTS | 21389 | 7.71% | 37163 | 5.47% | 68386 | 2.50% |
>       | WeCAN-Greedy | 19578 | 15.53% | 33428 | 14.97% | 62587 | 10.76% |
>       | WeCAN-S(256) | 18964 | 18.18% | 32814 | 16.54% | 61373 | 12.50% |
>    - L2D is a constructive approach suitable for JSSP but may not be suitable for our general setting. Instead, we have implemented two more recent Heterogeneous-Graph-Based scheduling approaches [2,3] adapted to our setting. Details are provided in our response to Weakness 1 & Question 7.

---

> ### Author Response · Authors · 2025-11-22
>
> ## Response to Question 3
>    While our method of introducing skip actions into the single-pass framework is theoretically validated to ensure optimal solutions can be found, we acknowledge that our specific scoring function may not be the only empirically effective design. We present two key intuitions behind our proposed skip score formulation:
>    1. Our skip score formulation, $u_a(1-\frac{k}{2n})^{u_b} + u_c$ (where $u_a,u_b\ge 0$),  is deliberately designed to be a decreasing function of $k$. This is because according to our theoretical results, a non-decreasing term cannot restore the surjectivity and guarantee the inclusion of an optimal solution in greedy mode. With a non-decreasing skip score, once a task's score is lower than the skip score, it may never be selected. Consequently, such a task will only be selected when no other tasks are running and the skip action is masked, leading to low resource utilization.
>    Consider the simple example in Figure 5 of our main paper in the greedy mode where the scheduler always selects the task or skip action with the highest score. Here 8 tasks need to be processed in a resource pool with only one resource type and resource capacity 3. All tasks have processing time 1 except task 2 with 1.1 and task 3 with 1.2, and all tasks have resource demand 1 except task 4 has resource demand 2. In this simple example, the optimal solution is depicted in subfigure (c), also as following:
>    [----1----]--[----4----][----6----]
>    [----2------][----4----][----7----]
>    [-----3-------][----5----][----8----]
>    This optimal solution needs skips "--" between task 1 and task 4 when task 1 finishes ($t=1$). Given that at this time task 5 is the only avaliable task, when skip is not allowed the task 5 will be directly processed and resulting in the sub-optimal schedule in subfigure (b), also depicted as follows:
>    [----2------][----6----][----8----]
>    [----1----][----5----][----4----]
>    [-----3-------]______[----4----][----7----]
>    Thus, for generating the optimal solution, a skip is needed when task 1 finishes ($t = 1$), and at this time the skip score needs to be larger than the score of task 5. For a non-decreasing skip score, the skip score will always be larger than the score of task $5$ when $t\ge 1$. Therefore, task $5$ will be always delayed (since its score is less than skips) until the skip is masked when task 6 and 7 finish (no task is running), leading an even worse schedule as follows:
>    [----1----]--[----4----][----7----][----5----][----8----]
>    [----2------][----4----]
>    [-----3-------][----6----]
>    This illustration shows that, for a non-decreasing skip score, the scheduler still cannot yield an optimal solution in such a simple example. Therefore, the skip score should be a decreasing function.
>    2. The $(1-\frac{k}{2n})$ serves as normalization for $k$. As established in Theorem 1, the algorithm is guaranteed to terminate within $2n$ steps, ensuring $k \le 2n$. Consequently, this term decreases from 1 to 0. Our results in Table R7 show that removing this normalization (i.e., replacing $(1-\frac{k}{2n})$ with $-k$, and $u_{skip} = -u_a k + u_c, (u_a\ge 0)$) leads to a performance decrease.
>    3. As suggested by reviewer DcSo, we also evaluated a linear form ($u_a(1-\frac{k}{2n})+u_b$) of the skip scores. The results in Table R7 show the linear form achieves performance comparable to our proposed design.
>
> **Table R7: Ablation Study on Skip Score Formulations.**
> |  | **tpch-30-heavy** | |
> | :--- | :--- | :--- |
> | | **Cost** | **Improvement** |
> | No-skip | 26175 | 0.00% |
> | Original-skip | 24611 | 5.97% |
> | Linear | 24692 | 5.67% |
> | Un-normalized | 24924 | 4.78% |

---

> ### Author Response · Authors · 2025-11-22
>
> ## References
> [1] W. Jeon, M. Gagrani, B. Bartan, W. W. Zeng, H. Teague, P. Zappi and C. Lott, "Neural DAG Scheduling via One-Shot Priority Sampling," The Eleventh International Conference on Learning Representations, 2023.
>
> [2] B. Altundas, Z. Wang, J. Bishop and M. Gombolay, "Learning Coordination Policies over Heterogeneous Graphs for Human-Robot Teams via Recurrent Neural Schedule Propagation," 2022 IEEE/RSJ International Conference on Intelligent Robots and Systems (IROS), Kyoto, Japan, 2022, pp. 11679-11686, doi: 10.1109/IROS47612.2022.9981748.
>
> [3] W. Song, X. Chen, Q. Li and Z. Cao, "Flexible Job-Shop Scheduling via Graph Neural Network and Deep Reinforcement Learning," in IEEE Transactions on Industrial Informatics, vol. 19, no. 2, pp. 1600-1610, Feb. 2023, doi: 10.1109/TII.2022.3189725.
>
> [4] Y. Zhou, X. Li, J. Luo, M. Yuan, J. Zeng and J. Yao, "Learning to Optimize DAG Scheduling in Heterogeneous Environment," 2022 23rd IEEE International Conference on Mobile Data Management (MDM), Paphos, Cyprus, 2022, pp. 137-146, doi: 10.1109/MDM55031.2022.00040.
>
> [5] Z. Wang, W. Zhan, H. Duan, G. Min and H. Huang, "Deep-Reinforcement-Learning-Based Continuous Workflows Scheduling in Heterogeneous Environments," in IEEE Internet of Things Journal, vol. 12, no. 10, pp. 14036-14050, 15 May15, 2025, doi: 10.1109/JIOT.2024.3524506.
>
> [6] H. Arabnejad and J. G. Barbosa, "List Scheduling Algorithm for Heterogeneous Systems by an Optimistic Cost Table," in IEEE Transactions on Parallel and Distributed Systems, vol. 25, no. 3, pp. 682-694, March 2014, doi: 10.1109/TPDS.2013.57.
>
> [7] H. Djigal, J. Feng, J. Lu and J. Ge, "IPPTS: An Efficient Algorithm for Scientific Workflow Scheduling in Heterogeneous Computing Systems," in IEEE Transactions on Parallel and Distributed Systems, vol. 32, no. 5, pp. 1057-1071, 1 May 2021, doi: 10.1109/TPDS.2020.3041829.
>
> [8] M. Gagrani, C. Rainone, Y. Yang, H. Teague, W. Jeon, R. Bondesan, H. van Hoof, C. Lott, W. Zeng and P. Zappi, "Neural Topological Ordering for Computation Graphs," Advances in Neural Information Processing Systems, vol. 35, pp. 17327-17339, 2022.

---

### Official Review · Reviewer_C3Wq · 2025-11-01

**Soundness:** 3
**Presentation:** 2
**Contribution:** 2
**Rating:** 4
**Confidence:** 3

**Summary:**

This paper presents WeCAN, an RL-based framework for heterogeneous DAG scheduling that integrates a weighted cross-attention mechanism and skip actions to model task–resource compatibility and generate optimized schedules in a single inference pass.

**Strengths:**

- The paper clearly defines the problem and formulates how cross-attention and GNN are used to improve scheduling.

- The proposed WeCAN maintains high performance across environments (varying pools, tasks, and resource types), with robust generality. And the single pass design enables fast, one-shot scheduling with minimal inference time.

**Weaknesses:**

1. The experimental RL-based baselines (HEFT, PPO-BiHyb, One-Shot) do not include more recent RL–based schedulers using advanced graph or attention mechanisms. This makes it difficult to assess WeCAN’s relative improvement over SOTA.

2. The integration of weighted cross-attention and skip actions is conceptually sound but not fundamentally novel. Both have been explored in prior works on RL-based resource allocation, so WeCAN’s contribution is primarily architectural refinement for heterogeneous environments rather than theoretical innovation.

3. The reward function focuses solely on makespan reduction, with little quantitative analysis of how skip frequency or policy variance affects convergence and stability during training.

4. How is it theoretically or empirically validated in WeCAN that advancing the physical time immediately after a skip action leads to optimal or near-optimal scheduling decisions, rather than introducing suboptimal temporal transitions?

5. The paper employs its LDDGNN but does not compare it with modern GNN variants such as Graph Transformer [1] or other graph architectures [2, 3]. Such comparisons are necessary to clarify the actual advantage of LDDGNN over existing methods beyond GAT (2018).

6. The proposed weighted cross-attention module for heterogeneous DAG scheduling appears similar to recent heterogeneous graph neural network designs [4, 5]. Including these models as baselines would strengthen the empirical justification and clarify the novelty of WeCAN’s architecture.

7. The definition of the skip-action function $u_{\pi_{skip}}$ is not well motivated. The paper provides limited analysis of alternative formulations or sensitivity to its hyperparameters, making it unclear whether the chosen form is empirically optimal or arbitrarily selected.

[1] Yun, Seongjun, et al. "Graph transformer networks." Advances in neural information processing systems 32 (2019).

[2] Corso, Gabriele, et al. "Principal neighbourhood aggregation for graph nets." Advances in neural information processing systems 33 (2020): 13260-13271.

[3] Wu, Qitian, et al. "Nodeformer: A scalable graph structure learning transformer for node classification." Advances in Neural Information Processing Systems 35 (2022): 27387-27401.

[4] Hu, Ziniu, et al. "Heterogeneous graph transformer." Proceedings of the web conference 2020. 2020.

[5] Jin, Yufei, et al. "HGDL: Heterogeneous Graph Label Distribution Learning." Advances in Neural Information Processing Systems 37 (2024): 40792-40830.

**Questions:**

See weaknesses.

---

> ### Author Response · Authors · 2025-11-22
>
> We thank the reviewer for the constructive comments. In this response, we present new experiments with more recent baselines and new ablations on our skip-action design. We will also revise the paper to better highlight our theoretical contributions and the novelty of our proposed methods in the final manuscript.
>
> ## Response to Weakness 1
>    We have added two new baselines [1,2] leveraging two advanced heterogeneous graph networks (HGNs) for handling compatibility in the scheduling field. In the field of scheduling, previous HGN-based works often treat compatibility (actual running time) as a simple edge attribute. This attribute is typically concatenated with node features for further computation within a graph convolution or attention mechanism, as illustrated by some common formulations:
>    $$ f_i \gets \sum_{e=(i,j) \in E} \text{MLP}([x_i || x_j || e_{ij}]), $$
>    $$ f_i \gets \sum_{e=(i,j) \in E} e_{ij} W_V[x_j||e_{ij}], \quad \text{where } e_{ij} \text{ is part of the attention score } \alpha_{ij}. $$
>    Here, $x_i, x_j$ are node embeddings and $e_{ij}$ is the edge attribute. When compatibility is integrated within the MLP or the attention score calculation, it is typically processed in the same manner as common features. For heterogeneous DAG scheduling, where these coefficients heavily influence the problem structure, a more direct representation of their structural role can be more effective. WeCAN is designed to explicitly preserve this structural importance by leveraging the non-negative compatibility coefficients as a post-softmax re-weighting factor, applying them outside the standard cross attention normalization:
>    $$q_v = W^Q f_v, \quad K^c = W^K [f_{c(1)}^c,...,f_{c(n_c)}^c], \quad V^c = W^V [f_{c(1)}^c,...,f_{c(n_c)}^c],$$
>    $$g_v = f_v + \text{softmax}\left(\frac{q_v^T K^c}{\sqrt{d}}\right) \cdot \text{diag}\{K_{acc}(v,c(1)), ..., K_{acc}(v,c(n_c))\} \cdot V^c.$$
>    This unique design allows a task to directly absorb information from its most compatible pools while naturally suppressing contributions from incompatible ones. As demonstrated in our ablation study, this external re-weighting mechanism is more powerful for capturing the problem's structure than designs that embed compatibility within the softmax normalization. The additional results of two HGN-based approaches in Table R1 show that WeCAN also significantly outperforms these HGN-based approaches on the heterogeneous DAG scheduling problem. Therefore, WeCAN offers a more effective and interpretable mechanism for representing compatibility, leading to superior performance in the heterogeneous DAG scheduling problem.
>
>    **Table R1: Performance of HGN-based Approach on TPC-H Dataset.**
>    |  | **tpch-30** | | **tpch-50** | | **tpch-100** | |
>    | :--- | :--- | :--- | :--- | :--- | :--- | :--- |
>    | | **Cost** | **Improvement** | **Cost** | **Improvement** | **Cost** | **Improvement** |
>    | HEFT | 23177 | 0.00% | 39315 | 0.00% | 70137 | 0.00% |
>    | HeGAT | 21226 | 8.41% | 36643 | 6.80% | 69342 | 1.13% |
>    | HGN-Two-Stage | 21080 | 9.04% | 37295 | 5.13% | 69818 | 0.45% |
>    | WeCAN-Greedy | 19578 | 15.53% | 33428 | 14.97% | 62587 | 10.76% |
>    | WeCAN-S(256) | 18964 | 18.18% | 32814 | 16.54% | 61373 | 12.50% |

---

> ### Author Response · Authors · 2025-11-22
>
> ## Response to Weakness 2
>    Our contributions extend significantly beyond architectural refinement to include substantial theoretical innovations. WeCAN introduces substantial theoretical innovations that fundamentally advance the field:
>
>    1. Novel Theoretical Framework for Optimality Gap Analysis: We provide the first systematic framework for analyzing action space optimality gaps in DAG scheduling (Section 4). This framework reveals why many existing approaches (such as [3,4,5]) inherently exclude optimal solutions, which is a theoretical insight overlooked by prior works. Our mathematical analysis proves that these previous action spaces fail to form surjections into feasible reduced space, creating unavoidable optimality gaps. Specifically, we show that the combination of generation maps and natural maps in these existing approaches cannot always cover optimal scheduling orders, and more generally, cannot cover all feasible scheduling orders, leading to guaranteed suboptimality in certain cases.
>
>    2. Theoretical Guidance for Gap Reduction: Our framework doesn't just identify problems - it provides theoretical principles for solutions. We prove that fixing surjectivity through skip actions can eliminate optimality gaps, while also characterizing when these gaps cause significant performance loss (specifically in heavy task scenarios where optimal scheduling is most critical).
>    This theoretical foundation guided our algorithm design, distinguishing between sufficient approaches (our skip design) and insufficient ones (constant-score skipping). Our theory provides clear criteria for when and how to apply corrective measures.
>
>    3. Interpretable Cross-Attention Innovation: Our weighted cross-attention mechanism is fundamentally different from prior attention designs, providing the interpretable weighted attention mechanism specifically designed for heterogeneous DAG scheduling.
>
>    In summary, these contributions represent significant theoretical advances, providing both mathematical insights and practical algorithmic guidance that extend well beyond simple architectural refinement.
>
> ## Response to Weakness 3
>    Whether skips will slow down the convergence rate highly relies on the way in which skips are introduced. Introducing skips by the natural projection map (mentioned in Section 4 and Appendix A) will slow down the convergence compared to our proposed skip action design. Table R2 compares the training loss between our skip design and a design based on the natural projection map. The results show that the natural projection map approach converges much more slowly.
>
>    **Table R2: Average training loss for different skip implementation methods.**
>    |  |  **Training loss**  |  |
>    | :--- | :--- | :--- |
>    | **Batch**$\quad\quad\quad\quad$ | **our skip** | **natural projection (arbitrary skip)** |
>    | 0 $\sim$ 50  | 236648 | 261769 |
>    | 50 $\sim$ 100  | 219312 | 246140 |
>    | 100 $\sim$ 150  | 215929 | 243324 |
>    | 150 $\sim$ 200  | 213196 | 241935 |
>    | 200 $\sim$ 250  | 213196 | 241105 |
>    | 250 $\sim$ 300  | 213281 | 241353 |
>    | 300 $\sim$ 350  | 211044 | 239268 |
>    | 350 $\sim$ 400  | 211354 | 240315 |
>
> ## Response to Weakness 4
>    The example in Appendix A.3 illustrates why skip actions are needed to achieve optimality. Specifically, Figure 5(a) presents the problem instance, Figure 5(b) displays the suboptimal schedule obtained without skip actions, and Figure 5(c) shows the optimal solution achieved by allowing a skip. For a more detailed explanation, please see Point 1 in our response to Weakness 7.
>
> ## Response to Weakness 5 & 6
>    Our key contribution lies in the weighted cross-attention layer, not the LDDGNN architecture itself. To better show the performance gains from our weighted cross-attention network, we have introduced two new baselines based on other heterogeneous graph networks. Further details are provided in our response to Weakness 1.

---

> ### Author Response · Authors · 2025-11-22
>
> ## Response to Weakness 7
>   While our method of introducing skip actions into the single-pass framework is theoretically validated to ensure optimal solutions can be found, we acknowledge that our specific scoring function may not be the only empirically effective design. We present two key intuitions behind our proposed skip score formulation:
>    1. Our skip score formulation, $u_a(1-\frac{k}{2n})^{u_b} + u_c$ (where $u_a,u_b\ge 0$),  is deliberately designed to be a decreasing function of $k$. This is because according to our theoretical results, a non-decreasing term cannot restore the surjectivity and guarantee the inclusion of an optimal solution in greedy mode. With a non-decreasing skip score, once a task's score is lower than the skip score, it may never be selected. Consequently, such a task will only be selected when no other tasks are running and the skip action is masked, leading to low resource utilization.
>    Consider the simple example in Figure 5 of our main paper in the greedy mode where the scheduler always selects the task or skip action with the highest score. Here 8 tasks need to be processed in a resource pool with only one resource type and resource capacity 3. All tasks have processing time 1 except task 2 with 1.1 and task 3 with 1.2, and all tasks have resource demand 1 except task 4 has resource demand 2. In this simple example, the optimal solution is depicted in subfigure (c), also as following:
>    [----1----]--[----4----][----6----]
>    [----2------][----4----][----7----]
>    [-----3-------][----5----][----8----]
>    This optimal solution needs skips "--" between task 1 and task 4 when task 1 finishes ($t=1$). Given that at this time task 5 is the only avaliable task, when skip is not allowed the task 5 will be directly processed and resulting in the sub-optimal schedule in subfigure (b), also depicted as follows:
>    [----2------][----6----][----8----]
>    [----1----][----5----][----4----]
>    [-----3-------]______[----4----][----7----]
>    Thus, for generating the optimal solution, a skip is needed when task 1 finishes ($t = 1$), and at this time the skip score needs to be larger than the score of task 5. For a non-decreasing skip score, the skip score will always be larger than the score of task $5$ when $t\ge 1$. Therefore, task $5$ will be always delayed (since its score is less than skips) until the skip is masked when task 6 and 7 finish (no task is running), leading an even worse schedule as follows:
>    [----1----]--[----4----][----7----][----5----][----8----]
>    [----2------][----4----]
>    [-----3-------][----6----]
>    This illustration shows that, for a non-decreasing skip score, the scheduler still cannot yield an optimal solution in such a simple example. Therefore, the skip score should be a decreasing function.
>    2. The $(1-\frac{k}{2n})$ serves as normalization for $k$. As established in Theorem 1, the algorithm is guaranteed to terminate within $2n$ steps, ensuring $k \le 2n$. Consequently, this term decreases from 1 to 0. Our results in Table R3 show that removing this normalization (i.e., replacing $(1-\frac{k}{2n})$ with $-k$, and $u_{skip} = -u_a k + u_c, (u_a\ge 0)$) leads to a performance decrease.
>    3. As suggested by reviewer DcSo, we also evaluated a linear form ($u_a(1-\frac{k}{2n})+u_b$) of the skip scores. The results in Table R3 show the linear form achieves performance comparable to our proposed design.
>
> **Table R3: Ablation Study on Skip Score Formulations.**
> |  | **tpch-30-heavy** | |
> | :--- | :--- | :--- |
> | | **Cost** | **Improvement** |
> | No-skip | 26175 | 0.00% |
> | Original-skip | 24611 | 5.97% |
> | Linear | 24692 | 5.67% |
> | Un-normalized | 24924 | 4.78% |

---

> ### Author Response · Authors · 2025-11-22
>
> ## References
> [1] B. Altundas, Z. Wang, J. Bishop and M. Gombolay, "Learning Coordination Policies over Heterogeneous Graphs for Human-Robot Teams via Recurrent Neural Schedule Propagation," 2022 IEEE/RSJ International Conference on Intelligent Robots and Systems (IROS), Kyoto, Japan, 2022, pp. 11679-11686, doi: 10.1109/IROS47612.2022.9981748.
>
> [2] W. Song, X. Chen, Q. Li and Z. Cao, "Flexible Job-Shop Scheduling via Graph Neural Network and Deep Reinforcement Learning," in IEEE Transactions on Industrial Informatics, vol. 19, no. 2, pp. 1600-1610, Feb. 2023, doi: 10.1109/TII.2022.3189725.
>
> [3] W. Jeon, M. Gagrani, B. Bartan, W. W. Zeng, H. Teague, P. Zappi and C. Lott, "Neural DAG Scheduling via One-Shot Priority Sampling," The Eleventh International Conference on Learning Representations, 2023.
>
> [4] Y. Zhou, X. Li, J. Luo, M. Yuan, J. Zeng and J. Yao, "Learning to Optimize DAG Scheduling in Heterogeneous Environment," 2022 23rd IEEE International Conference on Mobile Data Management (MDM), Paphos, Cyprus, 2022, pp. 137-146, doi: 10.1109/MDM55031.2022.00040.
>
> [5] Z. Wang, W. Zhan, H. Duan, G. Min and H. Huang, "Deep-Reinforcement-Learning-Based Continuous Workflows Scheduling in Heterogeneous Environments," in IEEE Internet of Things Journal, vol. 12, no. 10, pp. 14036-14050, 15 May15, 2025, doi: 10.1109/JIOT.2024.3524506.

---

### Meta-Review · Area_Chair_Z8vG · 2026-01-05

**Summary:**

Reviewers' concerns are mainly around the following points:

W1. Lacking more recent and stronger baselines

W2. The method is not fundamentally novel, and the proposed weighted cross-attention module and appears similar to recent heterogeneous graph neural networks

W3. Lacking analysis on training stability and the skip action design

W3. Incomplete experimental evaluation

W4. Some presentation is dense and difficult to understand

W5. Lacking necessary ablation

**Reviewer Concerns:**

Authors provided a detailed rebuttal with new results. However, the missing baseline issue seems require a systematic experiment update and a major manuscript revision, since most reviewers share this concerns and require baselines from different angles (RL, graph learning, scheduling).

**Reviewer Scores:**

The only positive reviewer DcSo participated the discussion and maintained his/her evaluation. However, it would be hard for other negative reviewers to raise their evaluation.

---

### Decision · Program_Chairs · 2026-01-26

Reject